# Distribution, microphysical properties, and tectonic controls of deformation bands in the Miocene subduction wedge (Whakataki Formation) of the Hikurangi subduction zone

Kathryn E. Elphick[1], Craig R. Sloss[1], Klaus Regenauer-Lieb[2], Christoph E. Schrank[1]

[1]School of Earth and Atmospheric Sciences, Queensland University of Technology, GPO Box 2434, Brisbane, QLD 4001, Australia.
[2]School of Petroleum Engineering, University of New South Wales, Sydney, NSW, Australia.

*Correspondence to*: Kathryn Elphick (elphickk@qut.edu.au)

**Abstract**

We analyse deformation bands related to horizontal contraction with an intermittent period of horizontal extension in Miocene turbidites of the Whakataki Formation south of Castlepoint, Wairarapa, North Island, New Zealand. In the Whakataki Formation, three sets of cataclastic deformation bands are identified: [1] normal-sense Compactional Shear Bands (CSBs); [2] reverse-sense CSBs; and [3] reverse-sense Shear-Enhanced Compaction Bands (SECBs). During extension, CSBs are associated with normal faults. When propagating through clay-rich interbeds, extensional bands are characterised by clay smear and grain size reduction. During contraction, sandstone-dominated sequences host SECBs, and rare CSBs, that are generally distributed in pervasive patterns. A quantitative spacing analysis shows that most outcrops are characterised by mixed spatial distributions of deformation bands, interpreted as a consequence of overprint due to progressive deformation or distinct multiple generations of deformation bands from different deformation phases. Since many deformation bands are parallel to adjacent juvenile normal- and reverse-faults, bands are likely precursors to faults. With progressive deformation, the linkage of distributed deformation bands across sedimentary beds occurs to form through-going faults. During this process, bands associated with the wall-, tip-, and interaction damage zones overprint earlier distributions resulting in complex spatial patterns. Regularly spaced bands are pervasively distributed when far away from faults. Microstructural analysis shows that all deformation bands form by inelastic pore collapse and grain crushing with an absolute reduction in porosity relative to the host rock between 5 and 14%. Hence, deformation bands likely act as fluid flow barriers. Faults and their associated damage zones exhibit a spacing of 9 metres on the scale of 10 kilometres and are more commonly observed in areas characterised by higher mudstone to sandstone ratios. As a result, extensive clay smear is common in these faults, enhancing the sealing capacity of faults. Therefore, the formation of deformation bands and faults leads to progressive flow compartmentalisation from the scale of 9 metres down to about ten centimetres, the typical spacing of distributed, regularly spaced deformation bands.

## 1. Introduction

Deformation Bands (DBs) are defined as tabular, oblate zones of localised plastic shear and volume change produced through inelastic yielding in granular material (porosity >5%) (Aydin, 1977, 1978; Aydin and Johnson, 1983; Okubo and Schultz, 2005; Schultz and Siddharthan, 2005). Bands often exhibit mm- to cm- offset and have lengths most commonly >100 m, but do not act as planes of significant displacement discontinuity, distinguishing them from faults (Aydin, 1977; Antonellini et al., 1994; Świerczewska and Tokarski, 1998; Aydin et al., 2006; Fossen et al., 2007). DBs are characterised by small displacement to thickness ratios and are thicker than faults of comparable length (Fossen et al., 2007). Kinematic classification recognises five types of deformation band: pure compaction bands, compactional shear bands, isochoric shear bands, dilatant shear bands, and pure dilation bands (Fossen et al., 2007; Eichhubl et al., 2010). The kinematics are commonly linked to stress state through the cam-cap model of yielding and band formation (Fig. 1) (Wong et al., 1992; Schultz and Siddharthan, 2005; Fossen et al., 2007). In this approach, the yield envelope of porous granular media is represented by a non-linear surface in Q-P space within which Q and P signify differential stress and effective mean stress, respectively:

$$Q = \sigma_1 - \sigma_3 \tag{1}$$

$$P = \left(\frac{(\sigma_1 + \sigma_2 + \sigma_3)}{3}\right) - P_f \tag{2}$$

where $\sigma_1$, $\sigma_2$ and $\sigma_3$ denote the principal stresses of the Cauchy stress tensor, and $P_f$ is pore-fluid pressure. The cam-cap yield surface links the state of stress at the time of deformation localisation to the kinematics and orientation of DBs. Current literature hypothesises that the point of intersection, and therefore the orientation and band kinematics, is controlled by the tectonic regime (Soliva et al., 2013; Ballas et al., 2014; Soliva et al., 2016; Fossen et al., 2018). During horizontal extension, higher Q and lower P are predicted, resulting in DBs characterised by a high shear component that are oblique (ca. 30-50°) to the principal compressive stress (Ballas et al., 2013; Soliva et al., 2013; Ballas et al., 2014; Soliva et al., 2016). In contrast, during horizontal contraction, higher P is predicted. This results in DBs characterised by low shear to compaction ratios that are perpendicular to sub-perpendicular to the principal compressive stress (Ballas et al., 2013; Soliva et al., 2013; Ballas et al., 2014; Soliva et al., 2016).

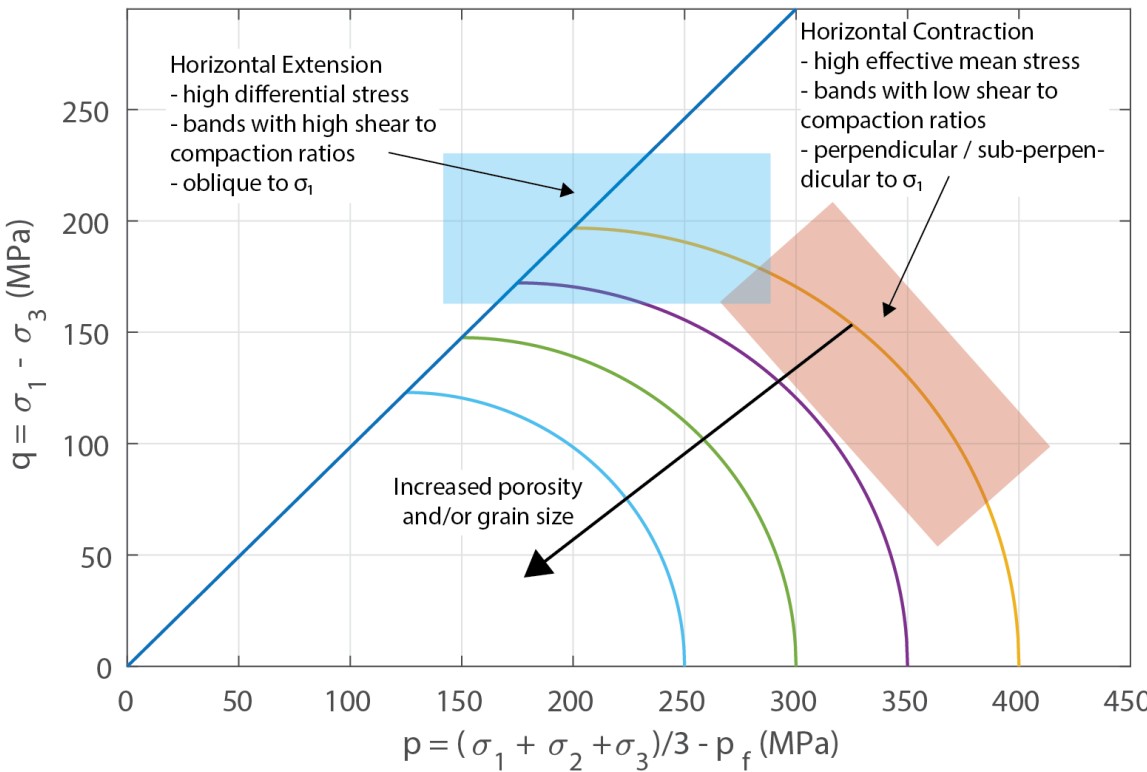

**Figure 1. Cam-cap yield envelope showing the relationship between the stress state at the point of inelastic yielding and the type and orientation of the deformation band that nucleates. In horizontal contraction, higher effective mean stresses are predicted which results in the formation of bands characterised by low shear to compaction ratios. Comparatively, during horizontal extension, higher differential stresses are predicted resulting in bands characterised by high shear to compaction ratios. The yield envelope shrinks with increased porosity and/or grain size of the material. Adapted from Fossen et al. (2007).**

DBs can also be classified by the deformation mechanism that was active during their formation. The dominant deformation mechanisms observed in deformation bands are granular flow, cataclasis, phyllosilicate smearing, and dissolution and cementation (see Fossen et al. (2007) for review). The deformation mechanism depends on the grain size, sorting, mineralogy, diagenetic history, porosity and stress state (Fossen et al., 2007) and controls petrophysical properties (Fossen and Bale, 2007; Ballas et al., 2015; Fossen et al., 2018). In nature, shear bands with associated compaction are most commonly observed, formed by grain rearrangement and porosity collapse with or without cataclasis (Fossen et al., 2007). Cataclasis is often the dominant deformation mechanism in bands formed >1 km depth. Disaggregation bands form via granular flow and are dominant at shallower levels (Fossen et al., 2007). Additionally, if the host rock contains >10% platy minerals, phyllosilicate bands, characterised by fine-grained, low-porosity zones containing aligned phyllosilicates can form (Knipe et al., 1997; Fossen et al., 2007). In these bands, platy minerals promote frictional grain boundary sliding and inhibit grain fracturing (Fossen et al., 2007). In clay-rich rocks (>40%), deformation can produce clay smears with very low permeability that commonly accumulate greater offsets than other types of bands (Antonellini et al., 1994; Fisher and Knipe, 2001; Fossen et al., 2007).

Besides its relevance for understanding strain localisation in rocks, the study of DBs is important because of their sealing potential and effect on fluid flow in hydrocarbon and groundwater reservoirs (Antonellini and Aydin, 1994; Ogilvie and Glover, 2001; Fossen and Bale, 2007; Fossen et al., 2007; Ballas et al., 2013; Ballas et al., 2015; Qu et al., 2017; Fossen et al., 2018). DBs commonly exhibit a reduction in permeability compared to the host rock, subsequently acting as barriers to fluid flow (Ogilvie et al., 2001; Ogilvie and Glover, 2001; Fossen and Bale, 2007; Fossen et al., 2007; Ballas et al., 2015; Fossen et al., 2018). The relative proportion of clay incorporation and cataclasis controls the petrophysical properties of the bands. Bands characterised by greater concentrations of clay and greater degrees of cataclasis are more effective barriers (Fossen et al., 2007; Ballas et al., 2015; Fossen et al., 2018). The extent to which DBs can impact reservoir fluid flow depends not only on their petrophysical properties but also their spatial distribution (Ogilvie and Glover, 2001; Sternlof et al., 2006; Fossen and Bale, 2007; Torabi et al., 2013; Ballas et al., 2015). In nature, deformation bands exhibit two spatial distributions: networks of bands concentrated into clusters or zones in the vicinity of faults (Antonellini and Aydin, 1995; Shipton and Cowie, 2001, 2003; Soliva et al., 2013; Ballas et al., 2014), and a pervasive periodic and/or clustered distribution across a deformed region, unrelated directly to faults (Saillet and Wibberley, 2010; Ballas et al., 2013; Soliva et al., 2013; Soliva et al., 2016). Field studies suggest that the spatial distribution of deformation bands is largely controlled by the tectonic regime in which they form (Saillet and Wibberley, 2010; Solum et al., 2010; Soliva et al., 2013; Ballas et al., 2014; Soliva et al., 2016). Deformation bands associated with an extensional regime are observed in the damage zone of faults where the band density increases as the fault plane is approached (Shipton and Cowie, 2003; Schueller et al., 2013). Conversely, deformation bands formed during horizontal contraction exhibit distributed, regular spacing (Ballas et al., 2013; Soliva et al., 2013; Fossen et al., 2015). However, the control of tectonic regime has only be tested in one field site (Provence, France) where host rock, comprising mainly quartz-rich deltaic sandstone characterised by large heterogeneities in terms of grain size and porosity, contains bands formed in horizontal extension and horizontal contraction (Saillet and Wibberley, 2010; Ballas et al., 2013; Soliva et al., 2013; Ballas et al., 2014; Soliva et al., 2016). Thus, other field studies are required to assess the control of tectonic regime.

Here, we analyse faults and deformation bands formed in poly-deformed Miocene siliciclastic turbidites of the exposed Hikurangi subduction wedge in eastern North Island of New Zealand to test if the influence of tectonic regime documented in previous studies applies. The studied rock package has been deformed in both horizontal extension and horizontal contraction, with strike-slip reactivation of some extensional faults. The host rock comprises interbedded sand-, silt-, and mudstones of the Whakataki Formation (Lee and Begg, 2002). Previous studies on the relationship between deformation band distribution and the tectonic setting analysed deformation bands hosted within mineralogically relatively homogeneous sandstones in two key locations: [1] Utah, USA, where the deformation bands are associated with the intracontinental Laramide orogeny (Aydin, 1978; Antonellini et al., 1994; Solum et al., 2010; Fossen et al., 2011), and [2] Provence, France, where bands formed during the Cretaceous Pyrenean intracontinental mountain building event and the subsequent Oligocene-Miocene rifting (Saillet and Wibberley, 2010; Ballas et al., 2013; Soliva et al., 2013; Ballas et al., 2014; Soliva et al., 2016). In addition, band kinematics and orientations have been studied in DBs that are hosted in the Nubian Sandstone, Western Sinai, Egypt that formed during Oligocene-Miocene rifting associated with the separation of the African and Arabian plates (e.g., Rotevatn et al., 2008;

Tueckmantel et al., 2010). In Utah, Provence, and Western Sinai, the host rocks mainly consist of quartz. The DBs exhibit porosity reduction and grain size reduction due to cataclasis compared to the host rock. The magnitude of porosity and grain size reduction depend on the burial depth at the time of deformation. The units studied in this research are compositionally less mature than the host rocks of the previous study sites. With greater clay content and more lithological heterogeneity, different deformation mechanisms are expected (Fossen et al., 2007).

DBs hosted within units that are characterised by greater lithological heterogeneity and higher clay content have been studied primarily through core data collected from the North Sea (Knipe et al., 1997; Fisher and Knipe, 1998; Knipe et al., 1998; Fossen and Hesthammer, 2000; Fisher and Knipe, 2001; Farrell et al., 2014). In these bands, extensive grain-scale mixing of sand and clay is observed (Fisher and Knipe, 2001; Kristensen et al., 2013). Some clay layers form clay smears that maintain integrity for distances several times that of the layer thickness (Kristensen et al., 2013). Deformation bands deformed at <1.4 km depths exhibit grain size reduction by cataclasis in addition to grain-scale mixing, consistent with previous studies showing increased cataclasis at >1 km depth (Antonellini et al., 1994). However, bands deformed at depths below ca. 1.5 km in mixed sequences have not been described in detail. Deformation bands formed in impure sandstones (15-40% clay) at >3 km depth within the Rotliegendes reservoir are uncommon and are thus not discussed (Fisher and Knipe, 2001). Therefore, a comprehensive description of the orientation, kinematics, and spatial distribution of deformation bands hosted in mixed sequences and deformed at >1.5 km depth is presently missing from the literature and is addressed in this research. Furthermore, the Whakataki Formation has been deformed in a subduction wedge (Cape et al., 1990; Chanier and Ferrière, 1991; Rait et al., 1991; Chanier et al., 1999; Nicol et al., 2007; Bailleul et al., 2013). While this tectonic setting attracts significant research attention because of its association with megathrust earthquakes and hydrocarbon reservoirs, studies of deformation bands within these settings are limited due to complex structural overprints by later-stage deformation structures. The studies that have analysed deformation bands in an active accretionary prism (Moore, 1986; Karig and Lundberg, 1990; Labaume et al., 1997; Maltman, 1998; Ujiie et al., 2004) have not done so with the primary goal of testing how the tectonic setting and stress state influences the kinematics and distribution of bands. Bands in accretionary prisms are mainly observed in the toe of the prism, are associated with shallow (<500 m) deformation and are only observed in silts or clays. Through continual accretion and internal deformation, many rock packages travel deep into the prism reaching burial depths of >5 km. This causes overprint of structures that form in the upper few kilometres of the wedges, thus limiting our understanding of shallow deformation processes (e.g., Maltman, 1998). The Whakataki Formation travelled along a shallow trajectory through the Hikurangi subduction wedge, with a maximum burial depth of 3-4 km (Wells, 1989). Analysis of the structures present within the Whakataki Formation can improve our understanding of how siliciclastic rocks accommodate deformation in the shallow regions of subduction wedges. Moreover, the chosen field site also offers the opportunity to study deformation bands hosted in a mixed clay-sand sequence that have formed under horizontal contraction. Finally, the Whakataki Formation has been buried to depths within the hydrocarbon formation window (2-3 km) (e.g., Bustin, 1991) and overlies potential source rocks of the underlying Waipawa Formation thereby making the unit a potential reservoir (Leckie et al., 1994; Field et al., 2006). Previous studies have analysed rocks buried to <2 km depth. For example, the sandstones of the Provence study site

were buried to 0.4-0.6 km (Soliva et al., 2013) while those in Utah were buried to 1-2 km (Fossen et al., 2011). Since DBs are commonly associated with a reduction in permeability and porosity (Fossen and Bale, 2007), it is useful to study their spatial distribution and microphysical properties in potential hydrocarbon reservoir rocks (Field et al., 2006) that travelled through the hydrocarbon window. Preliminary offshore drilling of the Miocene siliciclastic rocks has documented significant gas shows (Amoco New Zealand Exploration Ltd, 1992; Field et al., 2006; Archer et al., 2014), and a study of the onshore analogue could provide useful insights into the reservoir architecture at depth (Field et al., 2006). Therefore, our work complements existing ones by considering rocks with different host rock properties, which are actively targeted by hydrocarbon exploration (Lee and Begg, 2002; Hessler and Sharman, 2018).

## 2. Geologic setting

We study faults and deformation bands in the Whakataki Formation, an exhumed turbidite package of the subduction wedge of the Hikurangi subduction zone (Chanier and Ferrière, 1991; Neef, 1992a; Neef, 1992b, 1995; Chanier et al., 1999; Field, 2005; Nicol et al., 2007; Nicol et al., 2013). The Hikurangi subduction zone initiated ca. 25 Ma ago and accommodates oblique convergence between the Pacific and Australian plates (Walcott, 1984; Ballance et al., 1985; Walcott, 1987; Ballance, 1988; Chanier and Ferrière, 1991; Rait et al., 1991; Luyendyk, 1995; Chanier et al., 1999; Nicol et al., 2007; Spörli, 2009; Bailleul et al., 2013; McCoy-West et al., 2013; Mortimer et al., 2017; Strogen et al., 2017).

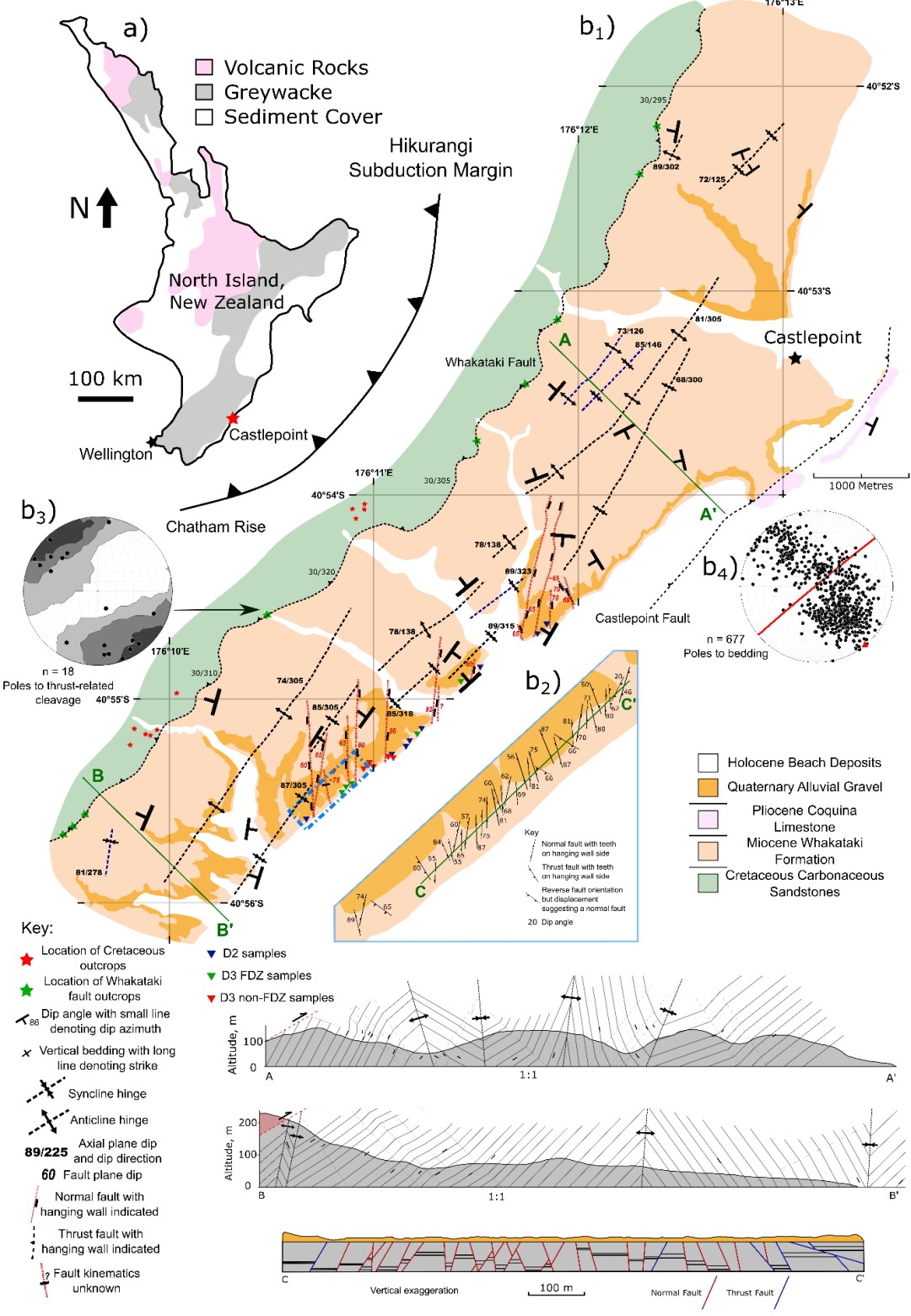

**Figure 2. a) Simplified geological map (Edbrooke, 2017) of New Zealand's North Island with study site location (red star) (coordinates: NZGD 2000, UTM Zone 60S, E: 0430670, N:5468211). b$_1$) Simplified geological map of the research area. The ominant structures in the field site are NE-SW trending folds and thrust faults with N-S trending normal faults along the coastline. Below are cross-sections through the area. A-A' and B-B' highlight the asymmetrical fold geometry with synclines resembling box-folds and anticlines resembling kink-folds. The unconformable overlying Quaternary sediments have not been plotted to highlight the geometry of the folds. C-C' is a schematic cross-section drawn perpendicular to the average strike of normal faults. An arbitrary geological bed has been added to show possible displacements associated with faults. b$_2$) A detailed map of fault location and orientation along a ca. 1 km stretch of coastline. Normal faults are dominant, with thrust faults found in areas of low normal-fault density. b$_3$) Schmidt net with poles to cleavage planes observed in the damage zone of the Whakataki Fault, a large-scale thrust fault associated with the first phase of compression, D1. b$_4$) Schmidt net of poles to bedding planes. The mean axial plane for all bedding is plotted as great circle (red line). Given that the axial plane trends parallel to the traces of the dominant thrust faults in the area, it is reasonable to assume, by assessing the poles to the axial planes, that the folding was induced by SE shortening. All Schmidt nets are plotted with the software Stereonet 10 of Allmendinger et al. (2011).**

The onset of subduction is expressed in the sedimentary record by the wide-spread deposition of olistostrome deposits in the earliest Miocene (Chanier and Ferrière, 1991; Bailleul et al., 2007; Bailleul et al., 2013), which also define the base of the studied Whakataki Formation. The basal olistostrome is overlain by deep-marine high energy flysch deposits (Neef, 1992a), consisting of a succession of laterally continuous fine-grained sandstone and siltstone turbidite beds with total estimated thickness of 900-1500 m (Neef, 1992a; Field, 2005; Bailleul et al., 2013).

The present study is based on field observations and samples collected from coastal exposures of the Miocene Whakataki Formation 5 km southwest of Castlepoint, in the Wairarapa region of the North Island of New Zealand (Fig. 2, coordinates: NZGD 2000, UTM Zone 60S, E: 0430670, N:5468211). The Whakataki Formation (deposited between ca. 25-17.5 Ma (Neef, 1992a; Field, 2005; Bailleul et al., 2013; Raine et al., 2015)) was emplaced in tectonically controlled confined basins on the lower trench-slope of the subduction margin (Bailleul et al., 2007; Bailleul et al., 2013). The overall sedimentary succession preserves a record of the onset of subduction associated with horizontal contractional deformation between ca. 25-19 Ma (Deformation phase 1 - D1), followed by a period of horizontal extension (ca. 15-5 Ma; Deformation phase 2 – D2) and renewed contraction from the Pliocene to Recent (Deformation phase 3 - D3) (Chanier and Ferrière, 1991; Chanier et al., 1999; Bailleul et al., 2007; Bailleul et al., 2013). Each deformation phase induced corresponding tectonic structures in the study area (Figs. 2 & 3):

D1: Margin-perpendicular contraction expressed by both landward and seaward emplacement of thrust-sheets, gentle folding, and reverse faulting along the Hikurangi margin, with the development of trench-slope basins bounded by structural highs (Chanier and Ferrière, 1991; Rait et al., 1991; Chanier et al., 1999; Bailleul et al., 2013; Maison et al., 2018). The main D1 structure in the field area is the NE-SW trending Whakataki thrust fault that has emplaced the Late Cretaceous Whangai Formation (Neef, 1992a) onto the Miocene Whakataki Formation and acts as the western boundary to the Whakataki Formation in that area (Fig. 2).

D2: An overall deepening of the forearc domain (Crundwell, 1987; Wells, 1989) and widespread subsidence involved margin-parallel and margin-perpendicular extension of the subduction wedge accommodated by normal faulting (Chanier et al., 1999; Bailleul et al., 2007; Bailleul et al., 2013). Tectonic erosion, with associated syn-sedimentary gravitational collapse, has been

proposed for being responsible for this reversal in stress regime (Chanier et al., 1999; Bailleul et al., 2013). The dominant D2 structures in the field area are ca. N-S striking normal faults.

D3: A renewal of dominant margin-perpendicular contraction of the wedge with associated folding, thrusting, and strike-slip faulting (Cape et al., 1990; Chanier et al., 1999; Nicol et al., 2002; Bailleul et al., 2007; Nicol et al., 2007; Bailleul et al., 2013). Also associated with this deformation phase is the uplift and exposure of the Coastal Ranges (Nicol et al., 2002). In the field site, this deformation phase is expressed by pervasive deformation bands, small thrust faults, and intense folding. Deformation bands have been identified in the Whakataki Formation previously (Nicol et al., 2013). However, they have not been described and analysed in detail.

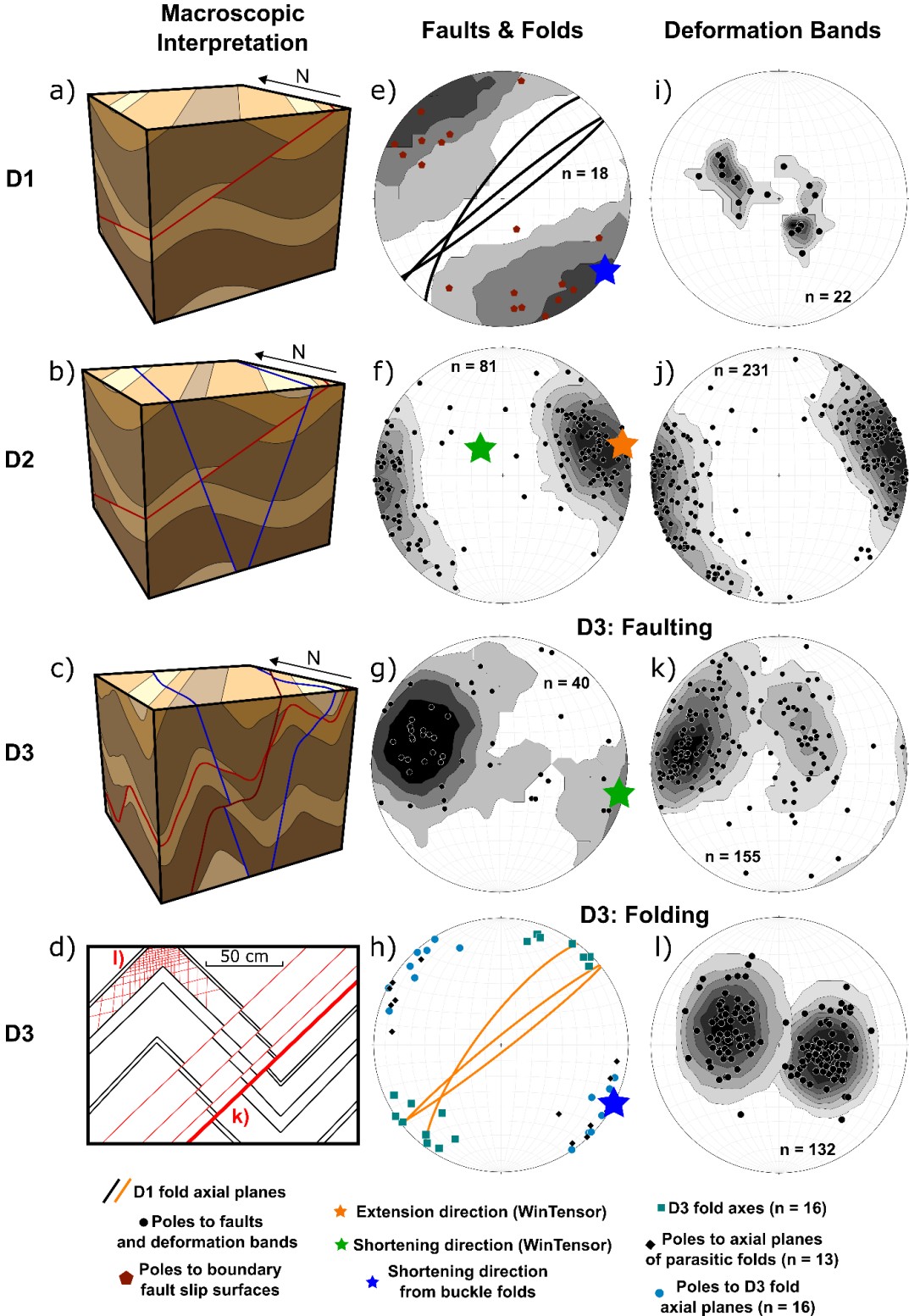

**Macroscopic Interpretation**

**Faults & Folds**

**Deformation Bands**

a) D1

e) n = 18

i) n = 22

b) D2

f) n = 81

j) n = 231

c) D3

**D3: Faulting**

g) n = 40

k) n = 155

d) D3

50 cm

l)

k)

**D3: Folding**

h)

l) n = 132

/ /  D1 fold axial planes

● Poles to faults and deformation bands

⬠ Poles to boundary fault slip surfaces

★ Extension direction (WinTensor)

★ Shortening direction (WinTensor)

★ Shortening direction from buckle folds

■ D3 fold axes (n = 16)

◆ Poles to axial planes of parasitic folds (n = 13)

● Poles to D3 fold axial planes (n = 16)

**Figure 3. Tectonic evolution of the mapping area with each tectonic regime, with corresponding structural data for each phase. a), b), c): schematic block diagrams of each deformation phase. d) sketch of deformation associated with D3. During D3, two clear distributions of deformation bands are observed: deformation bands associated with faults and deformation bands observed far from faults. Deformation bands observed in D2 are characterised by the same orientations as D3 DBs associated with faults (k). e) Schmidt net of D1 parasitic fold orientations (planes) with poles to slip surfaces observed in the damage zone of the large-scale Whakataki thrust fault. f), g): Schmidt nets of fault orientation data from D2 and D3 deformation, respectively. $\sigma_1$ is indicated by the green star. $\sigma_3$ is indicated by the orange star. h) The orientation of F3 folds with F1 fold axial planes plotted to highlight the similar orientations across both events. The blue star represents the shortening direction, estimated from the pole to the average fold axial plane. i) Poles to D1 deformation bands. j) Poles to D2 deformation bands. k) Poles to D3 fault-damage-zone-associated deformation bands. l) Poles to D3 deformation bands that are not located in fault damage-zones. The similarity of the fault and fault-associated deformation band orientations indicates that they formed in the same stress field. Also, axial plane data from D3 aligns with D3 non-fault damage-zone associated bands as shown in (l). All deformation band and fault data are back-tilted as there is evidence of folding occurring coeval to, or later than all events. Stereoplots were produced using Stereonet 10 software (Allmendinger et al., 2011). Paleostress analysis completed using WinTensor (Delvaux and Sperner, 2003).**

## 3. Methods

### 3.1. Field Data

Field mapping covered a 17 km$^2$ area south of Castlepoint (coordinates: NZGD 2000, UTM Zone 60S, E: 0430670, N:5468211), in the Wairarapa region of the North Island of New Zealand (Fig. 2). While significant hinterland mapping took place, coastal outcrops were the focus of the fieldwork because exposure is poor in the hinterland. A detailed sedimentological analysis was conducted on sediments of the uplifted Hikurangi subduction wedge to contextualise the structural data and will be presented elsewhere. Orientations of structural elements including bedding planes ($S_0$), faults, and deformation bands were taken across the area and plotted in lower-hemisphere, equal-area stereograms (Allmendinger et al., 2011). All data are shown with bedding restored to horizontal. The restoration was completed by rotating back from the associated $S_0$ measurement of the bed hosting the structures using Stereonet 10 (Allmendinger et al., 2011). Individual $S_0$ measurements were used rather than fold axes because the folds are non-cylindrical and plunge gently. All data were restored as there is evidence of rotation in almost all features (Figs. 2 & 5). Paleostress analysis of back-tilted faults was completed using Win-Tensor (Delvaux and Sperner, 2003). For this analysis, only faults with unambiguous slip vectors and shear sense were used. The PBT-kinematic-axes method was used to find the orientation of the principal stresses. An assessment of whether faults and deformation bands form a conjugate (bimodal) or polymodal pattern was conducted using a statistical test developed by Healy and Jupp (2018). The test analyses the orientation distribution of the poles to fault planes to distinguish between bimodal and polymodal patterns. Differentiation between bimodal and polymodal is established by first finding the ratio between the eigenvalues of the second-rank orientation tensor of the datasets and secondly by calculating the *p-value*. *P*-values close to 0 describe a polymodal pattern with higher values reflecting bimodal patterns. The dihedral angle, the angle bisected by $\sigma_1$ (Chemenda et al., 2012), between conjugate sets of deformation bands was estimated from a cylindrical best fit to orientation data. The ratio of net shear/compaction ($D_s$(net)/$D_c$) was calculated using the methods from Soliva et al. (2013) and Ballas et al. (2014). These values are used to define the kinematics of the deformation bands. However, these authors assumed that they cut their samples in the plane spanned by the smallest and largest principal strains, the orientation of which they determined from the rather tightly

bimodal orientation patterns of the studied bands. In our case, this approach is more difficult because the orientation distributions are less focussed (Fig. 3). Therefore, typical geometric section errors in terms of true bandwidth and true shear displacement can be expected.

### 3.2. Spacing Analysis

The spacing analysis was conducted using image analysis on two scales in the mapping area: [1] on macroscopic faults with >20 cm offset exposed along a 5 km stretch of coastline, and [2] on deformation bands at individual outcrops exposed in sandstone layers. The method used for analysis is a simplified version of the line sampling method outlined in Sanderson and Peacock (2019). For macroscopic fault spacing, the location, dip and dip direction, heave and throw, where possible, and shear-sense were recorded, and faults plotted onto a map. The spacing of macrofaults was measured along scan-lines oriented perpendicular to the average fault strike using the ruler tool in ImageJ (Schindelin et al., 2012). The scan-lines were 200 m in length in an E-W orientation (orientation perpendicular to the average fault orientation) and had a 50 m N-S spacing. Due to the oblique relationship between the coastline and the fault strike, scan-lines were shifted N after 400 m, with the first scan-line located in the SW of the coastline. Twenty scan-lines were used to measure the fault spacing, and the median is reported. The use of multiple scan-lines minimises, but does not eradicate, measurement errors arising from variation in fault strike and an average strike being used to generate the scan-line orientation (Sanderson and Peacock, 2019). Spacing values greater than 20 m were removed from the data as these points represent artefacts of exposure conditions. Horizontally measured spacing was corrected for the average dip of the faults using the sine transformation (Eq. 3) (Sanderson and Peacock, 2019).

True spacing $=$ spacing $\times$ sin (average dip angle) $\hspace{3cm}$ (3)

The correction procedure also introduces errors because the faults do not all dip the same way. Moreover, they were not rotated to their original position and orientation in space. However, these are common shortcomings of studying fracture spacing in 2D (Soliva and Benedicto, 2005; Laubach et al., 2018). Here, we are mainly interested in the general shape of the spacing distribution and the order of magnitude of absolute spacing. Pearson correlation coefficient values were calculated for the correlation between the corrected spacing and the associated cumulative distance from the start of the first scan-line. For spacing analysis of deformation bands, outcrops had to meet the following conditions: [1] the outcrop surface was fairly planar, and [2] the deformation bands were striking roughly perpendicular to the outcrop surface. Along the coastal outcrop, twelve locations showing deformation bands within damage zones (fault-damage-zone (FDZ) bands) and twenty-eight locations with bands showing seemingly regular spacing, not adjacent to faults (non-FDZ bands), were selected. Photographs were taken perpendicular to the outcrop surface. Intersections of the bands with the flat outcrop surface were mapped on a suitable photographic mosaic of the outcrop. For outcrops showing mutually overprinting conjugate sets of non-FDZ deformation bands, the dominant set was analysed.

For spacing measurements of deformation bands in damage zones, the fault was put into the origin of the digitised image. For non-FDZ bands, with apparent 'regular' spacing, the first band was treated as the origin of the image. Deformation band clusters were treated as single bands following Main et al. (2000). The resulting maps were exported as a binary image. Matlab

(Mathworks, 2011) was used to obtain spacing statistics along scan-lines at a spacing of 1% of the image height. When the horizontal image dimension was not perpendicular to the traces of the deformation bands in the map, spacing was again corrected with the sine transformation (Eq. 3). Normalisation was completed by dividing the spacing value by the width of the image. This permitted comparison of absolute spacing values between outcrops.

In addition to the analysis of natural deformation band distributions, synthetic images were created to show ideal spacing distributions and to highlight how natural variation in heterogeneous rocks and data collection error can impact the measured spacing of natural samples. Six synthetic images were produced:

[1] Deformation bands with constant spacing;

[2] Deformation bands with constant spacing and added noise to replicate measuring bias and outcrop conditions. The noise in image [2] was generated by adding an array of random values, between 0 and 0.8 of the constant spacing value, to the space. The random values are collectively characterised by a normal distribution and a median value of 0;

[3] Deformation bands spatially characterised by an exponential spatial decay away from a fault;

[4] Deformation bands characterised by an exponential spatial decay away from a fault with integrated noise. Noise in the image [4] is generated by adding values up to 0.4 of the maxima spacing onto each spacing measurement, with added values collectively characterised by a normal distribution with a median value of 0;

[5] Deformation band spacing that reflects the overprint of an equally spaced distribution [1] by an exponentially decaying damage zone [3]; and

[6] Deformation band spacing reflecting the overprint of an equal distribution with integrated noise [2] by an exponentially decaying damage zone also containing integrated noise [4].

### 3.3. Microstructural Analysis

Twenty polished thin-sections from samples of host rock and deformation band were examined by back-scattered electron (BSE) imaging using a VP Zeiss Sigma scanning-electron microscope (SEM). A 10 nm carbon-coat was applied to the samples. The instrument was run using a working distance of 8 mm, an acceleration voltage of 15 kV, an aperture size of 30 µm with an angle-selected backscattered detector. Images of host rock were taken at 400 x magnification and those of deformation bands at 800 x. Multiple images were taken with a 5-10% overlap and stitched together to produce ca. 1 mm$^2$ images. Of the 20 samples, 5 were taken from zones showing a 'regular' spacing of D3 deformation bands with zero to minimal apparent offset. The remaining 15 samples are from fault damage zones, 10 from D2 damage zones, and 5 from D3 damage zones (Fig. 2). For each sample, the host rock and deformation band were analysed in the same thin section. Samples were selected from different locations along the coastline in attempts to analyse a broad range of lithologies. Sample locations of bands with a 'regular' spacing are more clustered as these were the most appropriate outcrops with good exposure of the facies hosting non-FDZ bands.

BSE maps were analysed for porosity and mineralogy of the samples. Grey-level slicing was used to extract four different phases from the images [1] porosity, [2] quartz, [3] feldspar, and [4] 'other' which includes reflective oxide components, micas, and clays. For porosity phase analysis, the images were eroded and dilated to generate upper and lower bounds for the estimate (Liu and Regenauer-Lieb, 2011). Grain size estimates were obtained through manual tracing of grains from SEM images in ImageJ (Schindelin et al., 2012). An average of 100 grains were traced for the host rock and the deformation band of each sample, and the equivalent circle diameter was calculated for each polygon.

## 4. Results

### 4.1. Rock Descriptions and Structure

All three regional deformation phases discussed above (Section 2) can be recognised in the Whakataki Formation. Expressions of the deformation in the Whakataki Formation include folds, faults, and deformation bands. Deformation bands are most common within sandier units in areas characterised by equal sandstone to mudstone ratios, or areas where sandstone beds are dominant. In areas dominated by mudstone, bands are less common. Figure 4 shows examples for a sand-dominated facies and a facies characterised by equal proportions of sand- and mudstone.

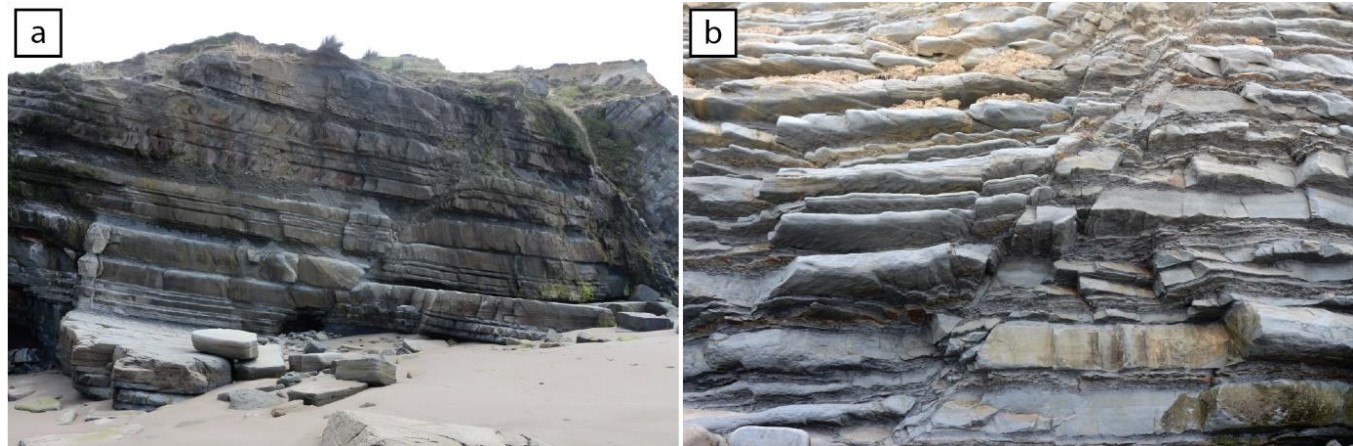

**Figure 4. Field images of the different facies of the Whakataki Formation that dominantly host deformation bands and faults. [a] An outcrop scale image of a sandstone-dominated sequence. [b] An outcrop scale image of a sequence characterised by equal ratios of sand and mud. Within this sequence more faults are observed compared to sandier sequences.**

In the following, we describe structures associated with each tectonic phase. Throughout this description, we use Anderson's nomenclature (Anderson, 1951) where the normal-faulting regime reflects a (sub-)vertical $\sigma_1$ (horizontal extension) and a thrust-faulting regime reflects a (sub-)horizontal $\sigma_1$ (horizontal contraction).

### 4.2. D1 Horizontal Contraction

#### 4.2.1. Faults and Folds

The main regional fault associated with D1 horizontal contraction is the NE-SW trending Whakataki thrust fault (Fig. 2) (Bailleul et al., 2013; Maison et al., 2018). Rare exposures of the thrust damage zone contain slip planes which indicate a NW dip (30°). NE-trending km-scale folds in the hinterland define the topography (Fig. 2). With poor hinterland exposure, it is not possible to determine the timing of large-scale fold nucleation. Regional studies indicate that large-scale folding was occurring during D1 and that these folds were tightened to their current morphology during D3 (Cape et al., 1990; Chanier et al., 1999;

Bailleul et al., 2013). Rare metre-scale, NE-trending upright, moderately NE-SW-plunging, open F1 folds are still preserved in the area (Figs. 3a & 3e).

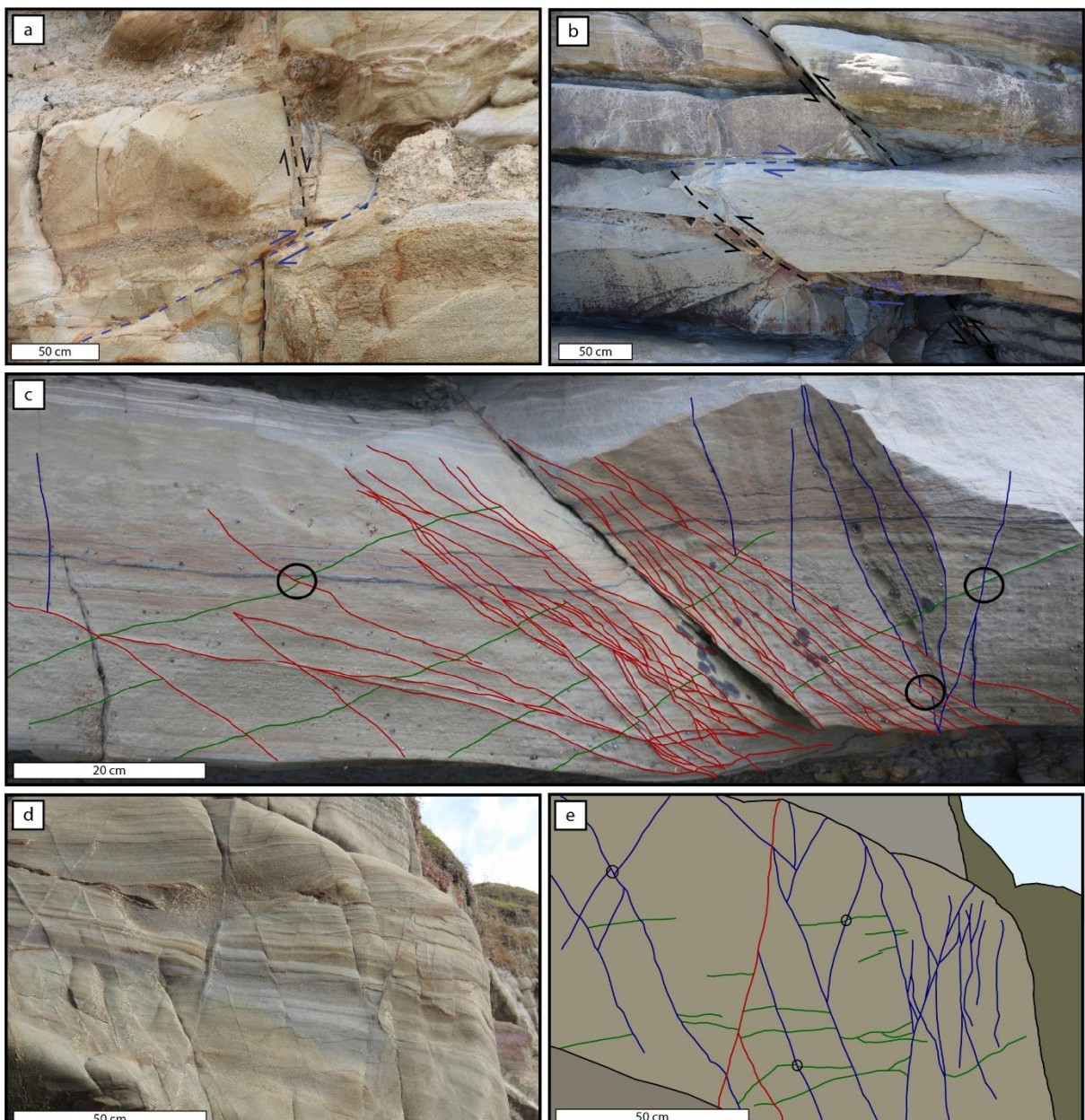

**Figure 5. Examples of overprinting criteria used to determine the relative timing of each deformation phase. [a] D3 reverse fault offsetting D2 normal fault; [b] layer-parallel slip offsetting a D3 reverse fault; [c] multiple generations of deformation band located within a D3 damage zone (green = D1, blue = D2, red = D3); [d] and [e] multiple generations of deformation band with overprints (green = D1, blue = D2, red = D3). Black circles in [e] signify some examples of overprints.**

### 4.2.2. Deformation Bands

D1 DBs are rare and are distinguished from D2 and D3 DBs through cross-cutting relationships (Figs. 5c, 5d, 5e). The orientation distribution of D1 DBs can be considered bimodal with a *p-value* of 0.5. D1 DB dihedral angles range from 65° to

84° with a mean of 73°. With bedding restored to horizontal, DBs trend 037° (Fig. 3i). The set dipping SE has a dip angle of 40° and the set dipping NW dips at 26° (Fig. 3i). When bands pass through beds with thin clay-rich layers, they have a dark colour without relief. In beds with lower clay content, bands show high to moderate relief and are lighter in colour (Figs. 5d & 5e). The width of bands ranges from 0.2 to 0.5 mm. Bands can extend for metres along strike. Eye and ramp structures are present. However, single strands are most commonly observed (Antonellini and Aydin, 1995). Offset associated with the bands is variable, with some characterised by minimal offset, yet others accommodate reverse shear offsets at the millimetre scale. Conclusively, these observations suggest that these bands are CSBs and Shear-Enhanced Compaction Bands (SECBs) formed during D1 horizontal contraction (Ballas et al., 2014; Fossen et al., 2018; Schultz, 2019). The rarity of D1 deformation bands does not permit for meaningful spacing analysis.

### 4.3.    D2 Horizontal Extension

#### 4.3.1.    Structures and Relative Timing

Field overprinting criteria demonstrate that an extensional event followed D1 horizontal contraction (Fig. 5) (Chanier et al., 1999). Overprinting relations include the displacement of F1 fold hinges by D2 normal faults and normal-sense shear displacement of D1 deformation bands by D2 deformation bands (Figs. 5c, 5d, 5e). The structures associated with D2 are [1] normal faults, [2] deformation bands with clear normal-sense offset, and [3] deformation bands with minimal offset (Figs. 2, 5a, 6, 7).

#### 4.3.2.    Faults

Normal faults are the main structures associated with D2 (Figs. 4a & 4b). They trend NNW with an average dip angle of 69° (Fig. 3f). Analysis of the fault pattern shows that it is polymodal with a *p-value* of 0.001 (Healy and Jupp, 2018). While polymodal across the field area, normal faults are commonly observed as two sets with opposing dips at individual outcrops (Fig. 3f). Fault displacement ranges from tens of centimetres to tens of metres. Many D2 faults have displacements of ca. 1 m (e.g., Fig. 7). The limited vertical extent of the cliff exposures and poor hinterland exposure preclude estimates of upper-bound displacement (Fig. 4). Fault length can also not be determined as they generally extend beyond cliff height. The faults are dominantly brittle, with gouge present in the core of larger faults (Figs. 4 & 6d).

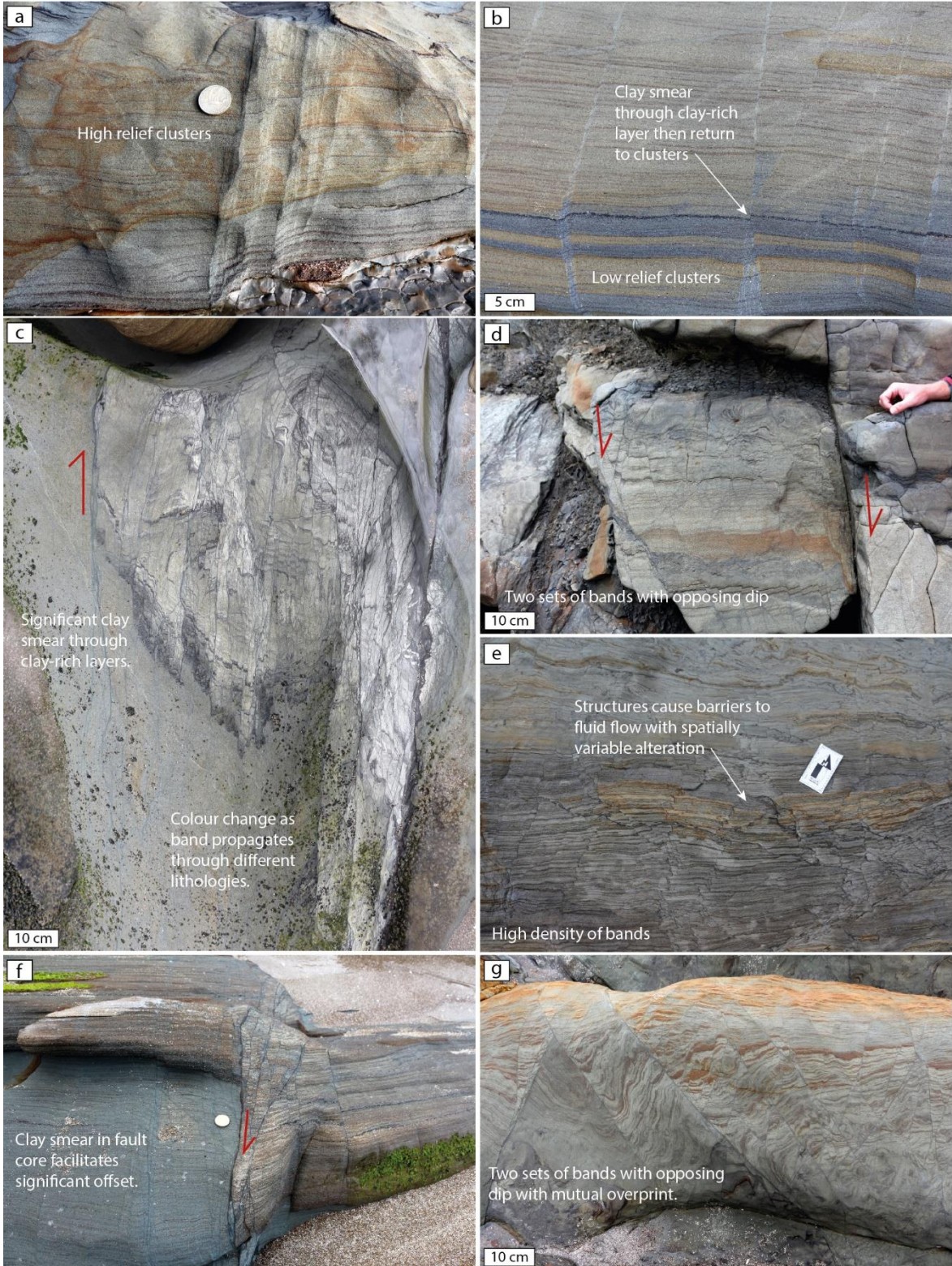

a — High relief clusters

b — Clay smear through clay-rich layer then return to clusters
Low relief clusters
5 cm

c — Significant clay smear through clay-rich layers.
Colour change as band propagates through different lithologies.
10 cm

d — Two sets of bands with opposing dip
10 cm

e — Structures cause barriers to fluid flow with spatially variable alteration
High density of bands

f — Clay smear in fault core facilitates significant offset.

g — Two sets of bands with opposing dip with mutual overprint.
10 cm

**Figure 6. Outcrop images of faults and deformation bands associated with normal faulting. [a] Cluster of normal-sense compactive shear bands characterised by high relief and <1 cm offset. [b] Lighter coloured normal-sense shear bands characterised by low relief. As a cluster propagates through a clay-rich layer it reduces in width to a single surface. [c] High density of normal-sense shear bands characterised by several centimetres of offset. The bands are hosted in the damage zone of a fault with >10 m offset. Offset cannot be constrained due to exposure conditions. The bands change colour as they propagate through different parts of the host rock with**
**bands towards the top of the image characterised by darker colours after passing through clay rich layers. To the left of the image a slip surface (red arrow) is present which removes the clay-rich layer from the image plane. [d] An interaction damage zone between two normal faults. Deformation bands within the damage zone have low relief and a darker colour than the host rock indicating clay incorporation into the band. [e] High density of normal-sense shear bands causing the host rock to be significantly compartmentalised. This is supported by the variable colour of the host rock indicating that later fluid flow is localised and restricted.**
**[f] Normal fault plane with a damage zones comprising many deformation bands that are darker in colour than the host rock and exhibit negative relief. [g] A dense array of two sets of normal sense shear bands. All examples cut $S_0$ and thus allow shear-sense determination.**

Paleostress analysis of 16 back-tilted normal faults indicates ENE-WSW extension ($\sigma_1$: 62/298° and $\sigma_3$: 23/080°) (Fig. 3f),

consistent with the more comprehensive D2 paleostress analysis presented by Chanier et al. (1999) for the North Island east

coast.

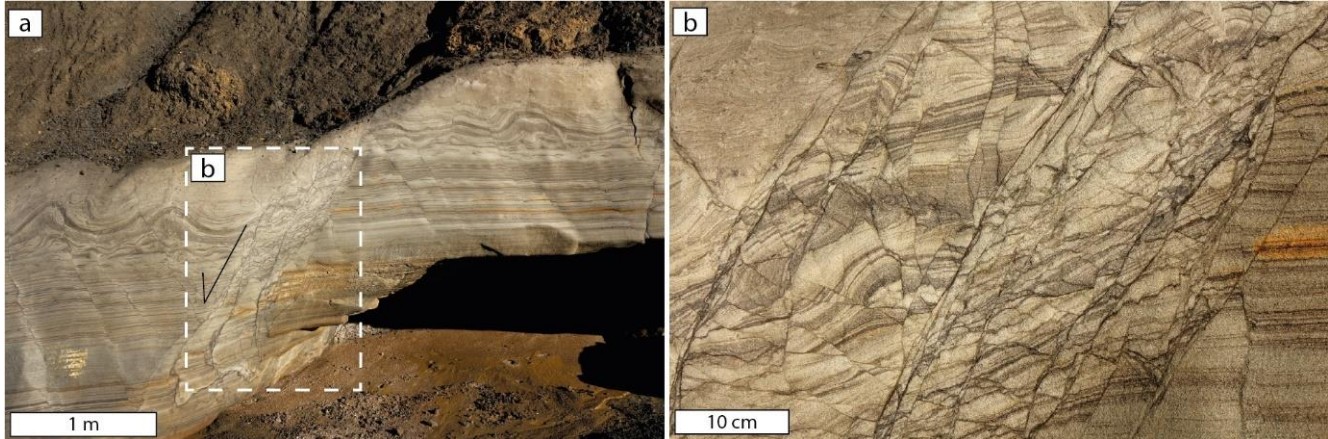

**Figure 7. Outcrop image of a normal fault hosted within a mudstone-rich sequence. The fault accommodates ca. 1 m of displacement in the image plane. The fault core consists of a dense array of shear bands. Adjacent to the fault core are deformation bands and shear bands with similar and opposing dip. All structures accommodate normal-sense shear displacement. [b] The compactive shear**
**bands accommodate displacement and can link to form a dominant slip surface. DBs have a darker colour compared to the host rock and recessive relief. Sedimentary layering is displaced with normal-sense shear offset. Within the fault core, DBs dominantly share the same orientation as the fault displacement. However, bands with variable orientations are present indicating high strain within the fault core.**

### 4.3.3. Deformation Bands

D2 deformation bands are primarily observed in the damage zones of D2 faults (Figs. 6c, 6d, 6f). However, some single bands

are observed between faults (Fig. 6b). They are generally darker in colour compared to the host rock and show no or negative

relief (Figs. 6c-g). When suitable host-rock layers are present, clay smear is common (e.g., Fig. 6c). When propagating through

layers rich in shell hash or coarser grains with less lithological heterogeneity, the bands show positive relief (Figs. 6a & 6b).

The average trend of bands is 340°, with two sets of poles spanning finite arcs dipping steeply NE and SW (Fig. 3j) (average

dip angle is 75°). Band orientation is ca. 20° to the $\sigma_1$ value reported by Chanier et al. (1999). However, angles ranging from

10° to 40° are present. This is indicative of CSBs with large shear components. Data is compared to the $\sigma_1$ value reported by Chanier et al. (1999) for the Wairarapa because it is calculated from analysis of a larger dataset. The bands most commonly occur as single bands; however, clusters restricted to sandstone beds are also observed (Figs. 6a & 6b). Clusters localise into single strands if they propagate into adjacent mudstone beds (Fig. 6b). Band thickness ranges from 0.1 cm to 0.35 cm (Fig. 6), with an average thickness of 0.16 cm. Thicker bands are common in sandier layers and exhibit positive relief (Fig. 6a). Thickness variations are present in the bands as they propagate through different lithologies (Fig. 6b). The length of bands is typically on the order of tens of metres. Many D2 bands extend beyond the outcrop scale, making accurate estimates of length impossible. Bands hosted in areas dominated by sandstone beds, with only very thin mudstone intervals, extend beyond the outcrop scale through the mudstone. In areas characterised by thicker mudstone intervals, bands are restricted to sandstone intervals. Displacement associated with the bands generally ranges from 0 cm to 4 cm with the mean displacement ca. 1 cm. Bands without apparent offset are rare and may be an artefact of a sectioning effect (Soliva et al., 2013). Pattern analysis of the band orientation distribution shows that bimodality can be rejected with a *p-value* of 0.01 (Healy and Jupp, 2018). However, when band orientations are analysed from individual outcrops, bimodality is present. The dihedral angle of bands from individual outcrops ranges from ca. 30° to 60° indicative of CSBs. While bands can accommodate several centimetres of offset, microstructural analysis confirms that shear displacement occurs with compactional strain. Therefore, while the range of dihedral angles and the angles to $\sigma_1$ may indicate shear bands, porosity and grain size reduction within the bands confirms that they are CSBs, often with large shear components.

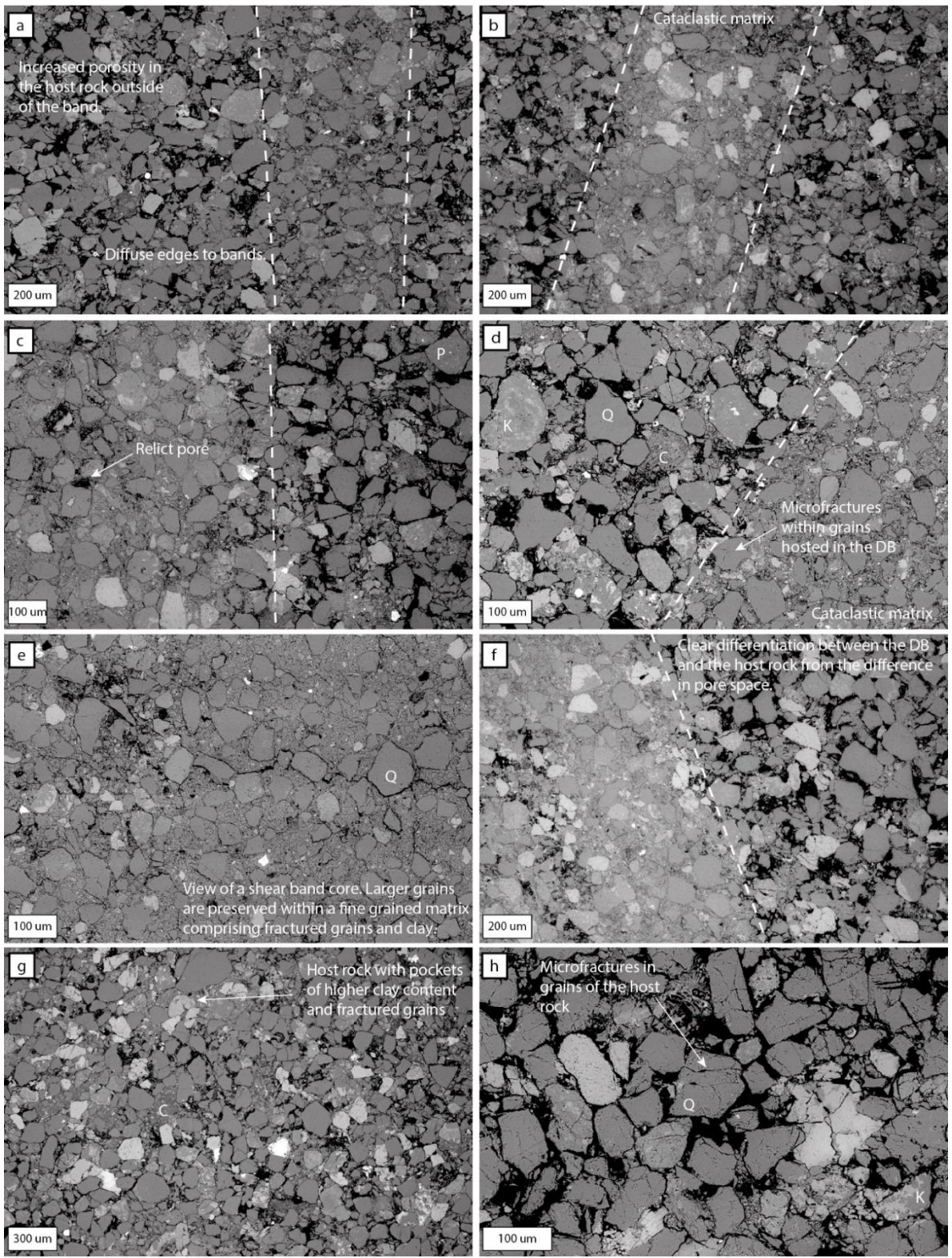

**Figure 8. Microscopic images of D2 normal-sense shear bands and host rock. Dashed lines indicate the edge of deformation bands. [a], [b], [c], [d], [e], [f] images of deformation bands. The bands have diffuse edges and are characterised by reduced porosity and increased clay content. [g] and [h] are images of host rock surrounding deformation bands. Within the host rock microfractures are present. In addition, pockets of smaller grain size can be seen with higher clay content indicating the lithological heterogeneity of the unit. Q = quartz. C = zones of clay-sized particles. K = potassium feldspar. P = plagioclase.**

At the microscale, D2 bands are characterised by a reduction in grain size and porosity compared to the host rock which makes the bands easy to identify under the microscope (Figs. 8a, 8b, 8c, 8d, 8e, 8f, 10). The bands show diffuse borders with the surrounding host rock (Fig. 8c). The grain size distribution of deformation bands generally shows positively skewed distributions with a lower median value compared to the host rock indicating a reduction in grain size due to cataclasis (Figs. 9ai & 9aii) (Fossen et al., 2007; Balsamo et al., 2010). The median grain equivalent circular diameter in deformation bands is 37 µm compared with 57 µm in the host rock, showing a 35% reduction (Figs. 9ai & 9aii). Deformation bands also show a smaller range in grain size at 156 µm ranging from 9 µm to 165 µm compared to 231 µm in the host rock ranging from 8 µm to 239 µm, with the host rock preserving larger grains (see Supplement Section S.8). Deformation bands, compared to the host rock, are characterised by a high matrix content, due to grain size reduction, and a concentration of clay sized grains permitting the distinction between the two (Figs. 8a, 8b, 8c, 8d, 8e, 8f). The amount of matrix decreases from the centre to the outside of the bands and becomes almost non-existent in the host rock, which is dominated by intact grains, with/without intragranular fractures, zones of increased clay content, and pore space. There is an average absolute porosity reduction of 8% (from ca. 13% in the host rock to ca. 5%) in deformation bands (Fig. 10). This equates to 59% relative porosity reduction. Relict medium-sized pores (30-50 µm) are present within some bands, accounting for much of the remaining ca. 5% porosity (Fig. 8c). By using porosity reduction as a proxy for inelastic volumetric strain, we obtain a ratio of $D_S/D_C$ that ranges from ca. 20->100 indicative of CSBs with large shear offset (Soliva et al., 2013; Ballas et al., 2014; Soliva et al., 2016; Fossen et al., 2018). Generally, the darker coloured bands containing higher concentrations of clay have higher $D_S/D_C$ ratios as indicated by shear displacement observed at outcrop scale (Figs. 6c-g). Grain fracture is observed at grain contacts and within grains in both host rock and in deformation bands (e.g., Figs. 8d & 8h). The presence of microfractures in the host rock shows that the deformation is not solely concentrated within deformation bands. Due to a reduction in grain size, a significant reduction in porosity and a low density of microfractures, D2 deformation bands are classified as *cataclastic* CSBs (Antonellini et al., 1994; Mair et al., 2000; Fossen et al., 2007; Ballas et al., 2015).

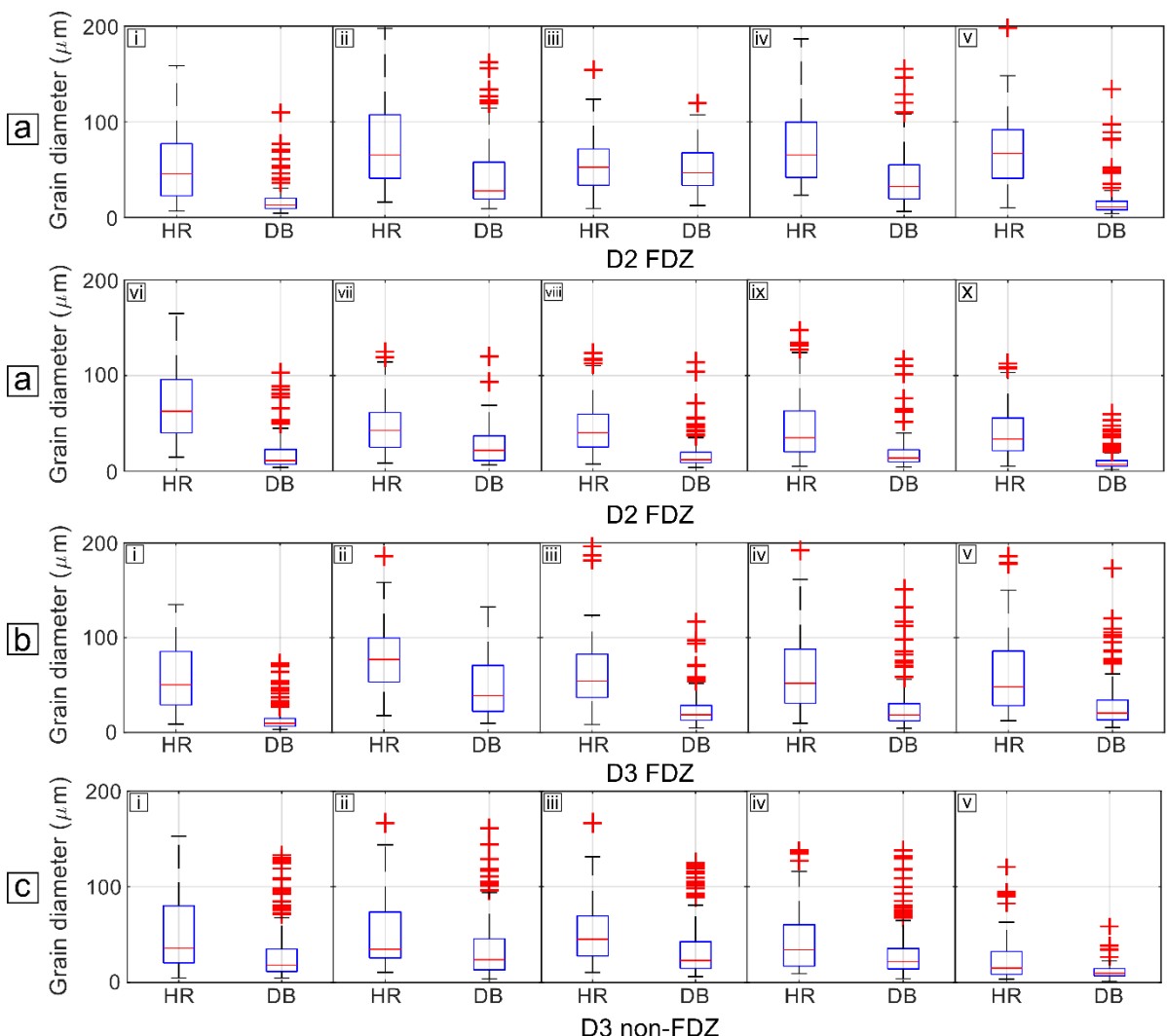

**Figure 9. Box and whisker plots to show the difference in grain size distribution in deformation bands compared to host rock. [a] Grain size distributions associated with deformation bands located in D2 fault damage zones. [b] Grain size distributions associated with deformation bands located in D3 fault damage zones. [c] Grain size distributions associated with D3 non-FDZ deformation bands. For each sample, the interquartile range and the median are smaller in the deformation bands compared to the host rock. The tops and bottoms of each box are the 25th (Q1) and 75th (Q3) samples, respectively. The centre line represents the median. The whiskers are drawn from the ends of the interquartile ranges to the adjacent values (Q1 - 1.5 * interquartile range and Q3 + 1.5 * interquartile range). Data that lies beyond the whisker length are outliers (red crosses).**

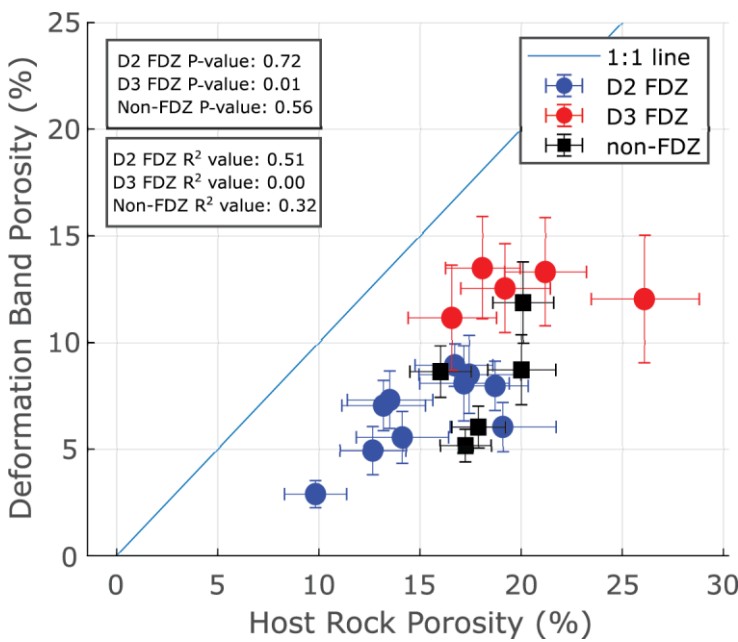

**Figure 10. a) Measured host rock porosity % plotted against porosity measured in the deformation band (%). Each data point represents a thin section where porosity was measured in host rock and deformation band. All deformation bands exhibit a reduction in porosity compared to the host rock. P-value = Pearson correlation coefficient value. $R^2$ value: R-squared, representing the variance from a best-fit. Raw statistical data are shown in Supplement Section S.7.**

### 4.4. D3 Horizontal Contraction

#### 4.4.1. Structures and Relative Timing

Field overprinting criteria show that a contractional event followed D2 horizontal extension (Figs. 5a, 5c, 5d, 5e) (Chanier et al., 1999). Macro-scale D3 overprinting relations include: folding-induced dissection of D2 faults at bedding interfaces through flexural shear of and slip along weak mudstone layers (Fig. 5a); multiple generations of slickenfiber veins in D2 faults where strike-slip orientations overprint dip-slip ones; cross-cutting of D2 faults by D3 thrusts – easily observed along the coast (also observed at the deformation band scale) (Figs. 5c-e); and rotation of D2 faults around F3 fold hinges. These relations have also been described by Chanier et al. (1999) for the North Island east coast.

The structures associated with D3 are: [1] folds; [2] reverse faults; [3] reverse-sense deformation bands located in the damage zones of faults (Fig. 13); and [4] deformation bands with constant spacing and minimal to no offset (Fig. 11). Field overprinting criteria cannot be used to discern the relative timing of these structures during this contractional phase as all structures are not seen interacting in a single outcrop. The structures have a similar orientation to D1 structures (Fig. 3).

#### 4.4.2. Faults and Folds

Upright, NE-SW trending folds with 100 m-scale wavelengths and along-strike extents of several km constitute the most obvious large-scale D3 structures (Fig. 2). Fold axes are shallowly dipping (ca. 10°) and doubly plunging NE and SW (Fig.

3h). Unlike many F1 folds, F3 folds are asymmetrical, with steeper eastern limbs and non-cylindrical geometry (Fig. 2). Synclines commonly have open hinges and resemble box folds in their geometry. Anticlines resemble kink folds with closed hinges and long straight limbs. Higher-order parasitic folds with matching geometry are observed throughout the mapping area and best exposed in road cuts in the hinterland (Fig. 3h). The difference in the geometry of F1 and F3 folds indicates that there were two folding events. However, due to the similar orientations of structures accommodating D1 and D3 deformation, it is not possible to state whether F3 folds represent tightened F1 folds or if they have nucleated solely during D3 horizontal contraction. Within fold limbs, layer-parallel slip and shear can be observed in interbedded mudstone layers, along with thinning-out of these layers (Figs. 5a & 5b). Slip surfaces with slickenfiber veins exhibit opposing shear sense on opposing fold limbs. These observations could be associated with both bending and buckling folding mechanisms (Donath and Parker, 1964; Chapple and Spang, 1974). Given the fact that F3 axial planes trend parallel to the traces of the dominant thrust faults in the area, it is reasonable to assume, by assessing the poles to the axial planes, that F3 folds formed by SE shortening (Grujic and Mancktelow, 1995). This is consistent with the shortening direction indicated by regional thrust faults (Chanier et al., 1999).

D3 reverse faults can be observed along the coastline (Figs. 2 & 13). Reverse faults are less commonly observed compared to D2 normal faults. Poor hinterland exposure prevents their observation away from the coastline. Fault length is often indiscernible as they extend beyond cliff height (>10 m). However, some are shorter than the cliff face (<10 m in length). Fault displacement ranges from 0.2 m to 5 m and is ca. 1 m on average (Fig. 13). Reverse faults trend NNE-SSW with an average dip of 48° (Fig. 3g). The orientation of reverse faults is sub-parallel to the coastline. This generates observational bias when assessing the pervasiveness of reverse faults. A dominant set dips ESE, and a secondary set dips WNW. Statistical analysis shows that the fault orientations are not distributed bimodally (Healy and Jupp, 2018). D3 reverse faults are brittle structures with cataclastic fault cores and occasional gouge. Paleostress analysis of back-tilted D3 reverse faults with slickenfiber veins and shear sense is consistent with the literature and is in alignment with the present-day movement of the Pacific plate ($\sigma_1$: 01/092°) (Fig. 3g) (Chanier et al., 1999; Bailleul et al., 2013). We have analysed back-tilted D3 faults because deformation bands in the damage zone of these faults mutually crosscut conjugate D3 bands with no apparent offset, and orientation analysis of the latter shows that they have been passively rotated during folding (see Supplement Section S.4). However, we cannot rule out that faulting could have occurred anytime during D3, especially in steep fold limbs when deformation can no longer be accommodated by folding. We do not have evidence for this as most of the fieldwork took place at coastal outcrops that are in the gentle to moderately dipping back-limb of a syncline.

### 4.4.3. Deformation Bands

When comparing dihedral angle and $D_S/D_C$ ratio, two types of deformation bands are associated with D3: reverse-sense CSBs and SECBs. SECBs are the most common.

### 4.4.3.1.    D3 DBs not associated with fault zones

SECBs are mainly hosted in areas dominated by sandstone beds, with less interbedded mudstone. SECBs primarily form single bands (Fig. 11e) but also occur in narrow clusters in areas with higher band densities (e.g., Fig. 11d). On average, single bands are 5 mm in width with a range from 0.26 mm – 1 cm. Clusters are on average 2 cm wide. Eye and ramp structures can be recognised in the networks. The bands are lighter in colour than the host rock and show positive relief (Fig. 11). SECBs exhibit no observable shear displacement, except where clusters are present and millimetre offset can be observed. Rare CSBs are present within zones of SECBs. The CSBs are identified by the associated millimetre-scale macroscopic offset of pre-existing structures and sedimentary layering. Some bands pass through multiple beds and can extend beyond the scale of exposure. Most commonly, however, SECBs are confined to individual sand- and siltstone beds where they form conjugate bands that strike NNE and have opposite dips (Figs. 3l & 11). When comparing back-tilted orientation data with present orientations, analysis from 11 outcrops indicates that there is a tighter clustering of band orientations after back-tilting, suggesting that SECBs formed before fold tightening or initiation. In addition, bands orientations mapped on the same bed across a fold hinge lie on the same great circle suggesting that these bands predate F3 folding (see Supplement Section S.4). Pattern analysis shows that most SECBs have a bimodal orientation distribution and can be considered as conjugate sets (Healy and Jupp, 2018). The average dihedral angle is 82° (after back-tilting), ranging from 68-89°.

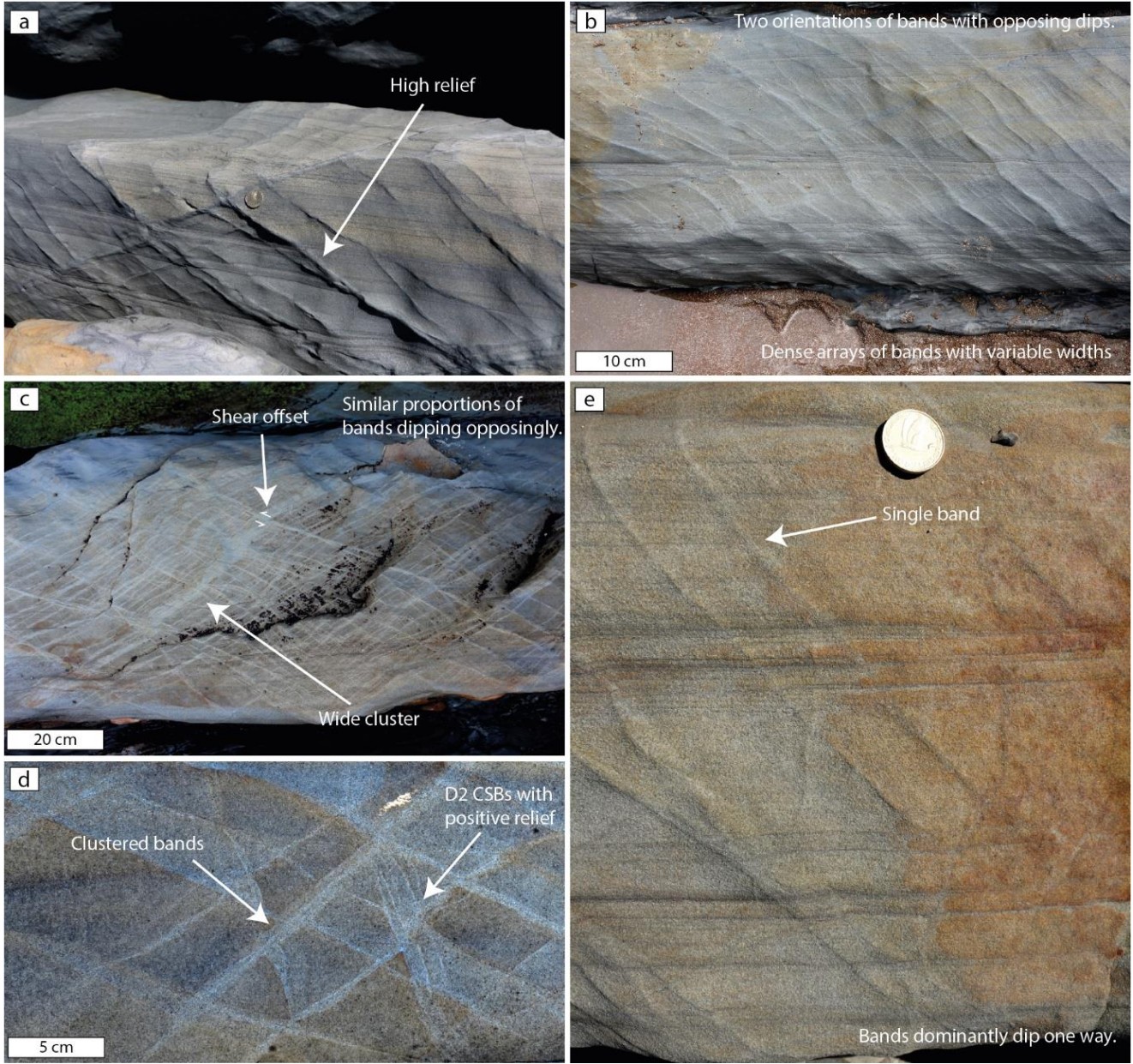

**Figure 11. Field images of regularly spaced SEBCs and CSBs. CSBs are differentiated from SECBs by their mm-cm scale offset. Regularly spaced bands are primarily hosted within single sandstone layers and do not propagate into adjacent layers. [a] Two orientations of D3 SECBs with high relief that exhibit undulations along strike. [b] Two orientations of D3 SECBs. The D3 SECBs in this image are dense and have variable thickness. In between thicker SECBs and clusters are narrow SECBs with moderate relief.**
**[c] Densely spaced SECBs. Similar proportions of SECBs with each dip are observed. Through the centre of the image is a wide cluster of SECBs. Cutting the cluster are CSBs with reverse-sense offset. [d] Wide clusters of SECBs. The SECBs cut high relief D2 CSBs. [e] SECBs that dominantly dip one way and are observed as single bands. Macroscopic offset of sedimentary layers is not present.**

Microscopically, SECBs have diffuse borders with the surrounding host rock (Figs. 12b-d). The bands are characterised by a reduction in grain size and porosity (Fig. 7). Host rock median grain size is 39 µm and deformation band median grain size is 22 µm, representing a 44% reduction (Fig. 8). The range in grain diameters inside and outside of the bands is similar: 157 µm for deformation bands (ranging from 5-162 µm) and 162 µm for host rock (ranging from 4-166 µm). However, the interquartile range is much less for deformation bands: 49 µm for host rock and 27 µm for deformation bands. This small range for deformation bands results in many outliers indicating relict host-rock grains within the bands (Fig. 9c). Grain size within the bands has a unimodal, positively skewed distribution, indicating cataclasis with a larger number of smaller grains. Positive skew suggests that not all grains have been equally fractured. SECBs are only observed in rocks with >15% host rock porosity (Fig. 10). The porosity within the deformation bands, on average, reduced from 18.5% in the host rock to 8.8%, a relative reduction of 52% (Fig. 10). Microfractures are observed in host rock grains and DB grains (Fig. 12). Within the bands, there is a cataclastic fine-grained matrix comprising clay-sized particles of fragmented quartz, feldspar, and lithic grains, which is generally absent in the host rock (e.g., Figs. 12a & 12d). Analysis of compaction versus shear shows an average $D_S/D_C$ value of 0.83 for SECBs, ranging from 0.2–1.5. This value shows that compaction and shear magnitudes are similar, and the bands are therefore microscopically classified as cataclastic SECBs (Ballas et al., 2015).

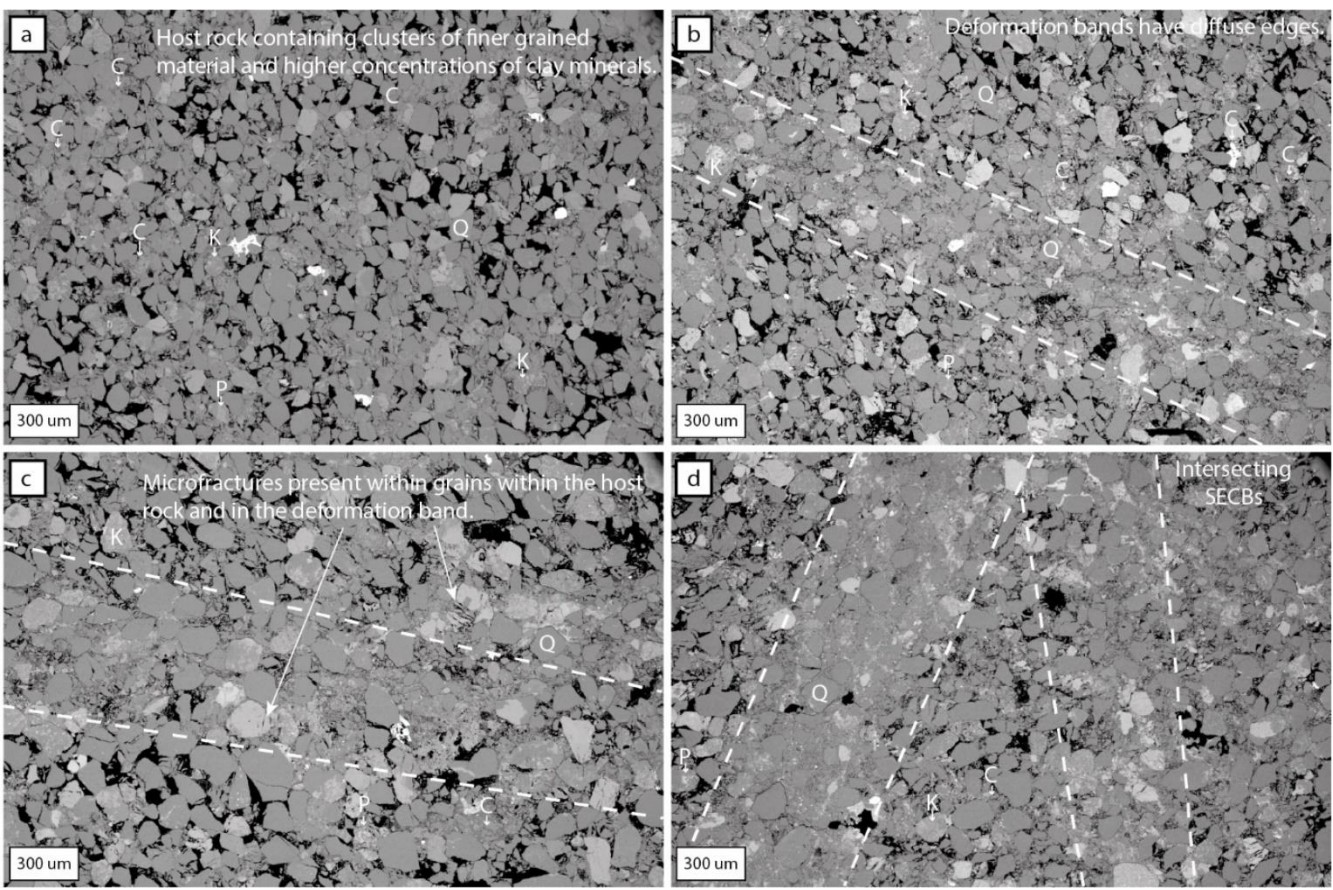

**Figure 12. Microscale images of D3 SECBs characterised by minimal shear offset and periodic spacing, and are distal to fault planes. The bands have diffuse edges and are characterised by a reduction in porosity and an increase in clay content compared to the surrounding host rock. Q = quartz. C = zones of clay-sized particles. K = potassium feldspar. P = plagioclase. [a] Image of host rock located within 5 cm of an SECB. Within the host rock, clusters of clay-sized particles and fragmented host rock grains are present. [b] Deformation band propagating through the host rock. Within the band, the pore space is significantly reduced although porosity is still present, especially closer to the edge of the band. This supports the notion of diffuse boundaries to bands. [c] Within the deformation band, host rock grains are preserved with little to no reduction in grain size. These grains are surrounded by fragments of host rock grains that are generally clay-sized. Within the grains inside the band, microfractures are present. Microfractures can also be observed in the host rock grains indicating that deformation was not isolated to the deformation band. [d] Two intersecting SECBs are present. Within both bands, there is a clear reduction in pore space. The pore space has been filled with fragments of host rock grains resulting in a fine-grained matrix indicative of cataclastic processes.**

### 4.4.3.2.    D3 DBs within fault damage zones

D3 reverse-sense CSBs are dominantly observed within the damage zone of reverse faults. However, SECBs are also present within fault damage zones. CSBs are very similar in macroscopic appearance to SECBs, often with high relief and a lighter colour compared to the host rock (Fig. 13a, 13c, 13d). CSBs, unlike SECBs, are consistently characterised by macroscopically visible shear offset. Fault-related CSBs resemble D2 CSBs when bands propagate though muddy layers and clay smear is present (Fig. 13b). They mainly occur as single bands but also in clusters (Fig. 13). Single bands are on average 5 mm wide and range from 3-7.5 mm. Clusters are ca. 2-3 cm in width (e.g., Fig. 13c). CSBs accommodate mm-cm scale reverse offset (e.g., Fig. 13a). In clay-rich host rocks, the bands can accommodate several centimetres of offset, indicating that some bands are CSBs with large shear offset (Fig. 13b). The bands trend NNE with one poorly clustered dominant set dipping ESE (Fig. 3k). The dominant set dips ESE at 63° and the less dominant set dips WNW at 29°. This orientation is very similar to D3 faults indicating that they formed in the same stress field. SECB orientation does not match with fault orientation as strongly (Figs. 3g & 3l). The bands generally propagate for multiple meters and extend out of outcrop observation, similar to D2 normal-sense deformation bands. However, some terminate against thicker mudstone intervals. Pattern analysis of all CSBs measured across the field site rejects bimodality in CSBs associated with faults, suggesting a polymodal orientation distribution (Healy and Jupp, 2018). However, when analysed at individual outcrops, pattern analysis suggests that the bands are bimodal with dihedral angles ranging from 51-80°.

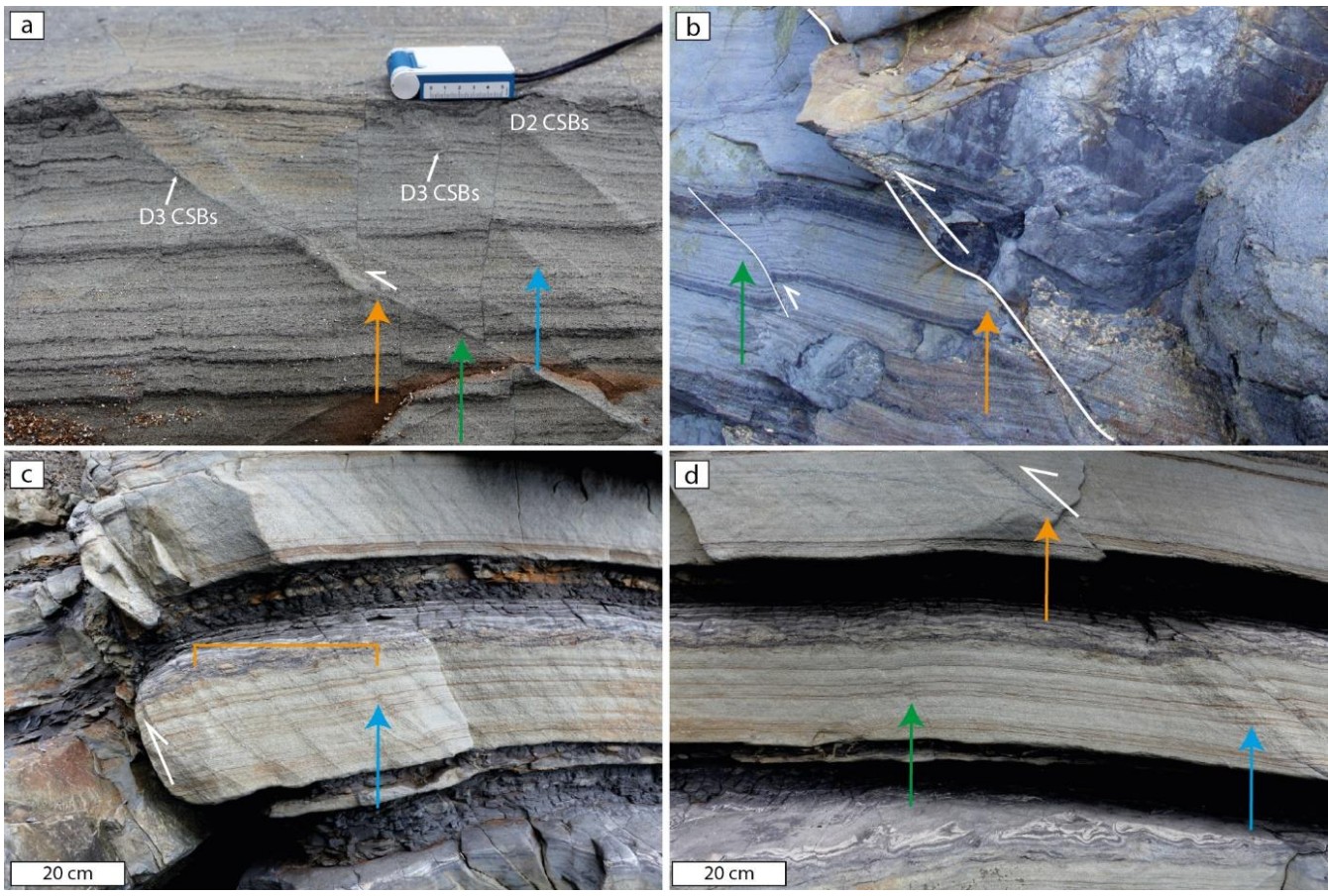

**Figure 13. Images of reverse-sense deformation bands and D3 reverse faults. [a] Reverse-sense compactional shear bands are observed. When propagating through clay-rich layers, the bands accommodate several centimetres of offset (orange arrow). Bands are characterised by low relief when darker than the host rock in colour (green arrow). When characterised by higher relief, the shear displacement magnitude reduces (blue arrow). Within the image, D2 CSBs are observed to be offset by D3 CSBs. [b] A D3 reverse fault (orange arrow) with an associated damage zone that comprises compactional shear bands that accommodate reverse offset in orientations that match the main fault plane (green arrow). [c] A D3 reverse fault with associated damage zone. The DBs within the damage zone do not show a clear decay in spacing (orange bracket) suggesting that periodically spaced structures may represent the first stage of deformation and then faults can propagate from the linkage of pre-existing bands. The DBs mainly comprise clusters with reverse-sense shear offset (blue arrow). [d] An image of the beds shown in image [c]. [d] was taken 2 m to the right of image [c]. Here, a slip surface can be seen that is characterised by a darker colour (orange arrow) compared to adjacent deformation bands that have a higher relief and are clustered (blue arrow). The green arrow highlights pre-existing D2 CSBs that have been incorporated into the damage zone of the fault shown in [c]. The incorporation of clay into the slip plane seems to promote shear localisation and failure.**

Microscopically, CSBs are characterised by diffuse edges (Figs. 14a & 14d). These bands show the largest reduction in grain size of all deformation bands observed at the field site. Median host rock grain size is 57 µm compared to 21 µm in the deformation bands, a reduction of 63%. The total range in grain size is similar for the host rock and deformation bands, 228 µm for the host rock (ranging from 8 µm to 236 µm) and 224 µm for the deformation band (ranging from 4 µm to 228 µm). However, the interquartile range is considerably different, 54 µm for host rock and 22 µm for the deformation band (Fig. 9b). This variation is similar to that seen in the D3 SECBs. As with the other bands, D3 CSBs have a lower porosity than the host

rock, 20% outside of the band and 12.3% inside the band – a reduction of 39% (Fig. 10). This is the smallest reduction in porosity observed across the varieties of deformation band. Pore space in deformation bands is filled with fragments of grains and clay minerals (Figs. 14a-e). Grains within the bands and the host rock show intragranular fractures radiating from grain contacts (Fig. 14f). Compaction and shear analysis show that the $D_s/D_c$ value is 63 ranging from 32 to 106. This, as with D2 bands, indicates considerably more shear strain than compaction, and characterises the bands as CSBs often with large shear offset. The grain size reduction, as indicated by the interquartile range, indicates that many are cataclastic CSBs.

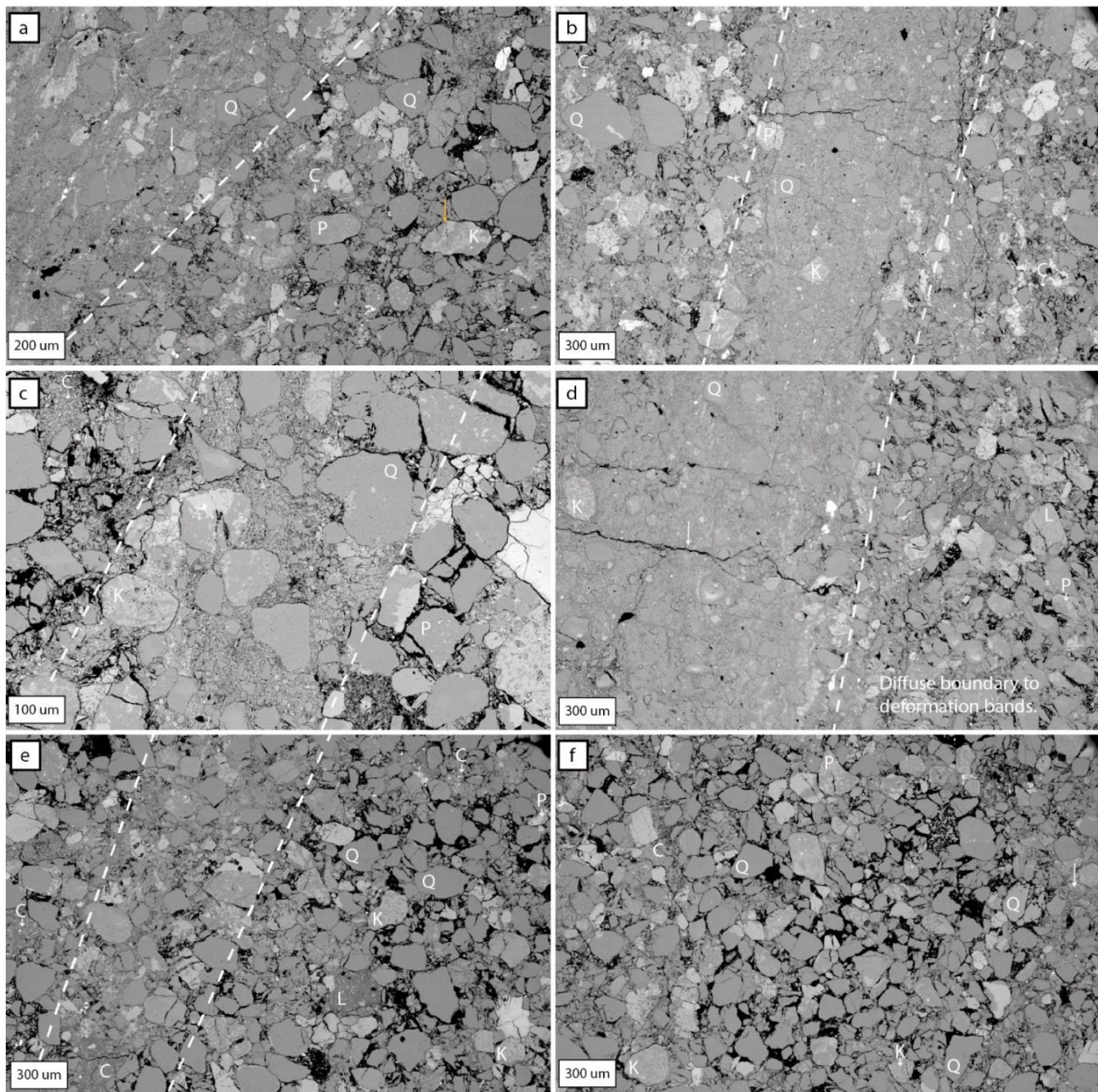

**Figure 14.** [a], [b], [c], [d] BSE images of deformation bands associated with D3 reverse-fault damage zones. The bands are characterised by reduced porosity and grain size. Dashed lines denote the boundaries to DBs. [a] Image of DB and host rock. Within the DB there is limited remaining pore space. The pore space has been filled with fine grained fragments of host rock grains. Microfractures can be observed within the preserved host rock grains within the DB (white arrow). Additionally, microfractures are present in the host rock grains (yellow arrow). [b] The deformation band in this image is characterised by a significant reduction in porosity when compared to the host rock material. Cutting across the deformation band is a fracture. Within the band, some host rock grains are preserved. However, many grains are smaller than in the host rock. [c] The deformation band in this image is

**characterised by a reduction in pore space, however, many host rock grains are preserved. Delineation of this band into the host rock is difficult. This highlights the diffuse nature of the bands. [d] Within this image, the deformation band is clearly observed and is characterised by reduced grain size and pore space. Later-stage fractures cut the deformation band but do not propagate into the host rock (white arrow). [e] Image of host rock surrounding deformation bands. Within the image a zone of reduced porosity is observed but it is not characterised by the same magnitude of porosity reduction as is observed in other bands. This band possibly**

**represents the early stages of formation of a deformation band. [f] Image of host rock. Pore space is clearly observed in the host rock. Within the host rock, regions of higher concentrations of fragmented grains are present (white arrow). This highlights the heterogeneity of the host rock. Q = quartz. C = zones of clay-sized particles. K = potassium feldspar. P = plagioclase. L = lithic.**

### 4.5.    Spacing

### 4.5.1.    Synthetic Spacing

To explain the spacing analysis of natural data, six synthetic images representing different spatial distributions of deformation bands were constructed (Fig. 15). Bands with a strictly constant spacing show a zero Pearson correlation coefficient (Fig. 15a). With the addition of normally distributed 'noise' to case [a], a distributed set of spacing with Gaussian noise is obtained (Fig. 15b). In this case [b], the Pearson correlation coefficient of spacing over distance is -0.08. The third synthetic image [c] represents a damage zone with an exponential decay of deformation band density away from the fault plane (Fig. 15c). For

this example, the Pearson correlation coefficient between spacing and distance to the fault is exactly 1 (Fig. 15c). A histogram of this spacing distribution shows positive skew. The same results are expected for any other analytical expression in which spacing grows monotonously with distance from the fault (e.g., the power-law relationship established by Savage and Brodsky (2011)). With the addition of Gaussian noise to a damage zone [d], the Pearson correlation coefficient reduces to 0.56 (Fig. 15d). However, its exact value depends on the amount of noise and can be larger or smaller. Synthetic image [e] represents an

overprint of two distributions, equal spacing and variable spacing, simulating two different deformation events affecting the same bed subsequently (Fig. 15e). If the deformation bands resulting from the two events are morphologically similar, they may be difficult to distinguish in the field and can be mapped together as one. In this case, the Pearson value is 0.71, although it could be any value between 0 and 1 dependent upon the value of spacing in the background density compared with the damage zone spacing. In image [f], noise is integrated into both distributions and they are combined (Fig. 15f). This is the

most realistic outcome if there is an overprint of two events with different distributions. In the case shown, the Pearson value is 0.59. However, with different magnitudes of noise and varied initial spacings, any value from -1 to 1 could be obtained. For the interpretation of our field data, we assume a positive correlation between band spacing and spatial location if the Pearson value is > 0.5.

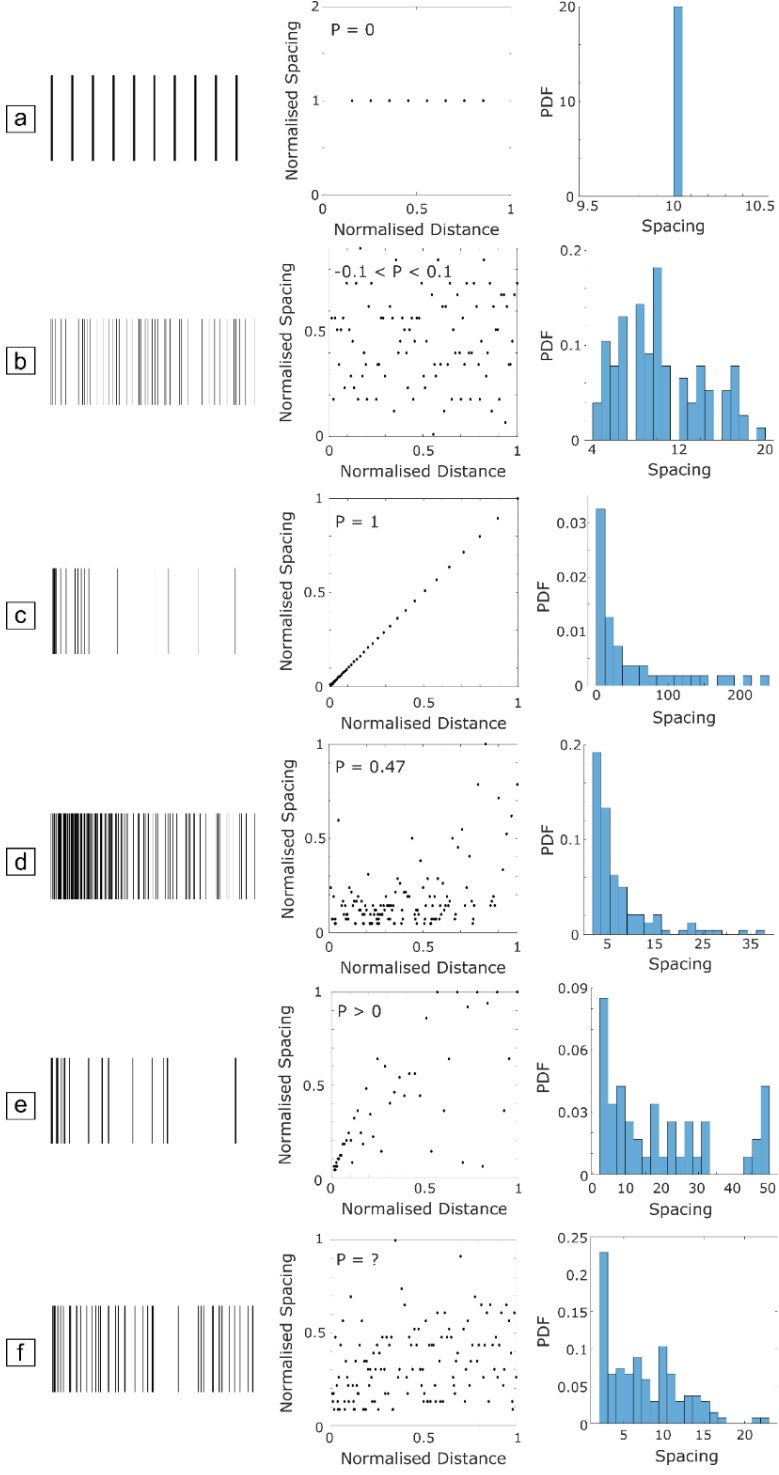

 **Figure 15. Synthetic images produced to replicate different distributions of the deformation bands. [a] deformation bands with a constant spacing, [b] deformation bands with a constant spacing with added Gaussian noise, [c] bands with an exponential decay away from a fault plane, and [d] deformation bands with an exponential decay away from a fault plane with added Gaussian noise. [e] and [f] represent images that combine constant spacing with a damage zone. These examples would be relevant if a damage zone overprinted a constant background spacing of the deformation band. [e] combines [a] with [c]. [f] combines [b] and [d] resulting in**
**a spacing that contains two distributions that have noise. The relationship between spacing and distance is analysed using the Pearson correlation coefficient. Values close to 1 show a positive relationship between the distance from a point and the spacing. This would be seen in a damage zone that shows an increase in spacing between deformation bands as the distance from the fault plane increases. Pearson correlation coefficient values close to 0 show no correlation between spacing and distance. A histogram of the spacing is also shown. Unimodal distributions with no skew reflect bands with near-constant spacing. A positive skew represents**
**damage zones. Distance and spacing are also plotted for different scan-lines. Distance is normalised to the maximum distance and spacing is normalised to the maximum spacing. P = Pearson correlation coefficient value.**

### 4.5.2. Natural Spacing

#### 4.5.2.1. Map Scale

Analysis of the location of the macro-scale faults, with displacements greater than 20 cm, show that many have a similar

spacing (Fig. 16). D2 faults have a positively skewed distribution with a skewness value of 0.3. The median spacing of the 114

D2 faults analysed is 8.7 m when corrected for true spacing. The Pearson correlation coefficient for the spacing against distance

is -0.07, with an analysis conducted from SW-NE. D3 faults have a bimodal distribution indicating two primary spacing

populations with modes of 5 m and 13 m when corrected for true spacing. Skewness for the 41 analysed D3 faults is 0.03. The

Pearson correlation coefficient is 0.44. Pearson values <0.5 indicate that there is no correlation between the spacing and

location of normal or reverse faults.

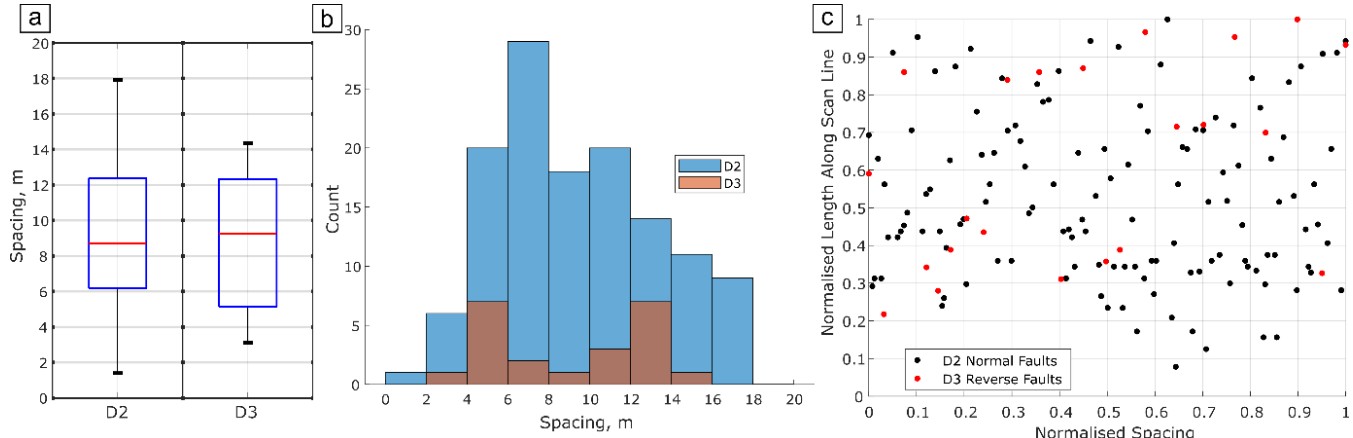

**Figure 16. [a] Boxplots of corrected spacing distribution of D2 normal faults and D3 reverse faults. [b] Histograms of the corrected spacing distribution of D2 normal and D3 reverse faults. A normality test (Lillie test (Mathworks, 2011)) shows that D2 has a normal distribution while normality is rejected for D3. D2 shows a positive skew, and D3 has a bimodal distribution. [c] The plot of the**
**normalised spacing against distance from the start of the scan-line for normal and reverse faults. The plot most closely resembles [b] and [f] from Figure 15.**

#### 4.5.2.2. Deformation Band Spacing

Qualitatively, deformation bands in the mapping area appear to exhibit two spacing distributions. Generally, those associated with faults (both normal- and reverse-sense) form fault damage zones with a variable distribution (Figs. 6, 7, 13) where spacing increases with distance from the fault plane. Damage zone width varies greatly along faults and from fault to fault. The maximum width of the damage zone is identified by a return to a background level of damage. Some damage zones can extend for up to 5 m before interacting with the damage zone of an adjacent fault, while other beds along the same fault show no macroscopic damage zone. The second spatial distribution is a regular spacing. Regular spacing is generally observed when there is no fault nearby (Fig. 11). Regularly spaced bands are also observed in damage zones as a separate pattern to bands with variable spacing (e.g., Fig. 13c). Quantitatively these observations were tested with the hypotheses: bands observed adjacent to a fault plane have a positive correlation between spacing and distance from the fault plane and bands not observed adjacent to a fault plane have no relationship between spacing and distance.

##### 4.5.2.2.1. Fault Damage Zone Deformation Bands

D2 normal and D3 reverse faults are bordered by deformation bands, forming a fault damage zone. Damage zone width varies from 0.1 cm to 272 cm for the analysed faults with associated thin section analysis. Twelve faults with damage zone width >10 cm were analysed for spacing statistics (e.g., Fig. 17). The average Pearson correlation coefficient for each analysed fault ranges from 0.07 to 0.62 with a combined average of 0.4 (see Supplement Section S.9 Fig. S9.5). 71% of D2 and 60% of D3 FDZ spatial distributions have Pearson coefficient distributions with a median <0.5 indicating that there is no correlation between spacing and distance (e.g., Figs. 19b & 19d) (see Supplement Section S.9 Fig. S9.5). Analysis of spacing distributions shows that all faults have positively skewed spacing values (see Supplement Section S.9 Fig. S9.7). This data most closely matches synthetic images (d) and (f) (Fig. 15) indicative of a damage zone or spatial overprints.

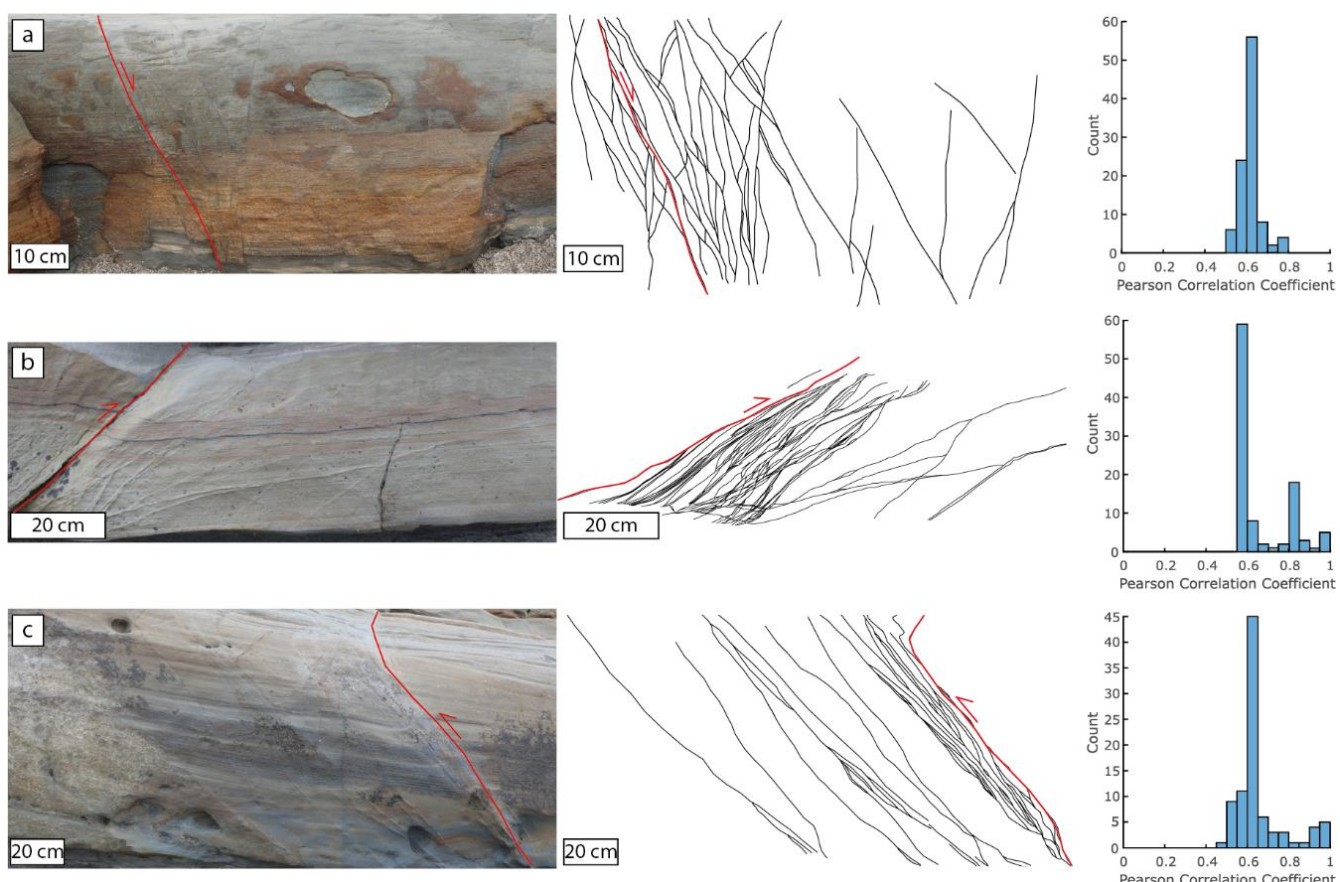

**Figure 17. Spacing analysis of CSBs associated with the damage zone of faults. 100 scan-lines were taken for each image during analysis. [a] represents a D2 normal fault and [b] and [c] show small D3 reverse faults. Each of the three faults show an increase in spacing between deformation bands with increasing distance from the fault plane. This is reflected in the Pearson correlation coefficient. Values are approaching 1, which shows a dependence of distance on spacing.**

### 4.5.2.2.2. Non-FDZ Deformation Bands

We examined D3 SECBs and CSBs with apparently constant spacing from twenty-eight outcrops where damage zones were not clearly detected, thus the damage was not associated with a nearby fault. The spacing generally does not correlate with distance from the first deformation band (Figs. 18 & 19e). Many peak Pearson correlation coefficients are close to 0, average of 0.02, with a normal distribution indicating no relationship between spacing and distance (Fig. 18). However, Pearson values range from -0.65 to 0.61, indicating that there are outcrops that are not associated with a fault that show a distance control on spacing (Fig. 19f) (see Supplement Section S.9 Fig. S9.6). 10% of analyses are complex with strongly skewed spacing frequency distributions and Pearson coefficient values indicative of a distance control on spacing (>|0.5|) (see Supplement Section S.9 Fig. S9.8). 43% of analysed outcrops, however, show variable Pearson values as the scan-lines analyse different parts of the outcrop. Large ranges of values are observed, and in some cases these values are negative, indicative of an

anticorrelation between spacing and distance. This effect can be seen in figure 18c where a scan-line along the top of the image would result in a negative Pearson value, as observed in the Pearson histogram (Fig. 18c). This highlights the importance of using multiple scan-lines (Sanderson and Peacock, 2019). Data for bands not associated with faults correlates with synthetic images [b] and [f], corresponding to constant spacing with 'noise' and overprinted distributions, respectively. Discrete median spacing values range from 1.3 cm to 21.8 cm. Equivalent outcrops with D2 bands were not analysed for spacing statistics as suitable outcrops were proximal to faults and, therefore, did not meet the criteria. Conclusively, spacing statistics have shown that both close to and far from faults, deformation bands associated with horizontal extension and horizontal contraction are characterised by regular spacing (Fig. 19b, 19d, 19e). Additionally, the statistics show that while regular spacing is predicted far from faults, aperiodic spacing can also be observed (Fig. 19f).

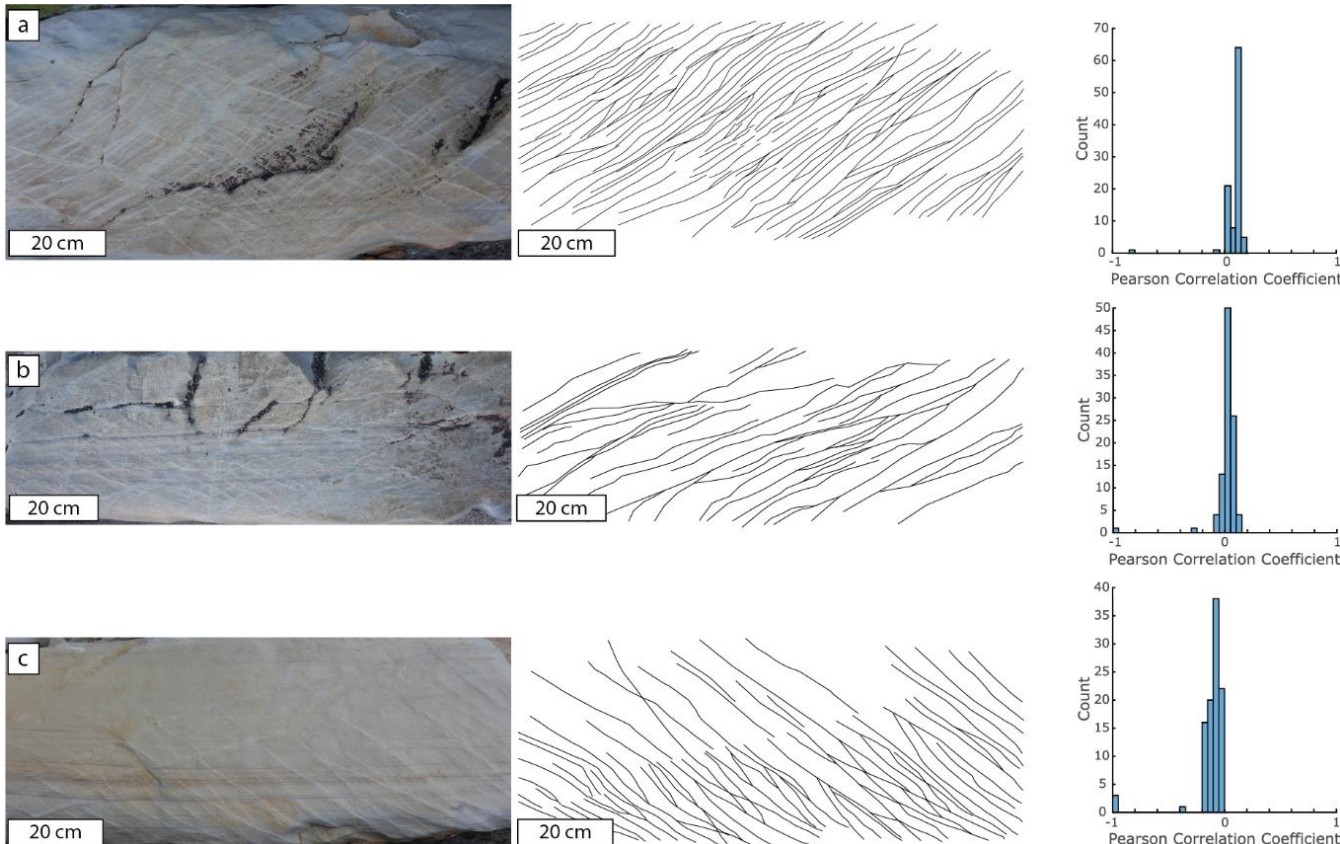

**Figure 18. Spacing analysis of dominantly SECBs with rare CSBs not associated with the damage zone of faults. 100 scan-lines were taken for each image during analysis. Each image shows a general lack of correlation of the distance with the spacing between deformation bands. Pearson correlation coefficient values close to 0 further suggest a lack of correlation. These bands show a near-constant spacing. Median spacing: [a] = 1.3 cm; [b] = 1.8 cm; and [c] = 2.3 cm.**

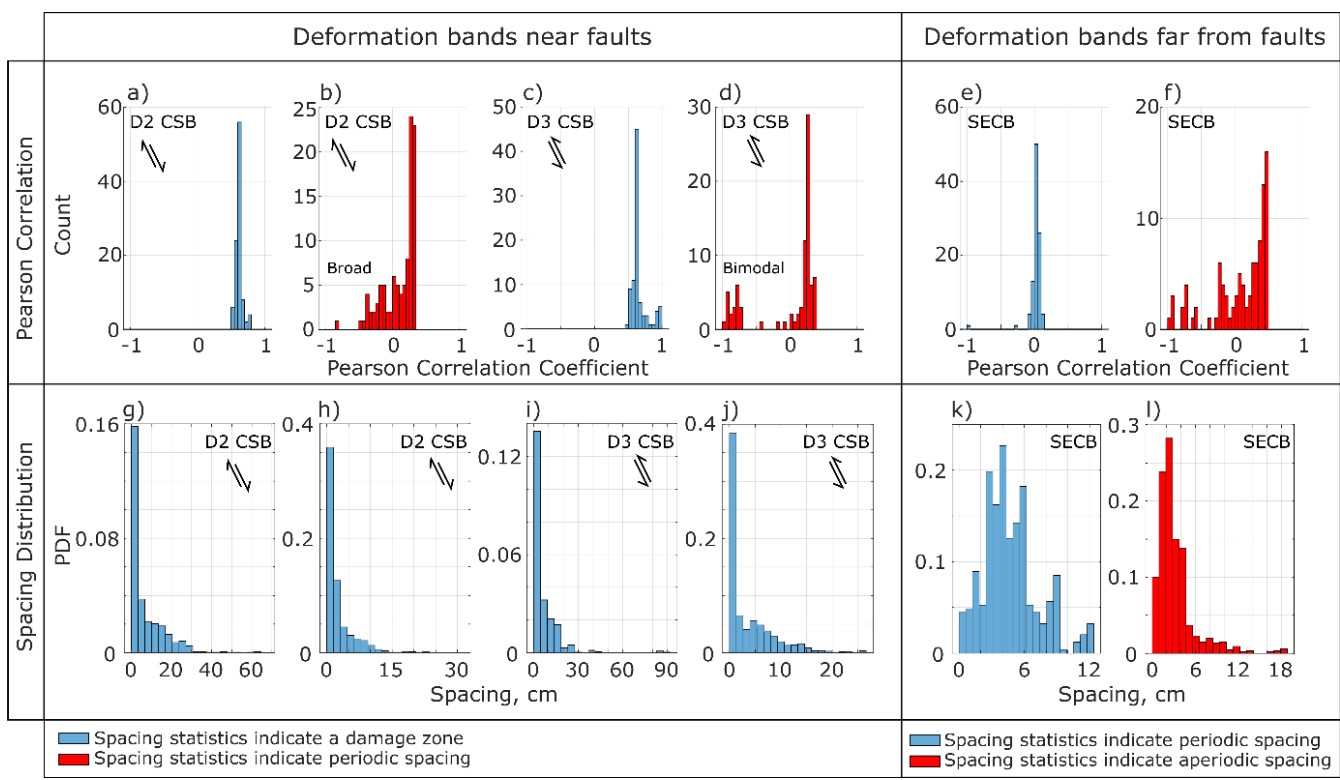

**Figure 19. Spacing statistics for selected outcrops to highlight the variation in deformation band distribution across the field site. The figure has been separated into bands that were analysed adjacent to normal- and reverse-faults, where a decay of spacing away from the fault plane is expected, and those that were analysed away from fault planes where no correlation between spacing and distance is expected. Two examples of each have been chosen to represent possible end-member datasets. a-f show distribution of Pearson correlation values for each analysed scan-line for the outcrops. CSBs are primarily associated with faults and are expected to show a positive correlation (Pearson > 0.5). SECBs, with some conjugate CSB sets, were measured >2 m from faults and should not show a correlation between spacing and distance resulting in a Pearson value close to 0. a, c, and e show examples that match these predictions. b, d, and f are examples that contradict these predictions. g-l shows the spacing frequency distributions for different outcrops. Band spacing associated with faults is expected to show a positive skew. All bands associated with faults show a positive skew. However, variation can be seen with large ranges and a non-smooth curve. Bands not associated with faults are expected to show a Gaussian distribution, representing a 'regular' spacing with noise (Fig. 10b). k shows this while l is characterised by a positive skew, as would be expected for a fault damage zone. Data for all analysed outcrops can be observed in Supplement Section S.9.**

**Table 1 Primary results from the analysis conducted at Castlepoint.**

|  |  | Normal Fault Regime (D2) | Thrust Fault Regime (D3) | |
|---|---|---|---|---|
| **Macroscopic** | Macroscopic structures | Normal faults | Reverse faults | Folds |
|  | Fault pattern | Polymodal | Polymodal | Bimodal |
|  | Mean vector | Not applicable | Not applicable | 87/310 (axial plane) |
|  | Corrected fault spacing (median) | 8.7 m | 9.3 m | Not applicable |
| **Mesoscopic** | Deformation band kinematics | Normal-sense | Reverse-sense | No observable offset/reverse-sense Shear enhanced compaction bands and Compactional shear bands |
|  | Deformation band type | Compactional shear band | Compactional shear band | |
|  | Location | Proximal to faults | Proximal to faults | Not associated with faults |
|  | Dominant deformation band distribution | Aperiodic | Aperiodic | Periodic |
|  | Fault pattern | Polymodal | Polymodal | Bimodal |
|  | Mean vector | Not applicable | Not applicable | 32/110 & 30/280 |
| **Microscopic** | Absolute porosity reduction | 7.90% | 7.70% | 9.70% |
|  | Grainsize reduction | 35% | 63% | 44% |
|  | Cataclastic matrix | Present | Present | Present |
|  | Band borders | Diffuse | Diffuse | Diffuse |
|  | Microscale width | 1.23 mm | 1.53 mm | 0.70 mm |

## 5. Discussion

In the following, we will discuss our observations of deformation bands from Castlepoint, New Zealand and compare them with previous studies to evaluate the control of host rock properties, tectonic regime, and tectonic setting. The primary results

are presented in Table 1. We shall first discuss the association of host rock properties and tectonic regime with deformation band kinematics and the spatial distribution of bands before concluding with remarks on the implications for fluid flow in this deformed rock package hosted within a subduction wedge.

### 5.1. Band Kinematics, Orientation, and Microstructure

Outcrop and microstructural data demonstrate that D2 bands, associated with a normal-fault regime, are Compactional Shear

Bands (CSBs) with normal-sense shear offset. This is consistent with previous studies of deformation bands formed in extensional tectonic regimes (Soliva et al., 2013; Ballas et al., 2014; Fossen et al., 2015; Soliva et al., 2016; Fossen et al., 2018). Associated with the D3 thrust-fault regime are reverse-sense CSBs and Shear-Enhanced Compaction Bands (SECBs), consistent with observations from other contractional regimes (Ballas et al., 2013; Soliva et al., 2013; Ballas et al., 2014; Fossen et al., 2015; Fossen et al., 2018). SECBs are identified by their lack of offset and higher angle to $\sigma_1$. The presence of

CSBs and SECBs in association with D3 horizontal contraction can be explained by variations in finite strain and/or age of the

structures or by intersection of the yield envelope in different sections (Fig. 1). The latter can be caused by variations in the stress state or the yield envelope of the host rock which is intrinsically linked to material properties (Fossen et al., 2007; Soliva et al., 2013). Since the transition between kinematic types is continuous, small changes in the stress state induced by local stress fields or host rock heterogeneity can explain the presence of both band types. Such changes, in addition to variations in finite strain and age of the structures, can explain the shear localisation and subsequent thrust faulting that is observed across the field area. Small-scale reverse faults and associated damage zones are not observed in other studies of deformation bands formed under horizontal contraction (e.g., Ballas et al., 2013; Ballas et al., 2015; Fossen et al., 2015; Soliva et al., 2016). Although, in the Jurassic Aztec sandstone, contractional CSBs have been observed that developed into faults (Fossen et al., 2015). However, associated damage zones are not described, and these bands have no well-defined network organisation (Fossen et al., 2015). In the Whakataki Formation, D3 reverse-fault damage zones comprising reverse-sense CSBs with similar orientations to the fault are observed, with an increasing density near the fault plane (Figs. 17b & 17c). We hypothesise that the DBs represent precursory damage (Antonellini and Aydin, 1994). DBs are associated with a strain hardening processes whereby continued deformation is accommodated through the nucleation of new bands, rather than continued deformation of pre-existing bands. Once strain can no longer be accommodated through nucleation of new bands, perhaps because the rock can no longer accommodate further compactional strain (Soliva et al., 2013), strain is accommodated through the formation of a localised slip zone (Antonellini and Aydin, 1994; Shipton and Cowie, 2001, 2003; Schueller et al., 2013). This results in the preservation of deformation bands surrounding or at the edge of slip surfaces to produce a precursory damage zone (Schultz, 2019). The similarity between orientations of DBs hosted with damage zones and the associated faults is observed for D2 normal faults and D3 reverse faults (Figs. 3f & 3f, Figs. 3g & 3k), indicating that DBs are common precursors to faults in both tectonic settings. However, mature D2 and D3 faults also exhibit other types of damage zones (Figs. 6c, 6d, 6e, 6f) such as interaction, approaching, and tip damage zones (Kim et al., 2004; Peacock et al., 2017).

Pattern analysis of orientation data at the map-scale reveals two distributions: [1] polymodal data associated with faults and CSBs and [2] bimodal data associated with SECBs and CSBs not associated with fault damage zones. Although, at the outcrop scale, deformation bands associated with faults are commonly characterised by bimodal patterns. Polymodal patterns are consistent with 3D strain, while bimodal patterns are consistent with the 2D plane strain (Fossen et al., 2018; Healy and Jupp, 2018; Cai, 2019). 3D strain is most commonly expected in nature (Healy et al., 2015; Healy and Jupp, 2018). We propose that the presence of a bimodal distribution in our data is a consequence of the measurement scale. Orientations of faults and associated CSBs were measured over a large area at the kilometre scale. At this scale, deformation is 3D in the research area, as seen qualitatively by the variability in the large-scale fold and fault geometry (Fig. 2). In contrast, conjugate SECBs and CSBs were documented in a much smaller area on the scale of hundreds of meters because of outcrop conditions. Furthermore, CSBs located in individual damage zones are measured on a decimetre scale. At this scale, structures can be fairly cylindrical (Fig. 2), and there is a higher chance to sample at a plane-strain scale. At the km-scale, there are many physical reasons for

expecting 3D strain in D2 and D3 structures of the Whakataki Formation: [1] the layers have already been folded and faulted in D1 horizontal contraction; [2] the Whakataki Formation has along-strike and stratigraphic thickness variations; [3] rough seafloor topography below the wedge which is predicted during D2 in the southern Hikurangi margin, combined with oblique subduction can induce strong stress/strain heterogeneity (Jones et al., 2005; Wang and Bilek, 2014); [4] the stress field evolved gradually with the principal stress orientations rotating in the transition from D2 (vertical $\sigma_1$) to D3 (horizontal $\sigma_1$); [5] faults locally perturb the stress field and generate heterogeneous orientations (Maerten et al., 2016); and [6] the material properties in different layers can be anisotropic. All these effects should result in spatial and temporal differences in stress state on a variety of length scales. Therefore, one can expect the contemporaneous formation of bands with quite different kinematics, possibly explaining the presence of CSBs with a range of shear offsets in normal-faulting regimes, CSBs and SECBs in thrust-faulting regimes, and the complex patterns observed in outcrop.

While pattern analysis indicates that at a smaller scale, plane-strain is sampled, analysis of orientation data reveals both symmetrical (conjugate sets) and asymmetrical (single orientation families dominate) patterns. SECBs mainly show a symmetrical orientation pattern with two sets characterised by opposing dips (Fig. 11d). Comparatively, D2 CSBs and D3 CSBs, which are often observed in the damage zones of normal and reverse faults, respectively, are commonly asymmetrically (or unimodally) oriented (Fig. 17). Within the damage zones of faults, the bands tend to share a similar dip to the main fault plane (Fig. 17). Whether the pattern is symmetric (bands of opposing dip directions are distributed homogeneously) or asymmetric (one band orientation dominates and/or bands cluster) in the thrust-fault regime depends on the elastic properties of the layers and the friction coefficient between layers (Chemenda et al., 2014). Generally, in association with damage zones, bands have an asymmetric pattern. In contrast, regularly spaced bands are dominantly symmetric. However, asymmetric patterns of regularly spaced bands are observed where one orientation set dominates (Fig. 11e). Therefore, the lithological and geometric heterogeneity in the Whakataki Formation results in variable material properties which in some places may favour symmetry, and in others asymmetry due to differences in the ratios of elastic stiffness between adjacent layers (Chemenda et al., 2014). In addition, as a bed experiences progressive deformation, material properties change, and transitions from asymmetry to symmetry can also occur (Saillet and Wibberley, 2010; Klimczak et al., 2011; Chemenda et al., 2014). Layers which promote asymmetry are associated with faulting. Therefore, the variations in material properties between layers appears to influence the locations of faults within the sequence.

The material properties of an outcrop are influenced by the presence of deformation bands and can be documented through microstructural analysis. Microstructurally, all bands are characterised by a reduction in porosity and grain size, which is consistent with literature describing cataclastic bands (Solum et al., 2010; Ballas et al., 2013; Soliva et al., 2013; Ballas et al., 2014; Fossen et al., 2015; Fossen et al., 2018). While some larger grains are preserved within bands, grain size reduction is statistically significant (see Supplement Section S8) and is indicative of cataclastic processes (Ballas et al., 2015). This is consistent with the reduction in porosity observed in the bands with smaller fragments filling in available pore space (Fig. 10).

Cataclastic processes are expected at burial depths >1 km (Fossen et al., 2018). The Whakataki Formation is predicted to be close to its maximum burial depth of ca. 3 km during D2 and to have reached it by the end of D2 (Wells, 1989; Lee et al., 2002). Although cataclastic processes are generally associated with bands exhibiting positive relief and a lighter colour compared to the host rock, many D2 CSBs and some D3 CSBs exhibit negative relief. Within the bands a fine-grained matrix is observed consisting of clay sized particles which give the band the macroscopic appearance of disaggregation bands that

form at shallower burial depths, but are not associated with grain size reduction (Kristensen et al., 2013). Disaggregation bands within the Whakataki Formation are ruled out by the consistent presence of a cataclastic matrix. The absolute average porosity reduction of ca. 10% observed in all deformation bands from the Whakataki Formation aligns with previous studies of cataclastic bands (Fig. 10) (Ballas et al., 2013; Soliva et al., 2013; Ballas et al., 2014; Ballas et al., 2015; Fossen et al., 2015; Soliva et al., 2016; Fossen et al., 2018). Permeability was estimated from porosity values using an empirical relationship (Wu,

2004) (see Supplement Section S.6). A comparison of host rock and deformation band reveals a reduction of ca. two orders of magnitude for the normal- and thrust-fault regimes. These values are within the bounds observed in previous studies (Ballas et al., 2015; Fossen et al., 2018). Such reductions can impact fluid flow through the unit.

### 5.2.    Deformation band spatial distribution

In the Whakataki Formation, deformation bands with a regular spacing, and deformation bands localised into zones and clusters

are observed in normal- and thrust-faulting tectonic regimes. The observation of regularly spaced bands associated with a normal-fault regime (Fig. 19b) contrasts with previous studies that identified a tendency of localisation rather than pervasively distributed deformation bands (Soliva et al., 2013; Ballas et al., 2014; Fossen et al., 2018). Distributed bands have only been observed in extensional settings when associated with; [1] large relay ramps, [2] when deformed layers are above soft layers such as shale or salt, and [3] when forming in large rollover structures (Fossen et al., 2018). The Whakataki Formation most

closely resembles case 2, with deformed strong sandstone beds embedded in soft mudstone layers. However, in the Whakataki Formation, elastic stiffness variation is on a metre-scale as opposed to the formation scale as described by Fossen et al. (2018). Conversely, in a thrust-fault regime, previous studies documented more evenly distributed bands (Soliva et al., 2013; Ballas et al., 2014; Soliva et al., 2016; Fossen et al., 2018). Variably spaced CSBs in clusters are rarely observed in the contractional regime, and only when associated with large-scale thrust faults (Ballas et al., 2014; Ballas et al., 2015; Soliva et al., 2016;

Fossen et al., 2018). In the Whakataki Formation, pervasively distributed deformation bands are documented for normal- and thrust-fault regimes, with localisation occurring in the vicinity of faults in both cases. Of the twelve faults studied here, only 29% of damage zones associated with a normal fault regime and 40% associated with a thrust fault regime exhibit a positive correlation between CSB spacing and distance from the fault plane. In the thrust fault regime, 90% of outcrops away from faults show distributed patterns. Only twelve faults were studied due to the difficulty in finding appropriately oriented and safe

outcrop surfaces. While this number is not large, the results indicate that there is a pervasive distributed deformation band

pattern that is overprinted by a localised pattern in the vicinity of faults. However, further research would be useful. In total, only seven of the forty studied outcrops are characterised by a clear positive correlation between spacing and distance producing a localised pattern. Therefore, both horizontal extension and contraction involve the pervasive formation of distributed DBs. Both stress regimes also come with localised damage-zone-type CSBs. In summary, our field observations

show a less distinct difference in the spatial distribution of deformation bands as documented previously for normal- and thrust-faulting regimes (Soliva et al., 2013; Ballas et al., 2014; Fossen et al., 2018).

The recognition of deformation bands with regular spacing near faults is based on our interpretation of the statistics of Pearson correlations. Where we observe bimodal or very broad distributions of Pearson coefficients, the spacing distribution is interpreted to reflect progressive deformation and/or multiple tectonic events that have superposed different generations of

840 deformation bands with similar attributes (Fig. 20). This seems to be the most common case in our research area and can be explained with the following conceptual model.

We propose that in the early stages of horizontal extension and horizontal contraction, pervasively distributed CSBs and SECBs form, the latter restricted to horizontal contraction (Fig. 20). The local orientation, spacing, and kinematics of individual bands in this early stage of distributed strain depend on the highly variable rheological properties of the sedimentary layers, layer

thickness, and the orientation of the layers relative to the far-field stress (Gross, 1993; Knott et al., 1996; Martel, 1999; Bai and Pollard, 2000a; Bai and Pollard, 2000b; Olsson and Holcomb, 2000; Ackermann et al., 2001; Rudnicki, 2003; Soliva and Benedicto, 2005; Soliva et al., 2006; Chemenda, 2009; Laubach et al., 2009; Chemenda et al., 2012; Regenauer-Lieb et al., 2013a; Regenauer-Lieb et al., 2013b; Chemenda et al., 2014; Zuza et al., 2017; Laubach et al., 2018). Our microstructural data demonstrate that there is no difference in the deformation mechanisms of CSB and SECB formation. Therefore, only small

changes in the local stress state and yield envelope are required to move from one structure to another because the kinematic transition is continuous and simply reflects different degrees of shear displacement and volume reduction (Fossen and Bale, 2007). With continued strain, favourably located deformation bands can link across sedimentary layers and cause the propagation and formation of throughgoing slip surfaces (Fig. 20) (Aydin and Johnson, 1983; Antonellini et al., 1994; Shipton and Cowie, 2001, 2003; Schueller et al., 2013). Many CSBs associated with immature small faults are sub-parallel to the fault

(Fig. 13), supporting the idea that faults nucleate through linkage of deformation bands across layers. Moreover, at the regional scale, normal- and reverse-faults in the Whakataki Formation are uniformly distributed (Fig. 11). This is consistent with the idea that fault spacing is strongly controlled by the mechanical stratigraphy of the turbidite stack (Fig. 4) (Knott et al., 1996; Martel, 1999; Ackermann et al., 2001; Soliva and Benedicto, 2005; Soliva et al., 2006; Laubach et al., 2009; Zuza et al., 2017; Laubach et al., 2018). Once throughgoing faults have formed, progressive slip is expected to cause the formation of the wall-,

interaction-, and tip-damage zones (Kim et al., 2004; Peacock et al., 2017) that then overprint CSBs with a variable spacing (Shipton and Cowie, 2003; Faulkner et al., 2011; Xu et al., 2012; Maerten et al., 2016). In this scenario of progressive deformation, one would expect a bimodal distribution of Pearson correlation coefficients where the mode that is close to zero

reflects the initial band generation with regular spacing, and a higher mode > 0.5 reflects that of the damage-zone overprint. Moreover, there are many outcrops with distinct generations of cross-cutting structures associated with the same deformation phase as well as those where D3 structures overprint D2 structures. Broad distributions of Pearson coefficients may reflect outcrops that experienced overprint by three or more distinct band-generating events (Figs. 19 & 20).

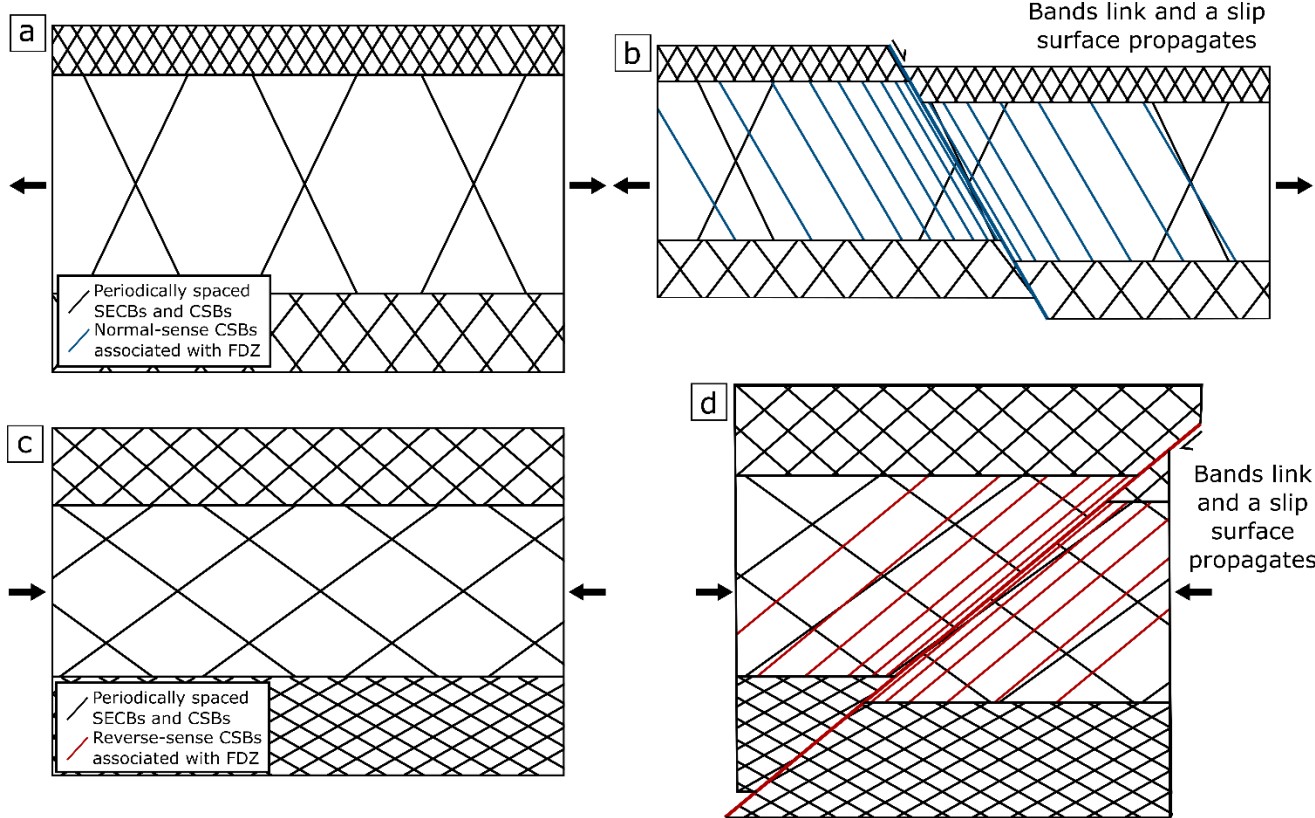

**Figure 20. A schematic to show the evolution of spatial distributions in normal- (a & b) and reverse-faulting (c & d) tectonic regimes. Initially, bands with an equal spacing form. Band spacing is theorised to be proportional to the layer thickness, as is observed with joints (Pollard and Aydin, 1988; Gross, 1993). Eventually, bands link across layers and a fault propagates (Aydin and Johnson, 1983). The fault movement causes the local stress field to be perturbed resulting in new band formation with orientations and kinematics matching that of the new local stress field.**

### 5.3. Tectonic control and the stress path

Stress path modelling was employed in previous field studies to explain the kinematics and orientation of the observed deformation bands as a function of the tectonic regime (Saillet and Wibberley, 2010; Solum et al., 2010; Soliva et al., 2013; Ballas et al., 2014; Fossen et al., 2015; Soliva et al., 2016). We do not use this approach here because of the inherent complexity of the evolution of the state of stress within subduction wedges. Critical wedge theory demonstrates that the state of stress in subduction wedges is controlled by the geometry of the wedge (slope angle and dip angle of the subduction master fault), pore-

fluid pressure, the frictional properties of the wedge, and the basal coefficient of the subduction thrust (Hu and Wang, 2006; Wang and Hu, 2006). The spatial distribution of seismicity along subduction faults implies that the wedge can be separated into a velocity-strengthening section (outer wedge) and a seismogenic velocity-weakening section (inner wedge) (Wang and Hu, 2006). This causes the inner and outer wedges to exhibit significant differences in the mechanical state during failure. The Whakataki Formation has travelled from the top of the outer wedge into the inner wedge and back to the surface. During this journey, the formation has experienced significant changes in a stress state, as clearly indicated by the broad structural inventory (Fig. 3). It is difficult to constrain this stress history from field observations. To emphasise this point, we recall that even within the inner wedge, one can obtain large vertical differences in stress regime, such as upper levels being under horizontal extension while lower levels of the wedge are simultaneously in horizontal contraction and vice versa (Hu and Wang, 2006). The stress regime in the subduction wedge is largely controlled by the basal friction coefficient of the subduction thrust and the pore-fluid pressure. Both parameters cannot be constrained reliably throughout the Miocene from our field observations. Therefore, to generate a comprehensive estimate of the stress path of the Whakataki Formation, at least 2D stochastic mathematical inverse modelling complemented by time-resolved geophysical data and strong geochronological constraints of basin and deformation history is required. This is beyond the scope of the current study.

However, one generic idea proposed in the literature surrounding stress path modelling may apply to our case study: during extension, the mean stress is smaller and the differential stress is higher than during contraction, resulting in an intersection of the yield envelope closer to the top of the cap and thus the formation of CSBs with high $D_s/D_c$ ratios, as observed in D2 bands (Soliva et al., 2013). Regional studies demonstrate that deposition continued following the deposition of the Whakataki Formation in the earliest Miocene throughout the remainder of the Miocene (Neef, 1992a; Neef, 1992b, 1995; Lee et al., 2002; Bailleul et al., 2007). Therefore, the overburden stress probably increased throughout D2. During D3, sediment thickness reached its maximum, and additional vertical thickening through D3 folding and thrust-stacking would have added to the vertical stress. It is, therefore, likely that the mean stress increased from D2 to D3. In addition, a change of the mechanical properties of sandstones from D2 to D3 can be expected, due to porosity reduction associated with ongoing compaction and cementation associated with burial and lithification. The combination of higher mean stresses and a bigger yield envelope in D3 compared to D2 could explain why we also observe more CSBs during D2 extension and more D3 SECBs during D3 contraction (Fig. 21), as proposed by Soliva et al. (2013), Ballas et al. (2014) and Soliva et al. (2016). This idea is certainly appealing in its simplicity. However, it remains to be tested with an inversion of the effective stress path for our study area, which is a challenging problem. Finally, while a static yield model can explain the critical stress state required for failure and the resulting failure angle and deformation band kinematics, it provides no information on the timing, rate of failure, spatial distribution or the number of deformation structures (Zhang et al., 1990; Wong et al., 1992; Wong et al., 1997; Schultz and Siddharthan, 2005; Wong and Baud, 2012). Therefore, one cannot predict the spatial distribution of localised instabilities based on the static far-field stress and static yield envelope alone. Our observations support the well-established notion that the

mechanical stratigraphy plays a major role in determining the spatial distribution of faults and deformation bands in layered rock sequences (Knott et al., 1996; Martel, 1999; Ackermann et al., 2001; Soliva and Benedicto, 2005; Soliva et al., 2006; Laubach et al., 2009; Laubach et al., 2018). In this case, predictive models for the time and spatial evolution of deformation structures must resort to at least 2D mathematical forward modelling (Chemenda et al., 2014).

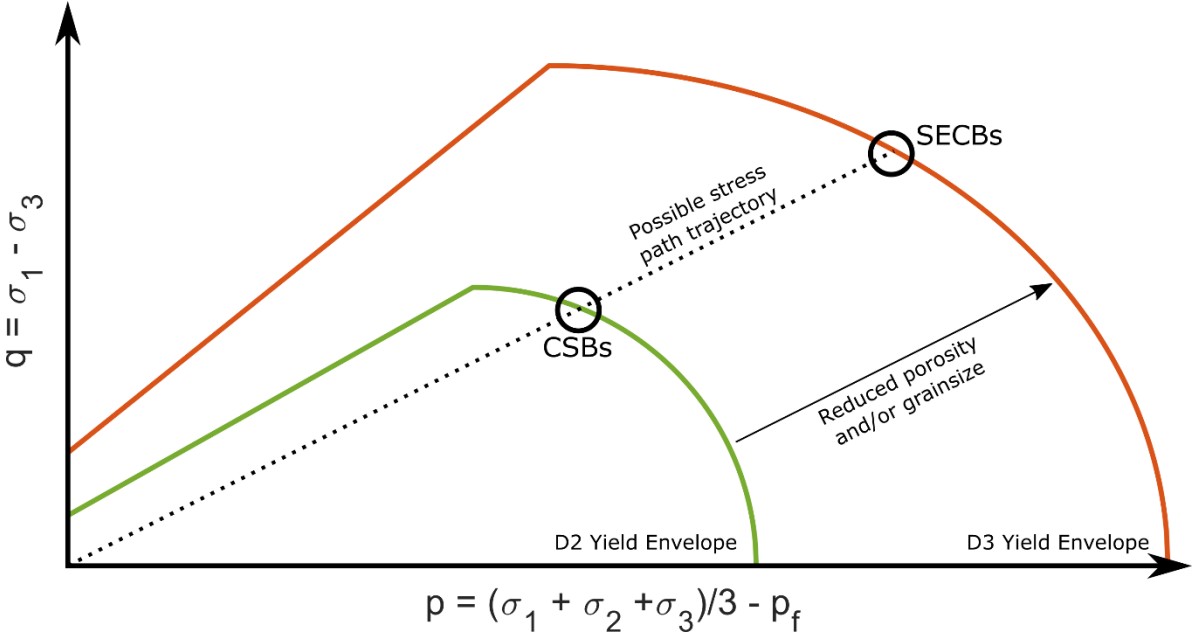

**Figure 21. Schematic diagram of the potential yield envelopes for the Whakataki Formation during D2 horizontal extension and D3 horizontal contraction. During D2, the unit is still lithifying and being buried, therefore, in D3 when the unit has lithified, the yield cap has expanded (red line). For a generic increase in effective stress (p) and differential stress (q), the yield cap may be intersected at different points due to the evolution of the yield envelope through time. With only two points to constrain the stress path, there**
**are many possible trajectories, all of which could result in the observed deformation band kinematics and distributions. In fact, the linear stress path shown here for illustration purposes cannot be correct because the stress regime in the Whakataki Formation switched at least two times. This figure follows the models developed in Soliva et al. (2013).**

### 5.4. Implications for fluid flow

Our field observations demonstrate that the heterogeneous sedimentary architecture, in concert with folds, faults, and
925 deformation bands, transforms the Whakataki Formation into a complex reservoir that is strongly compartmentalised over four orders of magnitude in length scale. At the kilometre scale, some large folds and thrusts juxtapose rock formations with different petrophysical properties. At the 100- to 10 m scale, normal- and reverse faults dissect the Whakataki Formation into monoclinic or triclinic blocks with a dominant thickness of ca. 10 m. D2 and D3 faults show lots of clay smear and gouge in our field area. Therefore, fluid flow inhibition is most likely as clay smears are associated with permeability reductions of
930 several orders of magnitude (Fulljames et al., 1997; Yielding et al., 1997; Nicol and Childs, 2018). At the m- to dm-scale, interbedded mudstones can restrict fluid flow in sandstone layers in the bedding-normal direction, resulting in an anisotropic

hydraulic conductivity on the scale of metres to tens of centimetres. Finally, at the 10 cm-scale deformation bands are present in complex 3D networks. 2D porosity analysis indicates that bands are characterised by porosity reduction compared to the host rock and increased clay content. Therefore, the bands are predicted to also be characterised by reduced permeability and

935 to generate 3D permeability anisotropy with D2 and D3 bands forming roughly perpendicular to each other (Antonellini and Aydin, 1994; Antonellini and Aydin, 1995; Fossen and Bale, 2007; Ballas et al., 2013). Thus, individual sand- and siltstone beds are subdivided yet again into even smaller blocks surrounded by low-permeability boundaries (the deformation bands). The southern Hikurangi subduction wedge is characterised by high rates of sediment accretion with high fluid expulsion rates predicted (Pecher et al., 2010). However, the low taper angle of the wedge indicates that, as a whole, it is overpressured and

940 poorly drained (Barnes et al., 2010). An assessment of the location of gas seeps above the southern Hikurangi subduction wedge indicates that significant volumes of fluid are mobilised along the large-scale thrust faults (Barnes et al., 2010; Pecher et al., 2010). While detailed analysis of the cores of large D2 and D3 faults was not possible due to preferential erosion, there is no reason to expect different behaviour of large faults in the field area. Additionally, the presence of tubular carbonate concretions which correspond to paleoseeps in lower Miocene mudrocks indicates that fluid also migrated along the crests of

945 anticlines and in the footwall of large thrust faults parallel to the subduction margin (Malie et al., 2017). Therefore, we can assume that fluid flowed along large-scale thrust faults and anticline crests in addition to flow through thrust footwalls with restricted flow at a smaller scale to maintain poor bulk drainage (Barnes et al., 2010). The networks of densely spaced compactive deformation bands in the Whakataki Formation are expected to have baffling effects on fluid flow throughout the unit (Sternlof et al., 2006; Fossen and Bale, 2007; Fossen et al., 2015). This is supported by leaching boundaries at the interface

of deformation bands. Regarding petroleum extraction, this can have significant implications if at a sub-seismic scale pervasively distributed deformation bands, formed in horizontal extension and horizontal contraction, can cause small-scale disruption of fluid flow through reservoirs. Although deformation bands rarely exhibit complete continuity in three dimensions for long enough distances to act as trap-forming elements (Fossen et al., 2015), their presence in densely spaced networks may impact fluid flow enough to essentially compartmentalise a reservoir. Additionally, when clusters are hosted in the damage

zones of faults, there is a potential to create hydrocarbon traps (Torabi et al., 2013). Small-scale faults and associated damage zones consisting of deformation bands have previously been observed in extensional regimes but not in contractional regimes (e.g., Fossen et al., 2015). Therefore, the presence of small-scale reverse faults in the Whakataki Formation indicates that similar considerations regarding fluid baffling potential from fault damage zones should be applied to contractional fold-and-thrust belts. However, due to the sub-seismic nature of the faults and damage zones observed in D3 contraction, their presence

may not be discernible with geophysical methods. Thus, the findings in this study are significant for reservoir quality assessments. Small-scale compartmentalisation would require multiple extraction wells and could make a reservoir that is appropriate when analysing seismic data, potentially logistically and economically unsuitable when considering the sub-

seismic structures (Ashton et al., 2018). An integration of the multi-scale structural and sedimentary architecture with computational upscaling of flow properties is required for the prediction of reservoir properties (Field et al., 2006).

## 6. Conclusions

In conclusion, our study on deformation bands in the poly-deformed Miocene turbidites of the Whakataki Formation provides the following key findings:

1. During extension, cataclastic CSBs form.

2. During contraction, cataclastic SECBs form most commonly, but CSBs are also present, particularly in the vicinity of reverse-faults.

3. All deformation bands form dominantly through cataclasis with pore collapse and grain size reduction and result in an absolute porosity reduction of 5-14%.

4. In sedimentary sequences characterised by clay-rich layers, clay smears are commonly observed associated with both horizontal extension and horizontal contraction.

5. Normal faults and thrust faults show regular spacing of ca. 9 m over 10 km.

6. Most outcrops subject to quantitative spacing studies show mixed distributions of Pearson coefficients for the correlation between band spacing and position, which are interpreted as a consequence of multiple generations of deformation bands with similar attributes.

7. Most macroscopic faults probably form through linkage of precursory deformation bands across the stratigraphy. During this process, bands with regular spacing may become overprinted by later wall-, tip-, and interaction damage zones.

8. From a fluid flow perspective, the poly-deformed turbidites of the Hikurangi subduction wedge constitute a highly compartmentalised rock body. Most faults and deformation bands are characterised by fault gouge and thus, likely act as significant barriers to fluid flow.

**Author Contributions**

CES designed the study, and KRL and CES obtained research funding. KEE, CES, and CRS conducted the fieldwork. KEE visualised and analysed field data and conducted all microstructural analyses. KEE prepared the first draft of the manuscript, and all authors contributed to subsequent revisions and interpretations of the data.

## Competing Interests

The authors declare that they have no conflict of interest.

## Acknowledgements

The authors gratefully acknowledge funding by the Australian Research Council through Discovery grant DP170104550, which also supports KEE with a doctoral stipend. Emily and Anders Crofoot of Castlepoint Station and their team are thanked warmly for access to their land and their hospitality. Gabriel Davey provided invaluable assistance in the field and with

digitisation. This work was enabled by the use of the Central Analytical Research Facility hosted by the Institute for Future Environments at QUT. In addition, we thank Gus Luthje and Donald McAuley for thin-section preparation. Finally, we thank G. Ballas and J. Bailleul for their invaluable comments which improved this manuscript.

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
