# Peer review of "Distribution, microphysical properties, and tectonic controls of deformation bands in the Miocene subduction wedge (Whakataki Formation) of the Hikurangi subduction zone"

_Solid Earth, 2020_

## Referee Comment (RC1) · Ballas Grégory (Referee) · 1 Jun 2020

This work is interesting as new field examples of deformation bands are described in the poorly investigated context of accretional prism. This work confirms some recent results concerning tectonic regime controlling deformation band patterns in sandstone and adds two uncommon patterns: (1) Localized faults and shear bands under contraction regime. This is the main result of this study and is potentially linked to the mecano-stratigraphy especially marked in this geological context. This point deserves to be better developed with additional description of bed stratigraphy and fault architecture.

[Figure]

(2) Distributed SECB under normal regime, potentially formed by burial (overloading).

However, these structures are not consistently described (not show in figures, some problems with dihedral angle, distribution is missing...) which affects the impact of this result. I underline that different consistent approaches are used (field mapping, microscopy, image analysis) and the number of data appears adequate to clearly characterize the fault/DB patterns. The literature appears also extensive and consistent with the paper aims.

Because of these reasons, this work deserves to be published in this special issue "Faults, Fractures and Fluid flow" of Solid Earth and could be of interest for any scientists dealing with mechanisms of deformation in porous materials or reservoir evaluations. However, this work contains numerous important issues in the methodology, the data description and the paper organization which have to be managed before any possible publication. I propose major revisions with numerous comments (see below and attached .pdf file with minor suggestions). A second review is certainly needed.

Main comments: *The introduction is in good shape with consistent references. However, the authors exposed the originality of their work with the fact that their study material is not Aeolian sandstones. That's right but I find the geologic context of accretional prism and permuting stress field rather original. At least, modify the text of this section to be consistent with the literature (sandstone of Provence are not Aeolian, maybe introduce also Nubian sandstone in Egypt? Or North Sea?);

*The section 2. Background presents lot of repetitions. I propose to remove or displace the 2.2, 2.3 and 2.4 and only preserve the 2.1. Geologic setting. Move some sentences of the 2.2 concerning the classification of deformation bands from micro mechanisms to the 1. Introduction, thus remove this section 2.2. and the table 1. Remove the 2.3 Spatial distribution already explained in the introduction. Move the 2.4 Conceptual mechanical model to the discussion (also fig. 3);

*I recommend to show outcrop image mapping used for scan-line distribution analysis.

Why not considered spacing > 20m? The general shape of the spacing distribution is generally discuss using Pearson coefficient which is considered as the principal parameter to discuss band patterns. This approach could be interesting if accompanied by precise field observations and descriptions but it can introduce wrong interpretation if consistent field investigations are not done. I encourage you to develop description of mean band spacing, if possible from field measurement, and show several examples in figures. Remove the section concerning theoretical structure distribution (l.288-304 and section 4.5.1);

*The Pearson coefficient is also use to describe fault patterns (bimodal – polymodal). Again, this approach can introduce major wrong interpretation as a function of the measurements were done on the field. I recommend to better developed observations and field description before to use this statistical approach;

*The microfracture density have to be quantify from surface mapping (on SEM image) and not from scanline orientated normal to the band. This introduces an important bias as micro-cracks strike along force chains with specific angle to the bands;

*The description of structures is confusing as the text is following a chronological order, whereas figures are classified by type of data (macro, micro, grain size analysis, petrophysics). I encourage you to clearly separate data of your 3 events D1, D2, D3 to match with the text description (just keep the figure 9a on porosity with the total dataset for comparison);

*Indicate more precisely which part of each figure (a, b, c, d, e. . .) is concerned by citation in the text will help. Start your figure in the consistent chronology (generally D3 structures are firstly described in figure). I encourage you to add some indications on the figure.

*The Ds/Dc values appears inconsistent with data description, please check them;

*Extend the description of micro-mechanisms of D2 structures, the image you shown

fig. 7f is too limited to clearly expose the deformation process and the band micro-geometry you described in the text. Concerning these bands, you described a cata-clastic process but this is not consistent with the negative relief they show on the field and their dark color. How explain that? It is not evident also in SEM images. Any important of clay (phyllo bands?) or disaggregation, or cementation (Organic Matter)? The observations of Fig. 6a-c rather argue for disaggregation bands.

*Qz overgrowths are not consistently described (wrong interpretation in the data de-scription), add more precise observations. It could be of interest to constrained how evolve the mechanical properties and the petrophysics of the material.

*Clearly separate DB in fault Damage Zone and DB out of fault DZ in the description of D3, as done within the following distribution description.

*Explain how damage zone of fault thickness is defined.

*Use figure 14 within the data description and remove the figure 15 (not used).

*Develop the description of normal-sense SECB if you want to maintain the discussion concerning band type and distribution vs. tectonic regime l.720-736 and l. 776-784. I encourage you to do it, these normal-sense structures potentially formed by burial in-crease could be very interesting. If it is not possible, remove this part of the discussion.

*The mechanical approach exposed in figure 17 is not enough constrained to be con-sistently discuss (both stress path and yield envelope are not estimate from data). The hypothesis of compaction between D2 and D3 appears inconsistent with description. Think about strengthening by the D2 band pattern or cementation process to explain a potential increase of the yield envelope. However, the change of boundary stress conditions (extensional regime – contractional regime) and the presence of localized faults could explain this change of band properties from D2 to D3.

Grégory Ballas Structural Geologist, Université de Montpellier, Laboratoire Géo-sciences Montpellier

Please also note the supplement to this comment:
https://www.solid-earth-discuss.net/se-2020-80/se-2020-80-RC1-supplement.pdf

---

## Referee Comment (RC2) · Julien Bailleul (Referee) · 4 Sep 2020

General comments

The study proposed by Elphick et al. appears to perfectly fit with this Special Issue of SOLID EARTH. The paper describes for the first time deformation bands in syn-subduction turbidites of the Hikurangi margin (New Zealand) and brings insights concerning their relation to deformation episodes that have affected the subduction wedge. This study is original in New Zealand and for such a geological context and relies on re-

cent development concerning the understanding of deformation bands. In my opinion, emphasis should be given on: 1) the geological context, and above all, 2) the turbidites which are particular host rocks for deformation bands. The paper is well written with globally a well-organized and clear structure even if some specific and technical points have to be amended to improve the manuscript (see next section and annotation in the PDF file).

Specific comments

- As accretion did not occurred all across the thrust wedge, I rather prefer to avoid the use of accretionary prism to talk of the entire deformed area between the trench and the Forearc basin (please, see also my point of view on that in the PDF file of the manuscript).

- An absolute porosity reduction of ca. 10% (from $\sim$ 20% to 10%) is given in the conclusions whereas between 5 and 14% is given in the abstract. Also, c.a. 9 m is given in the conclusions while 10 m is given in the abstract for the spacing between faults. I think it can be clearer to give the same values in both the abstract and conclusions.

- The Background section could be clarified with a better separation between what concern the geological context and what concern deformation bands.

- Proto-Deformation Bands (section 4.4.3.3): The description is quite short, could you give us more detail and explain more clearly why you interpret these as proto-DB?

- It is said in the discussion that deposition of the Whakataki Fm occurred till the end of D2. As far as I know, it's not the case, the typically thin-bedded turbidites of that Formation being Waitakian-Otaian in age, so latest Oligocene to earliest Miocene.

- The section 5.4. Implications for fluid flow is very general and could be consolidated a little bit, notably incorporating references to papers that specifically deals with fluids on the Hikurangi margin – that literature is abundant so only a few key papers could be cited: (e.g. …. See annotation in the manuscript for suggestion of references). That

would give weight to the consequences of the compartmentalization demonstrated in the paper on reservoir properties and for pressure regimes.

- The paper could gain from a last figure acting as a synthesis summarizing most of the observations and results, on a 3D block for example. Such a figure could highlight the work and make it more visible.

- Some figures deserve to be more cited in the text (e.g. fig. 15 is not cited at all) and some more precisely (i.e. what part of the figure is used).

- References: some inconstancies are still present in the list and a few references could be added. Please, verify all the references to be sure that they are OK.

Technical corrections

Technical suggestions are annotated in the PDF file of the manuscript.

Considering all the points raised above and those annotated in the PDF File of the manuscript, I recommend moderate revision for that paper proposal.

Thank you for having proposing me that review, Best Regards,

Julien Bailleul

Please also note the supplement to this comment:
https://se.copernicus.org/preprints/se-2020-80/se-2020-80-RC2-supplement.pdf

**Supplement:**

[revised manuscript text omitted]

---

## Author Response (AR1)

**Response to comments from Reviewer 1 (Gregory Ballas) and Reviewer 2 (Julien Bailleul) on "Distribution, microphysical properties, and tectonic control of deformation bands in the Miocene accretionary prism (Whakataki Formation) of the Hikurangi subduction zone"**

**Gregory Ballas**

**Comment**

This work is interesting as new field examples of deformation bands are described in the poorly investigated context of accretional prism. This work confirms some recent results concerning tectonic regime controlling deformation band patterns in sandstone and adds two uncommon patterns: (1) Localized faults and shear bands under contraction regime. This is the main result of this study and is potentially linked to the mecanostratigraphy especially marked in this geological context. This point deserves to be better developed with additional description of bed stratigraphy and fault architecture. (2) Distributed SECB under normal regime, potentially formed by burial (overloading).

However, these structures are not consistently described (not show in figures, some problems with dihedral angle, distribution is missing. . .) which affects the impact of this result. I underline that different consistent approaches are used (field mapping, microscopy, image analysis) and the number of data appears adequate to clearly characterize the fault/DB patterns. The literature appears also extensive and consistent with the paper aims.

Because of these reasons, this work deserves to be published in this special issue "Faults, Fractures and Fluid flow" of Solid Earth and could be of interest for any scientists dealing with mechanisms of deformation in porous materials or reservoir evaluations. However, this work contains numerous important issues in the methodology, the data description and the paper organization which have to be managed before any possible publication. I propose major revisions with numerous comments (see below and attached .pdf file with minor suggestions). A second review is certainly needed.

**Response**

We thank Gregory Ballas for reviewing this manuscript and for providing valuable critical insight and analysis that will allow us to improve and clarify the manuscript. We appreciate and are grateful that the reviewer thinks that this work deserves to be published within the special issue. Three main points are raised by the reviewer.

The first point refers to one of the main findings of the research that has not been given enough focus and description within the article. We thank the reviewer for pointing this out and have amended the manuscript to have a larger focus on the presence of localised faults and shear bands under a contractional regime. The role of mechanostratigraphy has been given more focus and discussion throughout the manuscript.

The second point highlights the documentation of SECBs within the normal fault regime. Upon reflection of the comments from the reviewer we have reassessed the data and have come to the conclusion that the structures do not represent SECBs but rather CSBs that have variable displacement along strike. While the structures do in places seem to have no apparent shear offset, this may be a sectioning effect. Furthermore, we observe that the lithology impacts the displacement associated with a structure with greater shear offset observed as bands propagate through clay-rich layers and reduced offset when propagating through shell-hash rich layers, for example. Therefore, in light of the comments from the reviewer, we have changed the results to show only CSBs present during horizontal extension, not SECBs, consistent with other studies examining deformation bands formed under an extensional regime (e.g., Rotevatn et al., 2008; Saillet and Wibberley, 2010; Solum et al., 2010; Soliva et al., 2013; Ballas et al., 2014; Soliva et al., 2016).

The third main point is regarding the methodology, data description, and paper organisation. In regard to methodology, issues surrounding analysis of complete datasets rather than individual outcrops has been addressed. We now discuss both scales and highlight a problem with only focussing on one of the two scales. The figures have been re-made to align with the order of data description within the text. Data has been described in more detail where necessary and appropriate. However, in our opinion, most of the data is described to a level that is required for the key points of the article and more description would result in lengthening of an already long manuscript.

**Main comments:**

**Comment**

*The introduction is in good shape with consistent references. However, the authors exposed the originality of their work with the fact that their study material is not Aeolian sandstones. That's right but I find the geologic context of accretional prism and permuting stress field rather original. At least, modify the text of this section to be consistent with the literature (sandstone of Provence are not Aeolian, maybe introduce also Nubian sandstone in Egypt? Or North Sea?);

**Response**

We have introduced the Nubian sandstone and sandstones hosted in the North Sea, in addition to providing more insight into the lack of studies focussing on DBs within subduction wedge settings. In doing so, we have also expanded the section referring to the lithology and how the lithology studied in this research is unique regarding DB studies, thereby further exposing the originality of the research.

**Comment**

*The section 2. Background presents lot of repetitions. I propose to remove or displace the 2.2, 2.3 and 2.4 and only preserve the 2.1. Geologic setting. Move some sentences of the 2.2 concerning the classification of deformation bands from micro mechanisms to the 1. Introduction, thus remove this section 2.2. and the table 1. Remove the 2.3 Spatial distribution already explained in the introduction. Move the 2.4 Conceptual mechanical model to the discussion (also fig. 3);

**Response**

Sections regarding classification of DBs have been shortened and incorporated into the introduction and a brief synopsis of 2.4 added into introduction. We did not, however, incorporate Figure 3 into the discussion as suggested because we believe that a figure to explain the cam-cap yield envelope is required in the introduction to provide a background into the idea that tectonic setting influences the band kinematics and orientation.

**Comment**

*I recommend to show outcrop image mapping used for scan-line distribution analysis. Why not considered spacing > 20m?

**Response**

We had already included outcrop images of the maps used for scan-line distribution analysis in Figs. 17 & 18. Spacing >20 m has not been considered as it represents areas of no exposure along beaches associated with stream mouths. To include such data would influence the spacing results erroneously.

**Comment**

The general shape of the spacing distribution is generally discuss using Pearson coefficient which is considered as the principal parameter to discuss band patterns. This approach could be interesting if accompanied by precise field observations and descriptions but it can introduce wrong interpretation if consistent field investigations are not done. I encourage you to develop description of mean band spacing, if possible from field measurement, and show several examples in figures. Remove the section concerning theoretical structure distribution (l.288-304 and section 4.5.1);

**Response**

We respectfully disagree with the reviewer and would like to point out that statistical approaches to characterising fracture patterns are common and very important in Structural Geology (add long list of references to methods papers in this field, including those from Sanderson's work, FracPaq, etc.). Most commonly, the aim of reporting a mean is to determine the central tendency of a distribution or its most probable, common value. This approach makes a lot of sense when one considers constant spacing with random noise (which should be normally distributed). However, it is not useful when considering spacing that varies systematically with position, especially when this variation is nonlinear. Our simple statistical approach can be seen as a simplified version of that of "Sanderson, D. J. and Peacock, D. C.: Line sampling of fracture swarms and corridors, Journal of Structural Geology, 122, 27-37, 2019.". It is designed to test if a deformation band distribution is a function of position (mean not useful) or if it is constant with random noise (mean useful). We have kept the section regarding theoretical structure distribution in the text because it is needed to understand our simplified statistical approach to the mapping and can act as a check for other researchers to compare their datasets with.

In terms of additional figures: we provide three representative examples for bands showing periodic and aperiodic spacing. While many others were measured (see Figs. 9 & 19 for statistical results), we did not feel it appropriate to include all examples within the manuscript. If the editorial team indicates a requirement for such data, it can be added to an appendix.

**Comment**

*The Pearson coefficient is also use to describe fault patterns (bimodal – polymodal). Again, this approach can introduce major wrong interpretation as a function of the measurements were done on the field. I recommend to better developed observations and field description before to use this statistical approach;

**Response**

Regarding fault patterns, we have used the approach of Healy and Jupp (2018) to test if the fault orientation distribution is bimodal. This is a published, mathematically sound workflow that can be compared to other datasets and is reproducible and reliable. It also minimises observer bias because it is quantitative.

Unfortunately, the reviewer does not state explicitly which specific problems can arise from the use of the objective statistical methods employed in our paper. Thus, it is difficult to respond to the criticism concisely. We are certainly aware of, and explicitly mention and discuss, the most important

source of sampling bias in our study: exposure bias. The hinterland outcrops are rare and poorly preserved. Therefore, our work is restricted to the coast which has excellent outcrop. However, the coast is sub-parallel to the dominant strike of major thrusts and folds. As a result, it is expected and not surprising that we observe fewer D3 thrust (mentioned in the paper in section 4.4.2.) than D2 faults, which strike at a high angle to the coast. In addition, most of our study sites sit on the back limb of a large D3 syncline, and because of this particular exposure bias, we cannot study how deformation band and fault distribution varies with position across large-scale folds. Moreover, the coastal outcrops are (fairly unstable) cliffs bordered by rock platforms. So not every single bed featuring deformation bands yields good opportunities to document them in 3D. Nevertheless, exposure bias is a common problem for all geological research. At the coast, exposure is > 80%, which we consider excellent. We do not see how the use of objective statistical methods in the analysis of fault/DB orientation data and line maps constitutes a major problem.

**Comment**

*The microfracture density have to be quantify from surface mapping (on SEM image) and not from scanline orientated normal to the band. This introduces an important bias as micro-cracks strike along force chains with specific angle to the bands;

**Response**

This dataset will be removed.

**Comment**

*The description of structures is confusing as the text is following a chronological order, whereas figures are classified by type of data (macro, micro, grain size analysis, petrophysics). I encourage you to clearly separate data of your 3 events D1, D2, D3 to match with the text description (just keep the figure 9a on porosity with the total dataset for comparison);

**Response**

We agree with this comment and have changed the figures accordingly. We thank the reviewer for this comment as it has enabled us to better describe the data with clearer figures.

**Comment**

*Indicate more precisely which part of each figure (a, b, c, d, e. . .) is concerned by citation in the text will help. Start your figure in the consistent chronology (generally D3 structures are firstly described in figure). I encourage you to add some indications on the figure.

**Response**

Agreed and implemented.

**Comment**

*The Ds/Dc values appears inconsistent with data description, please check them;

**Response**

The Ds/Dc values have inherent error because many bands do not show offset in the thin section, therefore, the offset value has come from the field measurements. The error in the measurement will then be carried into the estimate. Additionally, the porosity is measured in 2D for the DB and the HR so there is error. However, the Ds/Dc values do suggest more CSBs with large shear offset and that has been addressed in the text. We thank the reviewer for their detailed analysis of the research to highlight this.

**Comment**

*Extend the description of micro-mechanisms of D2 structures, the image you shown fig. 7f is too limited to clearly expose the deformation process and the band microgeometry you described in the text. Concerning these bands, you described a cataclastic process but this is not consistent with the negative relief they show on the field and their dark color. How explain that? It is not evident also in SEM images. Any important of clay (phyllo bands?) or disaggregation, or cementation (Organic Matter)? The observations of Fig. 6a-c rather argue for disaggregation bands.

**Response**

Clearer figures have been made for the microstructure (Figs. 8, 12, 14). There is always significant grainsize reduction in the bands accommodated by cataclasites. Therefore, the bands are generally cataclastic, even when they transect layers with high clay content. See the new Fig. 6 in the revised manuscript, please.

**Comment**

*Qz overgrowths are not consistently described (wrong interpretation in the data description), add more precise observations. It could be of interest to constrained how evolve the mechanical properties and the petrophysics of the material.

**Response**

We agree that quartz cementation has not been addressed properly. To alleviate this problem, substantial additional analyses such as high-resolution cathodoluminescence imaging would be required. Considering that this paper focuses on outcrop scale descriptions of DB and that the present

BSE images permit to identify cataclasis as certainly the dominant deformation mechanism, we removed the images indicating the presence of quartz cement and this will be the focus of further research.

**Comment**

*Clearly separate DB in fault Damage Zone and DB out of fault DZ in the description of D3, as done within the following distribution description.

**Response**

Figures have been added to the manuscript and previous image panels have been reordered to split the images into their respective deformation phases. The figures now align with the descriptions.

**Comment**

*Explain how damage zone of fault thickness is defined.

**Response**

This has been addressed in lines 660-661.

**Comment**

*Use figure 14 within the data description and remove the figure 15 (not used).

**Response**

Done as suggested.

**Comment**

*Develop the description of normal-sense SECB if you want to maintain the discussion concerning band type and distribution vs. tectonic regime l.720-736 and l. 776-784. I encourage you to do it, these normal-sense structures potentially formed by burial increase could be very interesting. If it is not possible, remove this part of the discussion.

**Response**

We thank the reviewer for this comment. As aforementioned, these structures cannot be definitively identified using multiple methods of identification. However, CSBs can. Therefore, we have changed the text to show CSBs associated with horizontal extension.

**Comment**

*The mechanical approach exposed in figure 17 is not enough constrained to be consistently discuss (both stress path and yield envelope are not estimate from data). The hypothesis of compaction

between D2 and D3 appears inconsistent with description. Think about strengthening by the D2 band pattern or cementation process to explain a potential increase of the yield envelope. However, the change of boundary stress conditions (extensional regime – contractional regime) and the presence of localized faults could explain this change of band properties from D2 to D3.

**Response**

In lines 893-914 we discuss in detail why it is very challenging to determine the stress path for our rocks of interest and state that it is beyond the scope of our study. We explicitly highlight that the simplistic model discussed in Fig. 21 is purely speculative and requires further testing based on an inversion of the stress path. All we do is to state that there is a non-zero possibility that the mean stress and the strength of the rock package increase from D2 to D3, and thus one could be tempted to apply similar arguments as those in literature to explain the difference in D2 and D3 bands. This issue is already discussed at great length and with due care in lines 904 to 913. We do not believe that it requires further discussion.

**Comments from within the text**

The reviewer has provided numerous other references to add into the text and to broaden the scope of the research. We have implemented these changes and read the relevant papers to improve the manuscript and thank the reviewer for adding the references to the comments.

The reviewer has identified the grouping of clusters to represent one band as a bias. It is certainly a simplification. However, as shown now in Figs. 6 & 13, a band can turn into a cluster downdip and back into a single band again. This is a function of host rock and now mentioned in the text. We follow the approach of Main et al (2000) and consider clusters as a single band. Any error introduced by this averaging is usually smaller than the pattern variability captured by using the scanline approach.

The reviewer has asked why we normalised spacing data. We did so to enable analysis and comparison of multiple outcrops.

The reviewer suggested that figure 20 be made more realistic. We have kept the original figure because this is a schematic to show a possible order of events in any tectonic setting and would like this to be simple and understandable. The process of overprinting structures within damage zones is very complex and we do not believe that there is a great enough understanding of the process for the image to be made more realistic because in doing so, it would be less realistic.

**Julien Bailleul**

**Comment**

The study proposed by Elphick et al. appears to perfectly fit with this Special Issue of SOLID EARTH. The paper describes for the first time deformation bands in synsubduction turbidites of the Hikurangi margin (New Zealand) and brings insights concerning their relation to deformation episodes that have affected the subduction wedge. This study is original in New Zealand and for such a geological context and relies on recent development concerning the understanding of deformation bands. In my opinion, emphasis should be given on: 1) the geological context, and above all, 2) the turbidites which are particular host rocks for deformation bands. The paper is well written with globally a well-organized and clear structure even if some specific and technical points have to be amended to improve the manuscript (see next section and annotation in the PDF file).

**Response**

We thank Julien Bailleul for the constructive comments and critical analysis of the research. The comments from the reviewer have enabled us to significantly improve the manuscript. The associated changes of the manuscript are described in detail in the following.

**Main comments**

**Comment**

- As accretion did not occurred all across the thrust wedge, I rather prefer to avoid the use of accretionary prism to talk of the entire deformed area between the trench and the Forearc basin (please, see also my point of view on that in the PDF file of the manuscript).

**Response**

Thank you for pointing out this semantic oversight. We have amended the manuscript accordingly.

**Comment**

- An absolute porosity reduction of ca. 10% (from ~ 20% to 10%) is given in the conclusions whereas between 5 and 14% is given in the abstract. Also, c.a. 9 m is given in the conclusions while 10 m is given in the abstract for the spacing between faults. I think it can be clearer to give the same values in both the abstract and conclusions.

**Response**

Thank you for alerting us to these inconsistencies. They have been corrected.

**Comment**

- The Background section could be clarified with a better separation between what concern the geological context and what concern deformation bands.

**Response**

In response to comments from Reviewer 1, we have changed and reduced the background and the main points have been added into the introduction. We believe that this change also accommodates the comment from this reviewer.

**Comment**

- Proto-Deformation Bands (section 4.4.3.3): The description is quite short, could you give us more detail and explain more clearly why you interpret these as proto-DB?

**Response**

Dataset removed. Not important for the study and can be focussed on with future study.

**Comment**

- It is said in the discussion that deposition of the Whakataki Fm occurred till the end of D2. As far as I know, it's not the case, the typically thin-bedded turbidites of that Formation being Waitakian-Otaian in age, so latest Oligocene to earliest Miocene.

**Response**

This has been amended in lines 895-897.

**Comment**

- The section 5.4. Implications for fluid flow is very general and could be consolidated a little bit, notably incorporating references to papers that specifically deals with fluids on the Hikurangi margin – that literature is abundant so only a few key papers could be cited: (e.g. . . .. See annotation in the manuscript for suggestion of references). That would give weight to the consequences of the compartmentalization demonstrated in the paper on reservoir properties and for pressure regimes.

**Response**

This section has been extended and given more focus surrounding literature about the Hikurangi margin to contextualise the study while still making it applicable to different settings. The southern Hikurangi subduction wedge has been given more focus (e.g., lines 938-950).

**Comment**

- The paper could gain from a last figure acting as a synthesis summarizing most of the observations and results, on a 3D block for example. Such a figure could highlight the work and make it more visible.

**Response**

We already include a 3D block diagram for the deformation phases in Figure 3, which focuses on the macro-structures and includes an inset for DB for D3. The orientation of DBs for D2 share the same orientations as D3 DBs hosted in damage zones. This relationship has been added to the figure caption for Figure 3.

**Comment**

- Some figures deserve to be more cited in the text (e.g. fig. 15 is not cited at all) and some more precisely (i.e. what part of the figure is used).

**Response**

We agree with this comment and have amended the text to have a better link between the text and figures.

**Comment**

- References: some inconstancies are still present in the list and a few references could be added. Please, verify all the references to be sure that they are OK.

**Response**

Agreed and modified. Again, we thank the reviewer for their attention to detail and it has made the work more reproducible.

**References**

[revised manuscript text omitted]

Commented [LE3]: Here, we have adapted the original Figure 3 from the background to describe the cam-cap yield envelope within the introduction in response to Gregory Ballas' comments.

[revised manuscript text omitted]

Commented [LE4]: Gregory Ballas: *The introduction is in good shape with consistent references. However, the authors exposed the originality of their work with the fact that their study material is not Aeolian sandstones. That's right but I find the geologic context of accretional prism and permuting stress field rather original. At least, modify the text of this section to be consistent with the literature (sandstone of Provence are not Aeolian, maybe introduce also Nubian sandstone in Egypt? Or North Sea?);

Response:
We have introduced the Nubian sandstone and sandstones hosted in the North Sea, in addition to providing more insight into the lack of studies focussing on DBs within subduction wedge settings. In doing so, we have also expanded the section referring to the lithology and how the lithology studied in this research is unique regarding DB studies, thereby further exposing the originality of the research.

[revised manuscript text omitted]

**Commented [LE6]:** Gregory Ballas: *The section 2. Background presents lot of repetitions. I propose to remove or displace the 2.2, 2.3 and 2.4 and only preserve the 2.1. Geologic setting. Move some sentences of the 2.2 concerning the classification of deformation bands from micro mechanisms to the 1. Introduction, thus remove this section 2.2. and the table 1. Remove the 2.3 Spatial distribution already explained in the introduction. Move the 2.4 Conceptual mechanical model to the discussion (also fig. 3);

Response:
Sections regarding classification of DBs have been shortened and incorporated into the introduction and a brief synopsis of 2.4 added into introduction. We did not, however, incorporate Figure 3 into the discussion as suggested because we believe that a figure to explain the cam-cap yield envelope is required in the introduction to provide a background into the idea that tectonic setting influences the band kinematics and orientation.

**Commented [LE7]:** Julien Bailleul:
- The Background section could be clarified with a better separation between what concern the geological context and what concern deformation bands.

Response:
In response to comments from Gregory Ballas, we have changed and reduced the background and the main points have been added into the introduction. We believe that this change also accommodates the comment from this reviewer.

[revised manuscript text omitted]

**Commented [LE8]:** Julien Bailleul:
- The paper could gain from a last figure acting as a synthesis summarizing most of the observations and results, on a 3D block for example. Such a figure could highlight the work and make it more visible.

Response:
We already include a 3D block diagram for the deformation phases in Figure 3, which focuses on the macro-structures and includes an inset for DB for D3. The orientation of DBs for D2 share the same orientations as D3 DBs hosted in damage zones. This relationship has been added to the figure caption for Figure 3.

[revised manuscript text omitted]

**Commented [LE9]:** Gregory Ballas:
*The Pearson coefficient is also use to describe fault patterns (bimodal – polymodal). Again, this approach can introduce major wrong interpretation as a function of the measurements were done on the field. I recommend to better developed observations and field description before to use this statistical approach;

Response:
Regarding fault patterns, we have used the approach of Healy and Jupp (2018) to test if the fault orientation distribution is bimodal. This is a published, mathematically sound workflow that can be compared to other datasets and is reproducible and reliable. It also minimises observer bias because it is quantitative. Unfortunately, the reviewer does not state explicitly which specific problems can arise from the use of the objective statistical methods employed in our paper. Thus, it is difficult to respond to the criticism concisely. We are certainly aware of, and explicitly mention and discuss, the most important source of sampling bias in our study: exposure bias. The hinterland outcrops are rare and poorly preserved. Therefore, our work is restricted to the coast which has excellent outcrop. However, the coast is sub-parallel to the dominant strike of major thrusts and folds. As a result, it is expected and not surprising that we observe fewer D3 thrust (mentioned in the paper in section 4.4.2.) than D2 faults, which strike at a high angle to the coast. In addition, most of our study sites sit on the back limb of a large D3 syncline, and because of this particular exposure bias, we cannot study how deformation band and fault distribution varies with position across large-scale folds. Moreover, the coastal outcrops are (fairly unstable) cliffs bordered by rock platforms. So not every single bed featuring deformation bands yields good opportunities to document them in 3D. Nevertheless, exposure bias is a common problem for all geological research. At the coast, exposure is > 80%, which we consider excellent. We do not see how the use of objective statistical methods in the analysis of fault/DB orientation data and line maps constitutes a major problem.

[revised manuscript text omitted]

**Commented [LE10]:** Gregory Ballas:
The general shape of the spacing distribution is generally discuss using Pearson coefficient which is considered as the principal parameter to discuss band patterns. This approach could be interesting if accompanied by precise field observations and descriptions but it can introduce wrong interpretation if consistent field investigations are not done. I encourage you to develop description of mean band spacing, if possible from field measurement, and show several examples in figures. Remove the section concerning theoretical structure distribution (l.288-304 and section 4.5.1);

Response:
We respectfully disagree with the reviewer and would like to point out that statistical approaches to characterising fracture patterns are common and very important in Structural Geology (Soliva and Benedicto, 2005; Tang et al., 2008; Rabinovitch et al., 2012; Hooker et al., 2013; Guo et al., 2017; Healy et al., 2017; Laubach et al., 2018; Sanderson and Peacock, 2019). Most commonly, the aim of reporting a mean is to determine the central tendency of a distribution or its most probable, common value. This approach makes a lot of sense when one considers constant spacing with random noise (which should be normally distributed). However, it is not useful when considering spacing that varies systematically with position, especially when this variation is nonlinear. Our simple statistical approach can be seen as a simplified version of that of "Sanderson, D. J. and Peacock, D. C.: Line sampling of fracture swarms and corridors, Journal of Structural Geology, 122, 27-37, 2019.". It is designed to test if a deformation band distribution is a function of position (mean not useful) or if it is constant with random noise (mean useful). We have kept the section regarding theoretical structure distribution in the text because it is needed to understand our simplified statistical approach to the mapping and can act as a check for other researchers to compare their datasets with.
In terms of additional figures: we provide three representative examples for bands showing periodic and aperiodic spacing. While many others were measured (see Figs. 9 & 19 for statistical results), we did not feel it appropriate to include all examples within the manuscript. If the editorial team indicates a requirement for such data, it can be added to an appendix.

[revised manuscript text omitted]

**Commented [LE11]:** Gregory Ballas has identified the grouping of clusters to represent one band as a bias. It is certainly a simplification. However, as shown now in Figs. 6 & 13, a band can turn into a cluster downdip and back into a single band again. This is a function of host rock and now mentioned in the text. We follow the approach of Main et al. (2000) and consider clusters as a single band. Any error introduced by this averaging is usually smaller than the pattern variability captured by using the scanline approach.

**Commented [LE12]:** Gregory Ballas has asked why we normalised spacing data. We did so to enable analysis and comparison of multiple outcrops.

**Commented [LE13]:** Gregory Ballas:
Remove the section concerning theoretical structure distribution (l.288-304 and section 4.5.1);

Response:
We have kept the theoretical structure distribution to support the discussion of results from the spacing analysis of outcrops.

[revised manuscript text omitted]

**Commented [LE17]:** Gregory Ballas:
*The description of structures is confusing as the text is following a chronological order, whereas figures are classified by type of data (macro, micro, grain size analysis, petrophysics). I encourage you to clearly separate data of your 3 events D1, D2, D3 to match with the text description (just keep the figure 9a on porosity with the total dataset for comparison);

Response:
We agree with this comment and have changed the figures accordingly. We thank the reviewer for this comment as it has enabled us to better describe the data with clearer figures.

morphology during D3 (Cape et al., 1990; Chanier et al., 1999; Bailleul et al., 2013). Rare metre-scale, NE-trending upright, moderately NE-SW-plunging, open D1F1 folds are still preserved in the area (Fig. 2Figs. 3a & 3e).

[Figure]

[Figure]

**Figure 5.5. Examples of overprinting criteria used to determine the relative timing of each deformation phase. [a)] D3 reverse fault offsetting D2 normal fault; [b] layer-parallel slip offsetting a D3 reverse fault;** b) D3 reverse fault offsetting D2 normal fault; [c)] **multiple generations of deformation band located within a D3 damage zone (green = D1, blue = D2, red = D3); [d] and [e)] multiple generations of deformation band with overprints.** The scale bar **(green = D1, blue = D2, red = D3). Black circles in** all images is 20 cm.**[e] signify some examples of overprints.**

**4.2.2. Deformation Bands**

D1 deformation bandsDBs are rare and are distinguished from D2 and D3 deformation bandsDBs through cross-cutting relationships (Fig. 5Figs. 5c, 5d, 5e). The orientation distribution of D1 deformation bandsDBs can be considered bimodal with a *p-value* of 0.5. D1 DB dihedral angle rangesangles range from 65° to 84° with a mean of 73°. With bedding restored to horizontal, deformation bandsDBs trend 037°.° (Fig. 3i). The set dipping SE has a dip angle of 40° and the set dipping NW dips at 26°.° (Fig. 3i). When bands pass through beds with thin clay-rich layers, they have a dark colour without relief. In beds with lower clay content, bands show high to moderate relief and are lighter in colour (Fig. 5Figs. 5d & 5e). The width of bands ranges from 0.2 to 0.5 mm. Bands can extend for metres along strike. Eye and ramp structures are present, however. However, single strands are most commonly observed (Antonellini and Aydin, 1995).(Antonellini and Aydin, 1995). Offset associated with the bands is variable, with some characterised by minimal offset, yet others accommodate reverse shear offsets at the millimetre scale. Conclusively, this data suggeststhese observations suggest that these bands are CSBs and Shear-Enhanced

Compaction Bands (SECBs) formed during D1 horizontal contraction (Ballas et al., 2014; Fossen et al., 2018; Schultz, 2019). The rarity of D1 deformation bands does not permit for meaningful spacing analysis.

**4.3. D2 Horizontal Extension**

**4.3.1. Structures and Relative Timing**

Field overprinting criteria demonstrate that an extensional event followed D1 horizontal contraction (Fig. 5) (Chanier et al., 1999).(Chanier et al., 1999). Overprinting relations include the displacement of D1F1 fold hinges by D2 normal faults and normal-sense shear displacement of D1 deformation bands by D2 deformation bands. (Figs. 5c, 5d, 5e). The structures associated with D2 are [1] normal faults;, [2] deformation bands with clear normal-sense offset, and [3] deformation bands with minimal offset. (Figs. 2, 5a, 6, 7).

**4.3.2. Faults**

Normal faults are the main structures associated with D2 (Fig. 6Figs. 4a & 4b). They trend NNW with an average dip angle of 69° (Fig. 23f). Analysis of the fault pattern shows that it is polymodal with a *p-value* of 0.001 (Healy and Jupp, 2018). Fault displacement ranges from centimetres to tens of metres. Many D2 faults have displacements of ca. 1 m.(Healy and Jupp, 2018). While polymodal across the field area, normal faults are commonly observed as two sets with opposing dips at individual outcrops (Fig. 3f). Fault displacement ranges from tens of centimetres to tens of metres. Many D2 faults have displacements of ca. 1 m (e.g., Fig. 7). The limited vertical extent of the cliff exposures and poor hinterland exposure preclude estimates of upper-bound displacement. (Fig. 4). Fault length can also not be determined as they generally extend beyond cliff height. The faults are dominantly brittle, with gouge present in the core of larger faults. (Figs. 4 & 6d).

[Figure]

**Commented [LE19]:** Gregory Ballas: align figures with text i.e., panels for each type of DB

[revised manuscript text omitted]

**Commented [LE20]:** Gregory Ballas:
*The Ds/Dc values appears inconsistent with data description, please check them;

Response:
The Ds/Dc values have inherent error because many bands do not show offset in the thin section, therefore, the offset value has come from the field measurements. The error in the measurement will then be carried into the estimate. Additionally, the porosity is measured in 2D for the DB and the HR so there is error. However, the Ds/Dc values do suggest more CSBs with large shear offset and that has been addressed in the text. We thank the reviewer for their detailed analysis of the research to highlight this.

[Figure]

Figure 7. (a) – (h) SEM images of deformation bands and host rock – scale bar in all images is 100 µm. (a) overview image of a D3 SECB within the host rock. Diffuse edges to the band can be observed. (b) D3 SECB showing radiating fractures from grain contacts and larger grains within a fine-grained cataclastic matrix. (c) and (d) host rock surrounding D3 SECBs. Areas of crushed grains can be observed. Generally, the grains show minor fracturing and are well preserved with clear pore space between grains. (e) D3 reverse-sense CSB showing cataclastic matrix, fractured grains and minimal pore space. (f) D2 normal-sense CSB with less deformed grains preserving a larger grain size within a cataclastic matrix containing flakes and fine-grained, fractured material from the surrounding grains. (g) host rock surrounding D2 normal-sense CSBs with pockets of cataclasis from crushed grains but general preservation of grain size and porosity. (h) host rock surrounding D2 normal-sense CSB showing the alignment of pockets of cataclasis from the crushing of grains and filling in of pore space

565

[Figure]

representing a nascent deformation band. (i) and (j) Light microscopy images of the host rock. In the image, quartz overgrowths (QO) are highlighted. P = Potassium Feldspar, Q = Quartz, F = Plagioclase Feldspar.

**Commented [LE21]:** Gregoy Ballas:
*Extend the description of micro-mechanisms of D2 structures, the image you shown fig. 7f is too limited to clearly expose the deformation process and the band microgeometry you described in the text. Concerning these bands, you described a cataclastic process but this is not consistent with the negative relief they show on the field and their dark color. How explain that? It is not evident also in SEM images. Any important of clay (phyllo bands?) or disaggregation, or cementation (Organic Matter)? The observations of Fig. 6a-c rather argue for disaggregation bands.

Response:
Clearer figures have been made for the microstructure (Figs. 8, 12, 14). There is always significant grainsize reduction in the bands accommodated by cataclasites. Therefore, the bands are generally cataclastic, even when they transect layers with high clay content. See the new Fig. 6 in the revised manuscript, please.

**Figure 8. Microscopic images of D2 normal-sense shear bands and host rock. Dashed lines indicate the edge of deformation bands. [a], [b], [c], [d], [e], [f] images of deformation bands. The bands have diffuse edges and are characterised by reduced porosity and increased clay content. [g] and [h] are images of host rock surrounding deformation bands. Within the host rock microfractures are present. In addition, pockets of smaller grain size can be seen with higher clay content indicating the lithological heterogeneity of the unit. Q = quartz. C = zones of clay-sized particles. K = potassium feldspar. P = plagioclase.**

At the microscale, D2 bands are characterised by a reduction in grain size and porosity compared to the host rock which makes the bands easy to identify under the microscope ().Figs. 8a, 8b, 8c, 8d, 8e, 8f, 10). The bands show diffuse borders with the surrounding host rock (Fig. 8c). The grain size distribution of deformation bands generally shows positively skewed distributions with a lower median value compared to the host rock indicating a reduction in grain size due to cataclasis (Figs. 9ai & 9aii) (Fossen et al., 2007; Balsamo et al., 2010). The median grain

equivalent circular diameter in deformation bands is 37 µm compared with 57 µm in the host rock, showing a 35% reduction (Figs. 9ai & 9aii). Deformation bands also show a smaller range in grain size at 156 µm ranging from 9 µm to 165 µm compared to 231 µm in the host rock ranging from 8 µm to 239 µm, with the host rock preserving larger grains (see Supplement Section S.8). Deformation bands, compared to the host rock, are characterised by a high matrix content, due to grain size reduction, and a concentration of clay sized grains permitting the distinction between the two (Figs. 8a, 8b, 8c, 8d, 8e, 8f). The amount of matrix decreases from the centre to the outside of the bands and becomes almost non-existent in the host rock, which is dominated by intact grains, with/without intragranular fractures, zones of increased clay content, and pore space. There is an average absolute porosity reduction of 8% (from ca. 13% in the host rock to ca. 5%) in deformation bands (Fig. 10). This equates to 59% relative porosity reduction. Relict medium-sized pores (30-50 µm) are present within some bands, accounting for much of the remaining ca. 5% porosity.  Using8c). By using porosity reduction as a proxy for inelastic volumetric strain, we obtain a ratio of $D_S/D_C$ that ranges from ca. 20->100 indicative of CSBs with large shear offset (Soliva et al., 2013; Ballas et al., 2014; Soliva et al., 2016; Fossen et al., 2018). Generally, the darker coloured bands containing higher concentrations of clay have higher $D_S/D_C$ ratios as indicated by shear displacement observed at outcrop scale (Figs. 6c-g). Grain fracture is observed at grain contacts and within grains in both host rock and in deformation bands (. The e.g., Figs. 8d & 8h). The presence of microfractures in the host rock shows that the deformation is not solely concentrated within deformation bands. Due to a reduction in grain size, a significant reduction in porosity and a low density of microfractures, D2 deformation bands are classified as *cataclastic* CSBs (Antonellini et al., 1994; Mair et al., 2000; Fossen et al., 2007; Ballas et al., 2015).

[Figure]

**Commented [LE22]:** Gregoy Ballas:
*Extend the description of micro-mechanisms of D2 structures, the image you shown fig. 7f is too limited to clearly expose the deformation process and the band microgeometry you described in the text. Concerning these bands, you described a cataclastic process but this is not consistent with the negative relief they show on the field and their dark color. How explain that? It is not evident also in SEM images. Any important of clay (phyllo bands?) or disaggregation, or cementation (Organic Matter)? The observations of Fig. 6a-c rather argue for disaggregation bands.

Response:
There is always significant grainsize reduction in the bands accommodated by cataclasites. Therefore, the bands are generally cataclastic, even when they transect layers with high clay content. See the new Fig. 6 in the revised manuscript, please.

**Commented [LE23]:** Gregory Ballas:
*Qz overgrowths are not consistently described (wrong interpretation in the data description), add more precise observations. It could be of interest to constrained how evolve the mechanical properties and the petrophysics of the material.

Response:
We agree that quartz cementation has not been addressed properly. To alleviate this problem, substantial additional analyses such as high-resolution cathodoluminescence imaging would be required. Considering that this paper focuses on outcrop scale descriptions of DB and that the present BSE images permit to identify cataclasis as certainly the dominant deformation mechanism, we removed the images indicating the presence of quartz cement and this will be the focus of further research.

[revised manuscript text omitted]

**Commented [LE25]:** Gregory Ballas: correct sub-title to state whether or not DBs are associated with fault zones

**Commented [LE26]:** Gregory Ballas:
*Clearly separate DB in fault Damage Zone and DB out of fault DZ in the description of D3, as done within the following distribution description.

Response:
Figures have been added to the manuscript and previous image panels have been reordered to split the images into their respective deformation phases. The figures now align with the descriptions.

[revised manuscript text omitted]

**4.4.3.3.**

**Commented [LE27]:** Julien Bailleul:
- Proto-Deformation Bands (section 4.4.3.3): The description is quite short, could you give us more detail and explain more clearly why you interpret these as proto-DB?

Response:
Dataset removed. Not important for the study and can be focussed on with future study.

[revised manuscript text omitted]

**Commented [LE28]:** Gregory Ballas: *I recommend to show outcrop image mapping used for scan-line distribution analysis. Why not considered spacing > 20m?

Response: We had already included outcrop images of the maps used for scan-line distribution analysis in Figs. 17 & 18. Spacing >20 m has not been considered as it represents areas of no exposure along beaches associated with stream mouths. To include such data would influence the spacing results erroneously.

**Commented [LE29]:** Gregory Ballas:
*Explain how damage zone of fault thickness is defined.

with the hypotheses: bands observed adjacent to a fault plane have a positive correlation between spacing and distance from the fault plane and bands not observed adjacent to a fault plane have no relationship between spacing and distance.

**4.5.2.2.1. Fault Damage Zone Deformation Bands**

855 D2 normal and D3 reverse faults are bordered by deformation bands, forming a fault damage zone. Damage zone width varies from 0.1 cm to 272 cm for the analysed faults with associated thin section analysis.  Twelve faults with damage zone width >-10 cm were analysed for spacing statistics (e.g., Fig. 17). The average Pearson correlation coefficient for each analysed fault ranges from 0.07 to 0.62 with a combined average of 0.4 (see Supplement Section S.9 Fig. S9.5). 71% of D2 and 60% of D3

860 FDZ spatial distributions have Pearson coefficient distributions with a median <-0.5 indicating that there is no correlation between spacing and distance (e.g., Figs. 19b & 19d) (see Supplement Section S.9 Fig. S9.5). Analysis of spacing distributions shows that all faults have positively skewed spacing values (see Supplement Section S.9 Fig. S9.7). This data most closely matches synthetic images (d) and (f) (Fig. 15) indicative of a damage zone or spatial overprints.

[Figure]

**Commented [LE30]:** Gregory Ballas: *I recommend to show outcrop image mapping used for scan-line distribution analysis. Why not considered spacing > 20m?

Response: We had already included outcrop images of the maps used for scan-line distribution analysis in Figs. 17 & 18. Spacing >20 m has not been considered as it represents areas of no exposure along beaches associated with stream mouths. To include such data would influence the spacing results erroneously.

[revised manuscript text omitted]

**Commented [LE31]:** Gregory Ballas:
*Use figure 14 within the data description and remove the figure 15 (not used).

Julien Bailleul:
- Some figures deserve to be more cited in the text (e.g. fig. 15 is not cited at all) and some more precisely (i.e. what part of the figure is used).

Response:
We agree with this comment and have amended the text to have a better link between the text and figures.

Figure 15. Boxplots showing the frequency distribution of skewness values for the spacing distribution of deformation bands in fault damage zones and those located not adjacent to a fault (non-FDZ data). Periodic data shows less skew (0 being normally distributed). This correlated with synthetic images from Figure 10 showing regular spacing with noise. Fault data has a higher skew which also correlates with Figure 10 showing a damage zone with additional noise.

Table 1 Primary results from the analysis conducted at Castlepoint.

| | | Normal Fault Regime (D2) | Thrust Fault Regime (D3) | |
|---|---|---|---|---|
| **Macroscopic** | Macroscopic structures | Normal faults | Reverse faults | Folds |
| | Fault pattern | Polymodal | Polymodal | Bimodal |
| | Mean vector | Not applicable | Not applicable | 87/310 (axial plane) |
| | Corrected fault spacing (median) | 8.7 m | 9.3 m | Not applicable |
| **Mesoscopic** | Deformation band kinematics | Normal-sense | Reverse-sense | No observable offset/reverse-sense Shear enhanced compaction bands and Compactional shear bands |
| | Deformation band type | Compactional shear band | Compactional shear band | |
| | Location | Proximal to faults | Proximal to faults | Not associated with faults |
| | Dominant deformation band distribution | Aperiodic | Aperiodic | Periodic |
| | Fault pattern | Polymodal | Polymodal | Bimodal |
| | Mean vector | Not applicable | Not applicable | 32/110 & 30/280 |
| **Microscopic** | Absolute porosity reduction | 7.90% | 7.70% | 9.70% |
| | Grainsize reduction | 35% | 63% | 44% |
| | Cataclastic matrix | Present | Present | Present |
| | Band borders | Diffuse | Diffuse | Diffuse |
| | Microscale width | 1.23 mm | 1.53 mm | 0.70 mm |

**5. Discussion**

In the following, we will discuss how our observations of deformation bands from Castlepoint, New Zealand and compare them with previous studies and if similar relations to evaluate the control of host rock properties, tectonic regime are observed., and tectonic setting. The primary results are presented in Table 1. We shall first discuss the association of host rock properties and tectonic regime with deformation band kinematics, and the spatial distribution of bands before we concludeconcluding with a brief remarkremarks on the implications for fluid flow in this deformed rock sequence.package hosted within a subduction wedge.

**5.1. Band Kinematics, Orientation, and Microstructure**

Outcrop and microstructural data demonstrate that D2 bands, associated with a normal-fault regime, are dominantly Compactional Shear Bands (CSBs) with rare SECBs. SECBs are identified by their lack of normal-sense shear offset and higher angle to σ₁. In other. This is consistent with previous studies of deformation bands formed in extensional tectonic regimes, only CSBs and Shear Bands (SBs) are observed (Soliva et al., 2013; Ballas et al., 2014; Fossen et al., 2015; Soliva et al., 2016; Fossen et al., 2018). SECBs are not predicted to form during normal faulting because smaller mean stresses and

**Commented [LE32]:** Gregory Ballas:
*Develop the description of normal-sense SECB if you want to maintain the discussion concerning band type and distribution vs. tectonic regime l.720-736 and l. 776-784. I encourage you to do it, these normal-sense structures potentially formed by burial increase could be very interesting. If it is not possible, remove this part of the discussion.

Response:
We thank the reviewer for this comment. As aforementioned, these structures cannot be definitively identified using multiple methods of identification. However, CSBs can. Therefore, we have changed the text to show CSBs associated with horizontal extension.

[revised manuscript text omitted]

**Commented [LE33]:** Gregory Ballas suggested that figure 20 be made more realistic. We have kept the original figure because this is a schematic to show a possible order of events in any tectonic setting and would like this to be simple and understandable. The process of overprinting structures within damage zones is very complex and we do not believe that there is a great enough understanding of the process for the image to be made more realistic because in doing so, it would be less realistic.

Figure 20. A schematic to show the evolution of spatial distributions in normal- (a & b) and reverse-faulting (c & d) tectonic regimes. Initially, bands with an equal spacing form. Band spacing is theorised to be proportional to the layer thickness, as is observed with joints (Pollard and Aydin, 1988; Gross, 1993). Eventually, bands link across layers and a fault propagates (Aydin and Johnson, 1983). The fault movement causes the local stress field to be perturbed resulting in new band formation with orientations and kinematics matching that of the new local stress field.

**5.3. Tectonic control and the stress path**

Stress path modelling was employed in previous field studies to explain the kinematics and orientation of the observed deformation bands as a function of the tectonic regime (Saillet and Wibberley, 2010; Solum et al., 2010; Soliva et al., 2013; Ballas et al., 2014; Fossen et al., 2015; Soliva et al., 2016). We do not use this approach here because of the inherent complexity of the evolution of the state of stress within subduction wedges. Critical wedge theory demonstrates that the state of stress in subduction wedges is controlled by the geometry of the wedge (slope angle and dip angle of the subduction master fault), pore-fluid pressure, the frictional properties of the wedge, and the basal coefficient of the subduction thrust (Hu and Wang, 2006; Wang and Hu, 2006). The spatial distribution of seismicity along subduction faults implies that the wedge can be separated into a velocity-strengthening section (outer wedge) and a seismogenic velocity-weakening section (inner wedge) (Wang and Hu, 2006). This causes the inner and outer wedges to exhibit significant differences in the mechanical state during failure. The Whakataki Formation has travelled from the top of the outer wedge into the inner wedge and back to the surface. During this journey, the formation has experienced significant changes in a stress state, as clearly indicated by the broad structural inventory (Fig. 3). It is difficult to constrain this stress history from field observations. To emphasise this point, we recall that even within the inner wedge, one can obtain large vertical differences in stress regime, such as upper levels being under horizontal extension while lower levels of the wedge are simultaneously in horizontal contraction and vice versa (Hu and Wang, 2006). The stress regime in the subduction wedge is largely controlled by the basal friction coefficient of the subduction thrust and the pore-fluid pressure. Both parameters cannot be constrained reliably throughout the Miocene from our field observations. Therefore, to generate a comprehensive estimate of the stress path of the Whakataki Formation, at least 2D stochastic mathematical inverse modelling complemented by time-resolved geophysical data and strong geochronological constraints of basin and deformation history is required. This is beyond the scope of the current study.

However, one generic idea proposed in the literature surrounding stress path modelling may apply to our case study: during extension, the mean stress is smaller and the differential stress is higher than during contraction, resulting in an intersection of the yield envelope closer to the top of the cap and thus the formation of CSBs with high $D_s/D_c$ ratios, as observed in D2 bands (Soliva et al., 2013). Regional studies demonstrate that deposition continued following the deposition of the Whakataki Formation in the earliest Miocene throughout the remainder of the Miocene (Neef, 1992a; Neef, 1992b, 1995; Lee et al., 2002; Bailleul et al., 2007). Therefore, the overburden stress probably increased throughout D2. During D3, sediment thickness reached its maximum, and additional vertical thickening through D3 folding and thrust-stacking would have added to the vertical stress. It is, therefore, likely that the mean stress increased from D2 to D3. In addition, a change of the mechanical properties of sandstones from D2 to D3 can be expected, due to porosity reduction associated with ongoing compaction and cementation associated with burial and lithification. The combination of higher mean stresses and a bigger yield envelope in D3 compared to D2 could explain why we also observe more CSBs during D2 extension and more D3 SECBs during D3 contraction (Fig. 21), as proposed by Soliva et al. (2013), Ballas et al. (2014) and Soliva et al. (2016). This idea is certainly appealing in its simplicity. However, it remains to be tested with an inversion of the effective stress path for our study area, which is a challenging problem. Finally, while a static yield model can explain the critical stress state required for failure and the resulting failure angle and deformation band kinematics, it provides no information on the timing, rate of failure, spatial distribution or the number of deformation structures (Zhang et

**Commented [LE34]:** Julien Bailleul:
- It is said in the discussion that deposition of the Whakataki Fm occurred till the end of D2. As far as I know, it's not the case, the typically thin-bedded turbidites of that Formation being Waitakian-Otaian in age, so latest Oligocene to earliest Miocene.

Response:
Amended.

[revised manuscript text omitted]

* * *
**Commented [LE38]:** Julien Bailleul:
- References: some inconstancies are still present in the list and a few references could be added. Please, verify all the references to be sure that they are OK.

Response:
Agreed and modified. Again, we thank the reviewer for their attention to detail and it has made the work more reproducible.

[revised manuscript text omitted]

**Gregory Ballas**

**Comment**

This work is interesting as new field examples of deformation bands are described in the poorly investigated context of accretional prism. This work confirms some recent results concerning tectonic regime controlling deformation band patterns in sandstone and adds two uncommon patterns: (1) Localized faults and shear bands under contraction regime. This is the main result of this study and is potentially linked to the mecanostratigraphy especially marked in this geological context. This point deserves to be better developed with additional description of bed stratigraphy and fault architecture. (2) Distributed SECB under normal regime, potentially formed by burial (overloading).

However, these structures are not consistently described (not show in figures, some problems with dihedral angle, distribution is missing. . .) which affects the impact of this result. I underline that different consistent approaches are used (field mapping, microscopy, image analysis) and the number of data appears adequate to clearly characterize the fault/DB patterns. The literature appears also extensive and consistent with the paper aims.

Because of these reasons, this work deserves to be published in this special issue "Faults, Fractures and Fluid flow" of Solid Earth and could be of interest for any scientists dealing with mechanisms of deformation in porous materials or reservoir evaluations. However, this work contains numerous important issues in the methodology, the data description and the paper organization which have to be managed before any possible publication. I propose major revisions with numerous comments (see below and attached .pdf file with minor suggestions). A second review is certainly needed.

**Response**

We thank Gregory Ballas for reviewing this manuscript and for providing valuable critical insight and analysis that will allow us to improve and clarify the manuscript. We appreciate and are grateful that the reviewer thinks that this work deserves to be published within the special issue. Three main points are raised by the reviewer.

The first point refers to one of the main findings of the research that has not been given enough focus and description within the article. We thank the reviewer for pointing this out and have amended the manuscript to have a larger focus on the presence of localised faults and shear bands under a contractional regime. The role of mechanostratigraphy has been given more focus and discussion throughout the manuscript.

The second point highlights the documentation of SECBs within the normal fault regime. Upon reflection of the comments from the reviewer we have reassessed the data and have come to the conclusion that the structures do not represent SECBs but rather CSBs that have variable displacement along strike. While the structures do in places seem to have no apparent shear offset, this may be a sectioning effect. Furthermore, we observe that the lithology impacts the displacement associated with a structure with greater shear offset observed as bands propagate through clay-rich layers and reduced offset when propagating through shell-hash rich layers, for example. Therefore, in light of the comments from the reviewer, we have changed the results to show only CSBs present during horizontal extension, not SECBs, consistent with other studies examining deformation bands formed under an extensional regime (e.g., Rotevatn et al., 2008; Saillet and Wibberley, 2010; Solum et al., 2010; Soliva et al., 2013; Ballas et al., 2014; Soliva et al., 2016).

The third main point is regarding the methodology, data description, and paper organisation. In regard to methodology, issues surrounding analysis of complete datasets rather than individual outcrops has been addressed. We now discuss both scales and highlight a problem with only focussing on one of the two scales. The figures have been re-made to align with the order of

data description within the text. Data has been described in more detail where necessary and appropriate. However, in our opinion, most of the data is described to a level that is required for the key points of the article and more description would result in lengthening of an already long manuscript.

1575 **Main comments:**

**Comment**

*The introduction is in good shape with consistent references. However, the authors exposed the originality of their work with the fact that their study material is not Aeolian sandstones. That's right but I find the geologic context of accretional prism and permuting stress field rather original. At least, modify the text of this section to be consistent with the literature (sandstone of

1580 Provence are not Aeolian, maybe introduce also Nubian sandstone in Egypt? Or North Sea?);

**Response**

We have introduced the Nubian sandstone and sandstones hosted in the North Sea, in addition to providing more insight into the lack of studies focussing on DBs within subduction wedge settings. In doing so, we have also expanded the section referring to the lithology and how the lithology studied in this research is unique regarding DB studies, thereby further exposing the

1585 originality of the research.

**Comment**

*The section 2. Background presents lot of repetitions. I propose to remove or displace the 2.2, 2.3 and 2.4 and only preserve the 2.1. Geologic setting. Move some sentences of the 2.2 concerning the classification of deformation bands from micro mechanisms to the 1. Introduction, thus remove this section 2.2. and the table 1. Remove the 2.3 Spatial distribution already

1590 explained in the introduction. Move the 2.4 Conceptual mechanical model to the discussion (also fig. 3);

**Response**

Sections regarding classification of DBs have been shortened and incorporated into the introduction and a brief synopsis of 2.4 added into introduction. We did not, however, incorporate Figure 3 into the discussion as suggested because we believe that a figure to explain the cam-cap yield envelope is required in the introduction to provide a background into the idea that

1595 tectonic setting influences the band kinematics and orientation.

**Comment**

*I recommend to show outcrop image mapping used for scan-line distribution analysis. Why not considered spacing > 20m?

**Response**

We had already included outcrop images of the maps used for scan-line distribution analysis in Figs. 17 & 18. Spacing >20 m

1600 has not been considered as it represents areas of no exposure along beaches associated with stream mouths. To include such data would influence the spacing results erroneously.

**Comment**

The general shape of the spacing distribution is generally discuss using Pearson coefficient which is considered as the principal parameter to discuss band patterns. This approach could be interesting if accompanied by precise field observations and

1605 descriptions but it can introduce wrong interpretation if consistent field investigations are not done. I encourage you to develop description of mean band spacing, if possible from field measurement, and show several examples in figures. Remove the section concerning theoretical structure distribution (l.288-304 and section 4.5.1);

**Response**

We respectfully disagree with the reviewer and would like to point out that statistical approaches to characterising fracture

1610 patterns are common and very important in Structural Geology (Soliva and Benedicto, 2005; Tang et al., 2008; Rabinovitch et al., 2012; Hooker et al., 2013; Guo et al., 2017; Healy et al., 2017; Laubach et al., 2018; Sanderson and Peacock, 2019). Most commonly, the aim of reporting a mean is to determine the central tendency of a distribution or its most probable, common value. This approach makes a lot of sense when one considers constant spacing with random noise (which should be normally

distributed). However, it is not useful when considering spacing that varies systematically with position, especially when this variation is nonlinear. Our simple statistical approach can be seen as a simplified version of that of "Sanderson, D. J. and Peacock, D. C.: Line sampling of fracture swarms and corridors, Journal of Structural Geology, 122, 27-37, 2019.". It is designed to test if a deformation band distribution is a function of position (mean not useful) or if it is constant with random noise (mean useful). We have kept the section regarding theoretical structure distribution in the text because it is needed to understand our simplified statistical approach to the mapping and can act as a check for other researchers to compare their datasets with.

In terms of additional figures: we provide three representative examples for bands showing periodic and aperiodic spacing. While many others were measured (see Figs. 9 & 19 for statistical results), we did not feel it appropriate to include all examples within the manuscript. If the editorial team indicates a requirement for such data, it can be added to an appendix.

**Comment**

*The Pearson coefficient is also use to describe fault patterns (bimodal – polymodal). Again, this approach can introduce major wrong interpretation as a function of the measurements were done on the field. I recommend to better developed observations and field description before to use this statistical approach;

**Response**

Regarding fault patterns, we have used the approach of Healy and Jupp (2018) to test if the fault orientation distribution is bimodal. This is a published, mathematically sound workflow that can be compared to other datasets and is reproducible and reliable. It also minimises observer bias because it is quantitative.

Unfortunately, the reviewer does not state explicitly which specific problems can arise from the use of the objective statistical methods employed in our paper. Thus, it is difficult to respond to the criticism concisely. We are certainly aware of, and explicitly mention and discuss, the most important source of sampling bias in our study: exposure bias. The hinterland outcrops are rare and poorly preserved. Therefore, our work is restricted to the coast which has excellent outcrop. However, the coast is sub-parallel to the dominant strike of major thrusts and folds. As a result, it is expected and not surprising that we observe fewer D3 thrust (mentioned in the paper in section 4.4.2.) than D2 faults, which strike at a high angle to the coast. In addition, most of our study sites sit on the back limb of a large D3 syncline, and because of this particular exposure bias, we cannot study how deformation band and fault distribution varies with position across large-scale folds. Moreover, the coastal outcrops are (fairly unstable) cliffs bordered by rock platforms. So not every single bed featuring deformation bands yields good opportunities to document them in 3D. Nevertheless, exposure bias is a common problem for all geological research. At the coast, exposure is > 80%, which we consider excellent. We do not see how the use of objective statistical methods in the analysis of fault/DB orientation data and line maps constitutes a major problem.

**Comment**

*The microfracture density have to be quantify from surface mapping (on SEM image) and not from scanline orientated normal to the band. This introduces an important bias as micro-cracks strike along force chains with specific angle to the bands;

**Response**

This dataset will be removed.

**Comment**

*The description of structures is confusing as the text is following a chronological order, whereas figures are classified by type of data (macro, micro, grain size analysis, petrophysics). I encourage you to clearly separate data of your 3 events D1, D2, D3 to match with the text description (just keep the figure 9a on porosity with the total dataset for comparison);

**Response**

We agree with this comment and have changed the figures accordingly. We thank the reviewer for this comment as it has enabled us to better describe the data with clearer figures.

**Comment**

*Indicate more precisely which part of each figure (a, b, c, d, e. . .) is concerned by citation in the text will help. Start your figure in the consistent chronology (generally D3 structures are firstly described in figure). I encourage you to add some indications on the figure.

1660 **Response**

Agreed and implemented.

**Comment**

*The Ds/Dc values appears inconsistent with data description, please check them;

**Response**

1665 The Ds/Dc values have inherent error because many bands do not show offset in the thin section, therefore, the offset value has come from the field measurements. The error in the measurement will then be carried into the estimate. Additionally, the porosity is measured in 2D for the DB and the HR so there is error. However, the Ds/Dc values do suggest more CSBs with large shear offset and that has been addressed in the text. We thank the reviewer for their detailed analysis of the research to highlight this.

1670 **Comment**

*Extend the description of micro-mechanisms of D2 structures, the image you shown fig. 7f is too limited to clearly expose the deformation process and the band microgeometry you described in the text. Concerning these bands, you described a cataclastic process but this is not consistent with the negative relief they show on the field and their dark color. How explain that? It is not evident also in SEM images. Any important of clay (phyllo bands?) or disaggregation, or cementation (Organic

1675 Matter)? The observations of Fig. 6a-c rather argue for disaggregation bands.

**Response**

Clearer figures have been made for the microstructure (Figs. 8, 12, 14). There is always significant grainsize reduction in the bands accommodated by cataclasites. Therefore, the bands are generally cataclastic, even when they transect layers with high clay content. See the new Fig. 6 in the revised manuscript, please.

1680 **Comment**

*Qz overgrowths are not consistently described (wrong interpretation in the data description), add more precise observations. It could be of interest to constrained how evolve the mechanical properties and the petrophysics of the material.

**Response**

We agree that quartz cementation has not been addressed properly. To alleviate this problem, substantial additional analyses

1685 such as high-resolution cathodoluminescence imaging would be required. Considering that this paper focuses on outcrop scale descriptions of DB and that the present BSE images permit to identify cataclasis as certainly the dominant deformation mechanism, we removed the images indicating the presence of quartz cement and this will be the focus of further research.

**Comment**

*Clearly separate DB in fault Damage Zone and DB out of fault DZ in the description of D3, as done within the following

1690 distribution description.

**Response**

Figures have been added to the manuscript and previous image panels have been reordered to split the images into their respective deformation phases. The figures now align with the descriptions.

**Comment**

1695 *Explain how damage zone of fault thickness is defined.

**Response**

This has been addressed in lines 660-661.

**Comment**

*Use figure 14 within the data description and remove the figure 15 (not used).

1700 **Response**

Done as suggested.

**Comment**

*Develop the description of normal-sense SECB if you want to maintain the discussion concerning band type and distribution vs. tectonic regime l.720-736 and l. 776-784. I encourage you to do it, these normal-sense structures potentially formed by

1705 burial increase could be very interesting. If it is not possible, remove this part of the discussion.

**Response**

We thank the reviewer for this comment. As aforementioned, these structures cannot be definitively identified using multiple methods of identification. However, CSBs can. Therefore, we have changed the text to show CSBs associated with horizontal extension.

1710 **Comment**

*The mechanical approach exposed in figure 17 is not enough constrained to be consistently discuss (both stress path and yield envelope are not estimate from data). The hypothesis of compaction between D2 and D3 appears inconsistent with description. Think about strengthening by the D2 band pattern or cementation process to explain a potential increase of the yield envelope. However, the change of boundary stress conditions (extensional regime – contractional regime) and the presence of localized

1715 faults could explain this change of band properties from D2 to D3.

**Response**

In lines 893-914 we discuss in detail why it is very challenging to determine the stress path for our rocks of interest and state that it is beyond the scope of our study. We explicitly highlight that the simplistic model discussed in Fig. 21 is purely speculative and requires further testing based on an inversion of the stress path. All we do is to state that there is a non-zero

1720 possibility that the mean stress and the strength of the rock package increase from D2 to D3, and thus one could be tempted to apply similar arguments as those in literature to explain the difference in D2 and D3 bands. This issue is already discussed at great length and with due care in lines 904 to 913. We do not believe that it requires further discussion.

**Comments from within the text**

The reviewer has provided numerous other references to add into the text and to broaden the scope of the research. We have

1725 implemented these changes and read the relevant papers to improve the manuscript and thank the reviewer for adding the references to the comments.

The reviewer has identified the grouping of clusters to represent one band as a bias. It is certainly a simplification. However, as shown now in Figs. 6 & 13, a band can turn into a cluster downdip and back into a single band again. This is a function of host rock and now mentioned in the text. We follow the approach of Main et al. (2000) and consider clusters as a single band.

1730 Any error introduced by this averaging is usually smaller than the pattern variability captured by using the scanline approach.

The reviewer has asked why we normalised spacing data. We did so to enable analysis and comparison of multiple outcrops.

The reviewer suggested that figure 20 be made more realistic. We have kept the original figure because this is a schematic to show a possible order of events in any tectonic setting and would like this to be simple and understandable. The process of overprinting structures within damage zones is very complex and we do not believe that there is a great enough understanding

1735 of the process for the image to be made more realistic because in doing so, it would be less realistic.

**Julien Bailleul**

**Comment**

The study proposed by Elphick et al. appears to perfectly fit with this Special Issue of SOLID EARTH. The paper describes
for the first time deformation bands in synsubduction turbidites of the Hikurangi margin (New Zealand) and brings insights
concerning their relation to deformation episodes that have affected the subduction wedge. This study is original in New
Zealand and for such a geological context and relies on recent development concerning the understanding of deformation
bands. In my opinion, emphasis should be given on: 1) the geological context, and above all, 2) the turbidites which are
particular host rocks for deformation bands. The paper is well written with globally a well-organized and clear structure even
if some specific and technical points have to be amended to improve the manuscript (see next section and annotation in the
PDF file).

**Response**

We thank Julien Bailleul for the constructive comments and critical analysis of the research. The comments from the reviewer
have enabled us to significantly improve the manuscript. The associated changes of the manuscript are described in detail in
the following.

**Main comments**

**Comment**

- As accretion did not occurred all across the thrust wedge, I rather prefer to avoid the use of accretionary prism to talk of the
entire deformed area between the trench and the Forearc basin (please, see also my point of view on that in the PDF file of the
manuscript).

**Response**

Thank you for pointing out this semantic oversight. We have amended the manuscript accordingly.

**Comment**

- An absolute porosity reduction of ca. 10% (from ~ 20% to 10%) is given in the conclusions whereas between 5 and 14% is
given in the abstract. Also, c.a. 9 m is given in the conclusions while 10 m is given in the abstract for the spacing between
faults. I think it can be clearer to give the same values in both the abstract and conclusions.

**Response**

Thank you for alerting us to these inconsistencies. They have been corrected.

**Comment**

- The Background section could be clarified with a better separation between what concern the geological context and what
concern deformation bands.

**Response**

In response to comments from Reviewer 1, we have changed and reduced the background and the main points have been added
into the introduction. We believe that this change also accommodates the comment from this reviewer.

**Comment**

- Proto-Deformation Bands (section 4.4.3.3): The description is quite short, could you give us more detail and explain more
clearly why you interpret these as proto-DB?

**Response**

Dataset removed. Not important for the study and can be focussed on with future study.

**Comment**

- It is said in the discussion that deposition of the Whakataki Fm occurred till the end of D2. As far as I know, it's not the case,
the typically thin-bedded turbidites of that Formation being Waitakian-Otaian in age, so latest Oligocene to earliest Miocene.

**Response**

This has been amended in lines 895-897.

1780 **Comment**

- The section 5.4. Implications for fluid flow is very general and could be consolidated a little bit, notably incorporating references to papers that specifically deals with fluids on the Hikurangi margin – that literature is abundant so only a few key papers could be cited: (e.g. . . .. See annotation in the manuscript for suggestion of references). That would give weight to the consequences of the compartmentalization demonstrated in the paper on reservoir properties and for pressure regimes.

1785 **Response**

This section has been extended and given more focus surrounding literature about the Hikurangi margin to contextualise the study while still making it applicable to different settings. The southern Hikurangi subduction wedge has been given more focus (e.g., lines 938-950).

**Comment**

1790 - The paper could gain from a last figure acting as a synthesis summarizing most of the observations and results, on a 3D block for example. Such a figure could highlight the work and make it more visible.

**Response**

We already include a 3D block diagram for the deformation phases in Figure 3, which focuses on the macro-structures and includes an inset for DB for D3. The orientation of DBs for D2 share the same orientations as D3 DBs hosted in damage zones.

1795 This relationship has been added to the figure caption for Figure 3.

**Comment**

- Some figures deserve to be more cited in the text (e.g. fig. 15 is not cited at all) and some more precisely (i.e. what part of the figure is used).

**Response**

1800 We agree with this comment and have amended the text to have a better link between the text and figures.

**Comment**

- References: some inconstancies are still present in the list and a few references could be added. Please, verify all the references to be sure that they are OK.

**Response**

1805 Agreed and modified. Again, we thank the reviewer for their attention to detail and it has made the work more reproducible.

Throughout the document, grammatical errors that have been identified by both reviewers have been amended accordingly.

**AC References**

Ballas, G., Soliva, R., Benedicto, A., and Sizun, J.-P.: Control of tectonic setting and large-scale faults on the basin-scale distribution of
1810 deformation bands in porous sandstone (Provence, France), Mar. Pet. Geol., 55, 142-159, 2014.
Guo, L., Latham, J.-P., and Xiang, J.: A numerical study of fracture spacing and through-going fracture formation in layered rocks, International Journal of Solids and Structures, 110, 44-57, 2017.
Healy, D., Rizzo, R. E., Cornwell, D. G., Farrell, N. J. C., Watkins, H., Timms, N. E., Gomez-Rivas, E., and Smith, M.: FracPaQ: A MATLAB™ toolbox for the quantification of fracture patterns, Journal of Structural Geology, 95, 1-16, 2017.
1815 Hooker, J. N., Laubach, S. E., and Marrett, R.: Fracture-aperture size—frequency, spatial distribution, and growth processes in strata-bounded and non-strata-bounded fractures, Cambrian Mesón Group, NW Argentina, Journal of Structural Geology, 54, 54-71, 2013.
Laubach, S. E., Lamarche, J., Gauthier, B. D., Dunne, W. M., and Sanderson, D. J.: Spatial arrangement of faults and opening-mode fractures, Journal of Structural Geology, 108, 2-15, 2018.
Rabinovitch, A., Bahat, D., and Greenberg, R.: Statistics of joint spacing in rock layers, Geological Magazine, 149, 1065-1076, 2012.
1820 Rotevatn, A., Torabi, A., Fossen, H., and Braathen, A.: Slipped deformation bands: A new type of cataclastic deformation bands in Western Sinai, Suez rift, Egypt, Journal of Structural Geology, 30, 1317-1331, 2008.
Saillet, E. and Wibberley, C. A.: Evolution of cataclastic faulting in high-porosity sandstone, Bassin du Sud-Est, Provence, France, Journal of Structural Geology, 32, 1590-1608, 2010.
Sanderson, D. J. and Peacock, D. C.: Line sampling of fracture swarms and corridors, Journal of Structural Geology, 122, 27-37, 2019.

1825    Soliva, R., Ballas, G., Fossen, H., and Philit, S.: Tectonic regime controls clustering of deformation bands in porous sandstone, Geology, 44, 423-426, 2016.

Soliva, R. and Benedicto, A.: Geometry, scaling relations and spacing of vertically restricted normal faults, Journal of Structural Geology, 27, 317-325, 2005.

Soliva, R., Schultz, R. A., Ballas, G., Taboada, A., Wibberley, C., Saillet, E., and Benedicto, A.: A model of strain localization in porous

1830    sandstone as a function of tectonic setting, burial and material properties; new insight from Provence (southern France), Journal of Structural Geology, 49, 50-63, 2013.

Solum, J. G., Brandenburg, J., Naruk, S. J., Kostenko, O. V., Wilkins, S. J., and Schultz, R. A.: Characterization of deformation bands associated with normal and reverse stress states in the Navajo Sandstone, Utah, AAPG bulletin, 94, 1453-1475, 2010.

Tang, C. A., Liang, Z. Z., Zhang, Y. B., Chang, X., Tao, X., Wang, D. G., Zhang, J. X., Liu, J. S., Zhu, W. C., and Elsworth, D.: Fracture

1835    spacing in layered materials: A new explanation based on two-dimensional failure process modeling, American Journal of Science, 308, 49-72, 2008.

---

## Author Response (AR3)

Author Comments to editor Prof. Fabrizio Balsamo

**Editor Comment**

This new version is significantly improved with respect to the first submitted version, and I certainly support the publication of your paper. Hoverer, there are some minor issues that should be addressed before final acceptance for publication. They mostly pertain with some figures and text organization.

The major issue, in my view, pertains the data description and the use of figures. To describe your multidisciplinary results, you follow a criterion of phase of deformation: D1, then D2 and finally D3. One of the reviewer asked for a description of results based on type of data (field data, mesoscale pattern of D1-3, microtextural features of D1-3 DBs, grain size data obtained with image analyses, DBs spacing, and so on. I think that this suggestion was sound and probably would have forced you to summarize both text and figures, but you decided to keep your data presentation according to the phase of deformation. This is not a problem per se, and I support this decision. But please note that this way of data presentation needs very clear description and the use of figures in a systematic and chronological order. In the text, I found that (1) some figures are not fully descried despite composed of several images/panels, and (2) often the figures description jump from one figure to another (e.g., from 3 to 7, from 8 to 13, and so on… in many parts). To this end, I made several comments in the annotated PDF to simplify the readability of the manuscript, to follow strictly the figure numbering, and to reduce the numbers of figures (indeed, 21 figures are so much).

**Author Comment**

We thank the editor very much for his constructive and supportive feedback. Our rationale for retaining the original manuscript structure according to deformation phase reflects our paper's focus on testing if extensional (D2) and contractional (D3) tectonic regimes result in characteristic deformation-band types and spatial distributions, as described in the literature (e.g., Ballas et al., 2014). We apologise that the previous document included some minor formatting and structural problems in regard to figures, their order and their captions. We have addressed all concerns highlighted by the editor as described in detail below. In particular, we have removed images from panels that show similar features to improve readability, e.g., figure 8 now has only 2 images within a panel instead of 8. Furthermore, images have been reordered to match the text. For example, the original figure 7 has switched with the original figure 6 to match the flow of the text.

**Editor comment**

Introduction

It was a pleasure to read such extensive introduction which I think is very useful for researchers approaching to DBs studies. The literature cited on deformation band faulting is very broad, but I found that you could mention also other case study areas (listed in the annotated PDF). I think you can also include such papers in the nice review introduction you have done. Apologize for self-citation, so feel free to decline the invitation to extend the literature if you think such contributions are not relevant.

**Author comment**

Thank you very much for drawing our attention to the suggested papers. They are now included in the introduction (lines 75, 107, 112, and 145).

**Editor comment**

Methods
Line 268-274. I am wondering why you have not measured directly the true spacing between DBs in both faults (FDZ) and "background" (non-DFZ) domains. Which is the advantage of image analysis from outcrop images? Wouldn't be more easy use real spacing measured in the field to compare your own data from different outcrops, and also with other datasets from literature? People working in DBs often measure N° elements per meter, or cumulative frequency approaching faults. Maybe a sentence in the Method is needed.

**Author comment**

Thanks a lot for this opportunity to clarify our choice of methodology. Our use of image analysis is motivated by the following:
[a] Minimising geometric bias: the studied rock package is quite poorly consolidated. As a result, individual beds exposed along cliffs often have a surface which is not flat. 2D DB spacing is best measured in a plane orthogonal to the average strike of the bands. Photographic imaging can easily achieve this goal if the photo is taken in the appropriate orientation while manual counting along a tape measure becomes difficult if the tape measure does not rest firmly on the outcrop surface.
[b] Minimising sampling bias: on digital DB maps, we can easily analyse hundreds of scan lines in a second. The frequency distributions of spacing presented in this paper clearly demonstrate that the spacing distribution, to a degree, depends on the location of the scan line. Measuring hundreds of scan lines manually in the field is prohibitively time-consuming.
[c] We employ a simplified version of well-established statistical approaches to test if spacing varies as a function of location or not. This is a core aim of the paper, and the use of statistical methods for this purpose is clearly more objective and reproducible than qualitative decisions made by fieldworkers alone. This methodology is most readily applied to digital DB maps.
[d] Due to the poorly consolidated nature of the rocks and their exposure along cliffs in an intertidal domain, many of the outcrops are very unstable. In our repeat visits, we found that cubic metres of previously examined outcrops had fallen into the sea over the course of a few months. As a result, we decided to try to restrict working time near cliffs to a necessary minimum. Obtaining careful photographic mosaics is much faster and safer in these circumstances than manual measuring of DB spacing.

We have now included these motivations into the manuscript in lines 236-244.

**Editor comment**

Line 275-290. I acknowledge the value to have synthetic images with different types of DBs spacing pattern, but this further analyses risk to lengthen the paper without adding significant or relevant data. Maybe can be moved in an appendix?

**Author comment**

Thank you, we implemented this suggestion as requested. We link to the supplementary section in lines 275-277.

**Editor comment**

Results
Results are described according to the deformation phase. To keep this way of data presentation, you should describe figures progressively as much as possible. Probably, some images in several figures can be removed since they show similar features. Some examples below and in the annotated PDF.

**Author comment**

Thank you, this has been addressed, and redundant images have been removed from panels (for example: Figs, 8, 12, 14). Additionally, Figs. 5a and 5b have been removed from the document as the original Figs. 5c, d, e can document what is required to show overprint within the area. Thus, additional figures are redundant.

**Editor comment**

Line 610-640 and figure 15. This synthetic spacing chapter can be moved to the online repository?

**Author comment**

Thank you, we have implemented this suggestion as requested. We refer to the supplement in lines 611-617 with a synthesised explanation of how the data can be interpreted.

**Editor comment**

Figure 2. In the three geological cross sections, write AA', BB', CC' in green bold, as in the main map. This will facilitate the connection between the map and the cross sections.

**Author comment**

Implemented as suggested.

**Editor comment**

Figure 3. Such figure is mentioned in the Study area section. I am wondering if this figure pertains to the Result section, since new original data are presented.

**Author comment**

Originally, we placed the figure earlier in the manuscript to give an idea of the orientations. As proposed, we now place it in the results section to increase readability when referring to orientation data within the text. Thank you for the suggestion.

**Editor comment**

Figure 5a is mentioned after 5c, d, e. Figure 5b is never mentioned in the text. You may re-organize the figure panels and

text accordingly to describe features chronologically in order of appearance in the manuscript. This suggestion is valid for several other figures.

**Author comment**

5a and 5b have been removed from the manuscript as the overprints that they show can be explained in the text, and figure 5c, d, and e can support such overprinting relationships. Original figure 6 has been reordered to match with the flow of the text. Additionally, images have been removed from panels, (Figs. 8, 12, 14) to improve readability and flow between images and text.

**Editor comment**

Figure 7. The dotted rectangle in (a), that indicate (b), has a different shape respect image in (b). One is vertical, the other horizontal. Please correct. Further, (b) is stated as a fault core, but I am wondering if this is a real 30 cm thick fault core (for a fault with 1 m offset). Very unlike. Then, I think this is not a classical fault core. Seems like a high strain zone, indeed the fault zone, with multiple slip surfaces and distributed narrow "fault cores". Very nice structures. Could you please clarify?

**Author comment**

Thank you. We modified the marker rectangle accordingly and amended our nomenclature, as suggested.

**Editor comment**

Figure 8. In line 420 and line 428 the "Figures 8a to 8 f" (6 images) are mentioned all together, without distinguish them or describing them progressively. If they show the same feature (grain size reduction within D2 DBs) why you show 6 images and not only one image or two? You may save space and simplify the readability of the text. Same comment for the other microtextural figures.

**Author comment**

Thank you for your comment. For all microtextural figures the number of images within the panel has been reduced to show a deformation band and an image of host rock.

**Editor comment**

Figure 10 is described after figure 11. So, switch the figures.

145

**Author comment**

We discuss figure 10 in regard to D2 structures in line 442 before figure 11 is discussed. Hence, we have kept the order of these figures.

150

**Editor comment**

Figures 17, 18 and 19 can be merged in one figure? You can show one example of DBs associated to fault, one example of DB non associated to faults (background) and all the diagrams in fig. 19. In this way you summarize everything in one figure.
155 The rest of the figures can be moved in the supplementary materials. Or at least figures 17 and 18 can be merged in one without losing resolution?

**Author comment**

Implemented as suggested.

160 **References**

[revised manuscript text omitted]

a — Increased porosity in the host rock outside of the band. · Diffuse edges to bands. 200 um

b — Cataclastic matrix 200 um

c — Relict pore · P 100 um

d — K · Q · C · Microfractures within grains hosted in the DB · Cataclastic matrix 100 um

e — Q · View of a shear band core. Larger grains are preserved within a fine grained matrix comprising fractured grains and clay. 100 um

f — Clear differentiation between the DB and the host rock from the difference in pore space. 200 um

g — Host rock with pockets of higher clay content and fractured grains · C 300 um

h — Microfractures in grains of the host rock · Q · K 100 um

[revised manuscript text omitted]
. ~~In the synthetic spacing images presented in Supplementary Section S10, it is shown that overprinting deformation events characterised by morphologically similar deformation bands, but with a different distribution can be identified using spacing statistics. In situations where two different deformation events affect the same bed, the Pearson value can range from -1 to 1. Histograms to show the absolute spacing will have a normal distribution and may be characterised by a secondary peak indicating a background equal spacing (see Supplementary Section Figure S10.9e). For interpretation of our field data, we assume a positive correlation between band spacing and spatial location if the Pearson value is > 0.5. Values greater than 0.5 would be expected in a damage zone, while values ranging from -0.5 to 0.5 are expected to occur far from faults.~~

[revised manuscript text omitted]